# RIEMANNIAN TRANSFORMATION LAYERS FOR GENERAL GEOMETRIES

## ABSTRACT

Recently, deep neural networks on manifold-valued representations have garnered significant attention in various machine learning applications. Several studies have attempted to generalize traditional Euclidean transformation layers, such as Fully Connected (FC) and convolutional layers, to non-Euclidean geometries. However, the previous approaches typically focus on a few selected manifolds and rely on the specific properties of the target manifold. In this work, we propose a theoretical framework for constructing Riemannian FC and convolutional layers over general geometries, providing broader applicability. Utilizing this framework, we design convolutional networks across five distinct geometries of the Symmetric Positive Definite (SPD) manifold, as well as networks under two Grassmannian perspectives. Extensive experiments demonstrate that the proposed Riemannian convolutional networks significantly outperform existing SPD and Grassmannian networks.

## 1 INTRODUCTION

Recently, deep neural networks on Riemannian manifolds have achieved remarkable success across a wide range of applications (Huang et al., 2017; Huang & Van Gool, 2017; Huang et al., 2018; Ganea et al., 2018; López et al., 2021; Huang et al., 2022; Nguyen, 2022a; Shimizu et al., 2020; Kobler et al., 2022; Wang et al., 2024b; Ju et al., 2024). Commonly encountered manifold-valued representations include spherical, hyperbolic, Symmetric Positive Definite (SPD), and Grassmannian manifolds, as well as matrix Lie groups like special orthogonal groups, to name a few. Due to the closed-form expressions of their Riemannian operators, such as geodesics, exponential and logarithmic maps, and parallel transport (PP), various fundamental building blocks have been extended to different manifolds, including normalization (Chakraborty, 2020; Brooks et al., 2019; Kobler et al., 2022; Chen et al., 2024b), attention (Gulcehre et al., 2019; Pan et al., 2022; Wang et al., 2024a), residual blocks (Katsman et al., 2024), and Multinomial Logistic Regression (MLR) (Ganea et al., 2018; Nguyen & Yang, 2023; Chen et al., 2024a;c).

**Research problem.** As transformation layers are fundamental building blocks in Euclidean deep networks, several works have designed Riemannian counterparts on different geometries. Huang & Van Gool (2017); Huang et al. (2017; 2018) developed ad hoc transformation layers for SPD, special orthogonal groups, and Grassmannian manifolds, respectively. Ganea et al. (2018) performed hyperbolic transformations via the tangent space. However, these transformations do not fully respect the underlying Riemannian geometries. To remedy this limitation, Shimizu et al. (2020) extended Fully Connected (FC) and convolutional layers into hyperbolic spaces based on latent Poincaré geometries. Additionally, Nguyen et al. (2024) extended these layers to SPD manifolds using gyro structures induced by three Riemannian metrics. Nonetheless, their methods strongly rely on specific properties, such as hyperbolic geometries and gyro structures, restricting their applicability. Furthermore, Chakraborty et al. (2020) extended convolution by the weighted Fréchet mean. Although the framework can be applied to various geometries, unlike traditional Euclidean convolution, it cannot change the manifold's dimensionality, limiting its flexibility. Therefore, a general and flexible framework for building FC or convolutional layers over diverse geometries remains unsolved.

**Proposed solution.** We propose a framework for constructing Riemannian FC and convolutional layers that naturally capture the underlying geometry. First, we introduce the Riemannian FC layer by reformulating the Euclidean FC layer. Since convolution is an extension of the FC layer, we derive the Riemannian convolution as a product of the proposed Riemannian FC layer. Unlike previous FC layers tailored for specific manifolds, our Riemannian layers depend solely on Riemannian operators, such as exponential and logarithmic maps, which have closed-form expressions across

various manifolds. This allows our framework to enjoy broader applicability. Moreover, when the latent geometry is reduced to Euclidean space, our Riemannian FC layer recovers the standard Euclidean FC layer.

After presenting the general framework, we provide concrete manifestations of our Riemannian FC and convolutional layers on SPD manifolds under five distinct Riemannian metrics, and Grassmannian manifolds under the Projector Perspective (PP) and OrthoNormal Basis (ONB) perspective. Our SPD FC layers also incorporate the previous three gyro SPD FC layers, the derivation of which requires additional gyro structures. Besides, our framework offers an intrinsic geometrical interpretation to understand the trick of generating manifold embeddings from the Euclidean feature as a Riemannian FC layer. Finally, we compare the performance of our Riemannian convolutional networks against existing manifold-specific networks on SPD and Grassmannian spaces, demonstrating that our networks significantly outperform current Riemannian networks. In summary, our **main contributions** are as follows:

1. **Generalization of convolution and FC layers to Riemannian manifolds.** We introduce a principled generalization of FC and convolutional layers to general Riemannian manifolds. The proposed framework relies solely on Riemannian operators such as exponential and logarithmic maps, faithfully respecting the underlying geometry.

2. **Building five SPD and two Grassmannian neural networks.** Empirically, we apply our theoretical framework to five geometries of the SPD manifold and two perspectives of the Grassmannian. Extensive experiments comparing our methods with existing SPD and Grassmannian networks demonstrate the superiority of our approach.

3. **Flexible latent geometry variations.** Our method enables direct variation of the latent geometry in neural networks without the need for specialized operations on a per-manifold basis. This novel flexibility allows for direct comparison of different geometric representations within the same network architecture.

**Main theoretical results:** Thm. 4.2 presents the expression of our Riemannian FC layer under general geometries. Prop. 4.4 indicates that our Riemannian FC layer is a natural generalization of the Euclidean FC layer, as it recovers the Euclidean FC layer under the Euclidean geometry. Sec. 4.2 discusses the Riemannian convolution based on the product of the Riemannian FC layer. Sec. 4.3 discusses optimizing the parameters involved in the Riemannian FC and convolutional layers. Thm. 5.1 showcases our framework on the SPD manifold under five Riemannian metrics, while Thms. 6.1 and 6.2 introduce the Grassmannian FC layers under the ONB and PP perspective, respectively. As shown in Tab. 1, the existing three gyro SPD FC layers are incorporated by our SPD FC layers. Besides, Tab. 2 compares our Grassmannian FC layers against other Grassmannian transformation layers, highlighting that our layers offer greater flexibility in altering dimensionality across different perspectives. Prop. 7.1 explains the widely used manifold embedding trick as a special instantiation of our Riemannian FC layer. Due to page limits, all proofs are placed in App. K.

## 2 PRELIMINARIES

Due to page limits, we provide only the essential background here. A review of relevant Riemannian ingredients across different geometries can be found in App. B. For better readability, a table of notations is presented in Tab. 5.

**The SPD manifold.** Let $\mathcal{S}_{++}^n$ be the set of $n \times n$ symmetric positive definite (SPD) matrices. As shown by Arsigny et al. (2005), $\mathcal{S}_{++}^n$ is an open submanifold of the Euclidean space $\mathcal{S}^n$ of symmetric matrices. There are five kinds of popular Riemannian metrics on $\mathcal{S}_{++}^n$: Affine-Invariant Metric (AIM) (Pennec et al., 2006), Log-Euclidean Metric (LEM) (Arsigny et al., 2005), Power-Euclidean Metrics (PEM) (Dryden et al., 2010), Log-Cholesky Metric (LCM) (Lin, 2019), and Bures-Wasserstein Metric (BWM) (Bhatia et al., 2019). Various applications involves the SPD features (Huang et al., 2017; Brooks et al., 2019; Wang et al., 2020; López et al., 2021; Nguyen, 2021; 2022b; Kobler et al., 2022; Pan et al., 2022; Bonet et al., 2023; Chen et al., 2021; 2023; Wang et al., 2024b). As shown by Chen et al. (2024b;c;a); Nguyen et al. (2024), the optimal metric usually differs across different tasks.

**The Grassmannian.** The Grassmannian is the set of $p$-dimensional subspaces of an $n$-dimensional vector space (Tu, 2011, Problem 7.8). It has two common matrix representations (Bendokat et al., 2024): the Projector Perspective (PP), where each element is embedded as an $n \times n$ symmetric

matrix, and the OrthoNormal Basis (ONB) perspective, which is the quotient of the Stiefel manifold $\mathrm{St}(p,n)$. Formally, these two perspectives are defined as

$$\text{Projector Perspective (PP): } \widetilde{\mathrm{Gr}}(p,n) = \{P \in \mathcal{S}^n : P^2 = P, \mathrm{rank}(P) = p\},$$

$$\text{ONB perspective: } \mathrm{Gr}(p,n) = \{[U] : [U] := \{\widetilde{U} \in \mathrm{St}(p,n) \mid \widetilde{U} = UR, R \in \mathrm{O}(p)\}\}, \quad (1)$$

where $\mathcal{S}^n$ is the Euclidean space of symmetric matrices, and $\mathrm{O}(p)$ is the orthogonal group. By abuse of notations, we use $[U]$ and $U$ interchangeably for the element of $\mathrm{Gr}(p,n)$. In many applications, measurements lie in the Grassmannian (Edelman et al., 1998; Huang et al., 2018; Nguyen et al., 2024; Wang et al., 2024a). Although the ONB and PP are diffeomorphic (Helmke & Moore, 2012), their effectiveness may vary depending on the specific tasks (Nguyen, 2022a).

*Remark* 2.1. This work utilizes Riemannian operators such as the Riemannian exponential and logarithmic maps. However, due to incompleteness and cut locus, these operators may not always be globally well-defined, such as the exponential map on the SPD PEM and BWM geometries, and the Grassmannian logarithmic map. Nevertheless, all constraints can be resolved numerically, as discussed in App. B. Therefore, without loss of generality, we assume these operators are well-defined.

## 3 REVISITING MLR AND FC LAYERS

### 3.1 EUCLIDEAN SPACES: FROM MLR TO THE FC LAYER

**Euclidean MLR.** Given $C$ classes, the Euclidean Multinomial Logistic Regression (MLR) computes the multinomial probability of each class $k \in \{1, \ldots, C\}$ for the input feature vector $x \in \mathbb{R}^n$:

$$p(y = k \mid x) \propto \exp\left(v_k(x)\right), \text{ with } v_k(x) = \langle a_k, x \rangle - b_k, b_k \in \mathbb{R}, a_k \in \mathbb{R}^n. \quad (2)$$

Lebanon & Lafferty (2004, Sec. 5) reformulated $v_k(x)$ by the margin distance to the hyperplane:

$$p(y = k \mid x) \propto \exp\left(\mathrm{sign}(\langle a_k, x - p_k \rangle) \|a_k\| d(x, H_{a_k, p_k})\right), \quad (3)$$

$$H_{a_k, p_k} = \{x \in \mathbb{R}^n : \langle a_k, x - p_k \rangle = 0\}, \quad (4)$$

where $\langle a_k, p_k \rangle = b_k$, and $H_{a_k, p_k}$ is a hyperplane.

**FC and convolutional layers.** The affine transformation in the FC layer, $y = Ax + b$, can be represented element-wise as $y_k = \langle a_k, x \rangle - b_k$, where $x, a_k \in \mathbb{R}^n$ and $b_k \in \mathbb{R}$. Additionally, the convolution is composed of FC transformations, as the transformation in each receptive field is essentially an FC transformation.

### 3.2 RIEMANNIAN MLR AND GYRO SPD & HYPERBOLIC FC LAYERS

According to Sec. 3.1, extending linear layers like FC and convolutional layers hinges on two key steps: 1. extending MLR or $v_k(\cdot)$ to the manifold; 2. obtaining $y_k$ from $v_k$ on the manifold. The first step has been well-studied, while the second one is only solved over specific geometries. We will first recap Riemannian MLR, and then discuss the existing FC layers on the hyperbolic and SPD manifolds.

**Riemannian MLR.** As shown by Chen et al. (2024c), Eqs. (3) and (4) can be naturally extended into the Riemannian manifold $\mathcal{N}$

$$p(y = k \mid X) \propto \exp\left(\mathrm{sign}(\langle A_k, \mathrm{Log}_{P_k}(X)\rangle_{P_k}) \|A_k\|_{P_k} d(X, \widetilde{H}_{A_k, P_k})\right), \quad (5)$$

$$\widetilde{H}_{A_k, P_k} = \{X \in \mathcal{N} : \langle \mathrm{Log}_{P_k}(X), A_k \rangle_{P_k} = 0\}, \quad (6)$$

where $X \in \mathcal{N}$ is the input manifold-valued feature, $P_k \in \mathcal{N}$ and $A_k \in T_{P_k}\mathcal{N}$ are parameters, $\langle \cdot, \cdot \rangle_{P_k}$ is the Riemannian metric at $P_k$, and $\mathrm{Log}_{P_k}$ is the Riemannian logarithm at $P_k$. Here, $d(X, \widetilde{H}_{A_k, P_k})$ is the margin distance to the hyperplane. Based on this reformulation, several works have extended the MLR into different geometries, such as Poincaré MLR on the hyperbolic space (Ganea et al., 2018, Thm. 5), gyro MLR on the SPD (Nguyen & Yang, 2023, Thms. 2.23-2.25) and Symmetric Positive Semi-Definite (SPSD) matrices (Nguyen et al., 2024, Thm. 3.11), and flat SPD MLR on the flat SPD geometries (Chen et al., 2024a, Thm. 3.8). However, all the above solutions rely on specific properties. To address this limitation, Chen et al. (2024c, Thms. 3.2-3.3) recently offered general expressions for the margin distance and the Riemannian MLR over general geometries solely based on Riemannian properties. We recap their results in the following.

**Theorem 3.1** (Riemannian Margin Distance & MLR (Chen et al., 2024c)). *Given $X \in \mathcal{N}$, the Riemannian margin distance and MLR over the Riemannian manifold $\{\mathcal{N}, g^{\mathcal{N}}\}$ is*

$$d(X, \widetilde{H}_{A_k, P_k}) = \frac{|\langle \mathrm{Log}_{P_k}(X), A_k \rangle_P|}{\|A_k\|_{P_k}}, \tag{7}$$

$$p(y = k \mid X \in \mathcal{N}) \propto \exp\left(v_k(X; A_k, P_k)\right), \tag{8}$$

*where $P_k \in \mathcal{N}$, $A_k \in T_{P_k}\mathcal{N}$, and $v_k(X; A_k, P_k) = \langle A_k, \mathrm{Log}_{P_k}(X) \rangle_{P_k}$*

**SPD and hyperbolic FC layers.** The FC layer has been extended to both the hyperbolic and SPD manifolds. Shimizu et al. (2020) proposed the Poincaré FC layer, which is based on the hyperbolic MLR and reformulation of the FC layer using hyperbolic geometry. Besides, Nguyen et al. (2024) introduced three gyro SPD FC layers, based on the gyro SPD MLRs and the reformulation of the FC layer via gyro structures. However, not all geometries admit gyro structures, such as BWM on the SPD manifold. Moreover, even for manifolds that admit gyro structures, the formulation of the FC layers needs to be addressed on a case-by-case basis. In contrast, this paper proposes a framework that can be readily applied across different geometries.

# 4 RIEMANNIAN FULLY CONNECTED AND CONVOLUTIONAL LAYERS

Since convolution can be derived from the FC layer, we first extend the FC layer to general manifolds, and then introduce the Riemannian convolution. Lastly, we address the manipulation of parameters.

## 4.1 RIEMANNIAN FULLY CONNECTED LAYERS

Shimizu et al. (2020, Sec. 3.2) interpreted the Euclidean FC layer as an operation that transforms the input $x$ via $v_k(x)$, treating the output $y_k$ as the signed distance from the hyperplane passing through the origin and orthogonal to the $k$-th axis of the output space $\mathbb{R}^m$. We now extend this idea into general manifolds.

The Riemannian $v_k(\cdot)$ can be obtained by Eq. (8), while the sign distance to a Riemannian hyperplane can also be derived from Eq. (7). The rest is to generalize the hyperplane containing the origin and orthogonal to the $k$-th axis. In the Euclidean space $\mathbb{R}^m$, this kind of hyperplane is formulated as

$$H_{e_k, 0} = \{x \in \mathbb{R}^m : \langle e_k, x \rangle = 0\}, \forall k \in \{1, \cdots, m\}, \tag{9}$$

where $e_k$ is a vector with its $k$-th element equal to 1 and all other elements equal to 0. The set $\{e_k\}_{k=1}^m$ is more generally characterized as the orthonormal bases over $\mathbb{R}^m$. Further considering $\mathrm{Log}_0(x) = x$ and $T_0\mathbb{R}^m \cong \mathbb{R}^m$, the counterparts of this kind of hyperplane on an $m$-dimensional Riemannian manifold $\mathcal{M}$ can be defined as

$$\widetilde{H}_{B_k, E} = \{S \in \mathcal{M} : \langle \mathrm{Log}_E S, B_k \rangle_E = 0\}, \forall k \in \{1, \cdots, m\}, \tag{10}$$

where $E \in \mathcal{M}$ is the origin, and $\{B_k\}_{k=1}^m$ are orthonormal bases over $\{T_E\mathcal{M}, g_E\}$. Essentially, Eq. (10) characterizes the hyperplane containing the origin and orthogonal to the geodesic starting from $E$ with initial velocity $B_k$. Therefore, it naturally generalizes Eq. (9) into manifolds. With all the above discussion, we define the Riemannian FC layer in the following.

**Definition 4.1** (Riemannian FC layers). Given $n$-dimensional manifold $\mathcal{N}$ and $m$-dimensional manifold $\mathcal{M}$, the Riemannian FC layer $\mathcal{F} : \mathcal{N} \to \mathcal{M}$ returns the output $Y \in \mathcal{M}$ by solving the following $m$ equations w.r.t. the input $X \in \mathcal{N}$:

$$s_k \, \mathrm{d}^{\mathcal{M}}(Y, H^{\mathcal{M}}_{B_k, E^{\mathcal{M}}}) = v_k^{\mathcal{N}}(X; A_k, P_k), 1 \le k \le m, \tag{11}$$

where $E^{\mathcal{M}} \in \mathcal{M}$ is the origin, $\{B_k\}_{k=1}^m$ is an orthonormal basis over $T_{E^{\mathcal{M}}}\mathcal{M}$. Here, $v_k^{\mathcal{N}}$ over $\mathcal{N}$ and $\mathrm{d}^{\mathcal{M}}$ over $\mathcal{M}$ are defined by Eq. (8) and Eq. (7), respectively. The sign for the margin distance is $s_k = \mathrm{sign}\left(\langle \mathrm{Log}_E^{\mathcal{M}}(Y), O_k \rangle_E^{\mathcal{M}}\right)$. Here, each $P_k \in \mathcal{N}$ and $A_k \in T_{P_k}\mathcal{N}$ are parameters.

The above definition has a general solution, which is presented in the following.

**Theorem 4.2** (Riemannian FC Layers). [↓] *Given an $n$-dimensional Riemannian manifold $\{\mathcal{N}, g^{\mathcal{N}}\}$, an $m$-dimensional Riemannian manifold $\{\mathcal{M}, g^{\mathcal{M}}\}$, and orthonormal bases $\{B_i\}_{i=1}^m$ over $T_E\mathcal{M}$ with $E \in \mathcal{M}$ as the origin, the Riemannian FC layer $\mathcal{F}(\cdot) : \mathcal{N} \to \mathcal{M}$ is*

$$Y = \mathrm{Exp}_E^{\mathcal{M}}\left(\sum_{i=1}^m v_i(X) B_i\right) = \mathrm{Exp}_E^{\mathcal{M}}\left(\sum_{i=1}^m \left(\langle \mathrm{Log}_{P_i}^{\mathcal{N}}(X), A_i \rangle_{P_i}^{\mathcal{N}} B_i\right)\right), \tag{12}$$

where $X \in \mathcal{N}$ is the input feature, and $P_i \in \mathcal{N}$ and $A_i \in T_{P_i}\mathcal{N}$ are the parameters. Here, $\mathrm{Exp}_E^{\mathcal{M}}$ is the Riemannian exponentiation over $\mathcal{M}$, while $\mathrm{Log}_{P_i}^{\mathcal{N}}$ and $\langle \cdot, \cdot \rangle_{P_i}^{\mathcal{N}}$ are Riemannian logarithm and metric over $\mathcal{N}$. We denote the above equation as

$$Y = \mathcal{F}(X; \mathbf{A}, \mathbf{P}), \tag{13}$$

with $\mathbf{P} = \{P_i \in \mathcal{N}\}_{i=1}^m$ and $\mathbf{A} = \{A_i \in T_{P_i}\mathcal{N}\}_{i=1}^m$ as the FC parameters.

*Remark* 4.3. When the inner product $g_E$ on $T_E\mathcal{M}$ is not the standard inner product, the familiar $\{e_i\}_{i=1}^m$ might be orthonormal. Please refer to App. C for details on identifying an orthogonal basis.

Our Riemannian FC layer is a natural generalization of the Euclidean FC layer.

**Proposition 4.4.** [↓] *When $\mathcal{M} = \mathbb{R}^m$ and $\mathcal{N} = \mathbb{R}^n$ are the standard Euclidean spaces, the Riemannian FC layer in Eq. (12) becomes the Euclidean FC layer.*

As isometric Riemannian metrics are frequently encountered across various geometries (Thanwerdas & Pennec, 2022; Chen et al., 2024d;c; Bendokat et al., 2024), we also present a theorem in App. D to facilitate constructing Riemannian FC layers under isometries.

### 4.2 RIEMANNIAN CONVOLUTIONAL LAYERS

**Disentangling the Euclidean convolution.** As mentioned in Sec. 3.1, the convolution can be viewed as the product of the FC layer on each receptive field. Let us focus on a single receptive field. Given a $c$-channel vector in a receptive field $\mathbf{x} = \mathrm{concat}(x_1, \cdots, x_c) \in (\mathbb{R}^n)^c$ with $x_i \in \mathbb{R}^n$ as the feature vector in the $i$-th channel, the Euclidean convolution within this receptive field can be expressed as

$$\mathrm{Conv}(\mathbf{x}) = \mathrm{concat}\left(f^1(\mathbf{x}), \cdots, f^k(\mathbf{x})\right), \text{ with } f^i(\cdot) : (\mathbb{R}^n)^c \to \mathbb{R}^m, \forall i = 1, \cdots k. \tag{14}$$

where $f^i$ is the affine (FC) transformation parameterized by the $i$-th convolutional kernel.

**Riemannian convolution.** Similarly, the Riemannian convolution is defined as the Riemannian FC layer within each receptive field. Given a $c$-channel manifold-valued input $\mathbf{X} = \{X_1, \cdots, X_c\} \in \mathcal{M}^c$ for a receptive field, the Riemannian convolution $\mathrm{Conv}(\cdot) : \mathcal{M}^c \to \mathcal{N}^k$ within this receptive field is

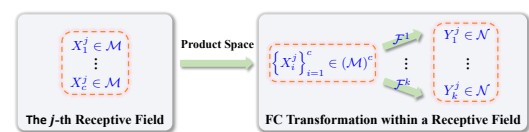

Figure 1: Conceptual illustration of Riemannian convolution within a reception field.

$$\mathrm{Conv}(\mathbf{X}) = \{\mathcal{F}^1(\mathbf{X}), \cdots, \mathcal{F}^k(\mathbf{X})\}, \text{ with } \mathcal{F}^i(\cdot) : \mathcal{M}^c \to \mathcal{N}, \forall i = 1, \cdots k. \tag{15}$$

The above process is illustrated in Fig. 1.

*Remark* 4.5. Chakraborty et al. (2020) proposed a convolution operation for manifolds. However, their convolution is based on the weighted Fréchet mean. Therefore, it is unable to alter the manifold dimension, such as performing dimensionality reduction. In contrast, our framework provides greater flexibility, as it allows for modifications in both the channel and manifold dimensions. Furthermore, while Nguyen et al. (2024) introduced gyro SPD FC and convolutional layers via gyro structures induced by LEM, AIM, and LCM, these gyro SPD transformation layers are special cases within our framework, which will be discussed in Sec. 5.

### 4.3 PARAMETERS MANIPULATION

Lastly, let us discuss the parameters. As convolution takes the FC layer as the prototype, we focus on the FC parameters $\mathbf{A}$ and $\mathbf{P}$. Since $P_i$ varies during the training, $A_i \in T_{P_i}\mathcal{N}$ cannot be directly updated by the Euclidean optimizer. As shown by Chen et al. (2024c, Eqs. (12)-(13)), $A_i \in T_{P_i}\mathcal{N}$ can be determined from the tangent space at the origin $E^{\mathcal{N}} \in \mathcal{N}$

$$f(\cdot) : T_{E^{\mathcal{N}}}\mathcal{N} \to T_{P_i}\mathcal{N}, \text{ with } f(Z_i) = A_i, Z_i \in T_{E^{\mathcal{N}}}\mathcal{N} \cong \mathbb{R}^n, \tag{16}$$

where $f$ could be parallel transport along the geodesic or the differential map of Lie group translations[1]. Besides, as shown by Shimizu et al. (2020, Sec. 3.1), $P_k$ might be overly parameterized, as there are countless many $p_k$ in Eq. (4) satisfying $\langle a_k, p_k \rangle = b_k$. Therefore, following Shimizu et al. (2020), each $P_i$ in the Riemannian FC layer is parameterized as $\mathrm{Exp}_{E^{\mathcal{M}}}^{\mathcal{M}}(\gamma_i[Z_i])$, where $\gamma_i \in \mathbb{R}$ and $[Z_i]$ is the unit vector of $Z_i$. In this way, all the FC parameters can be directly optimized by the well-established Euclidean optimizer. Note that modeling manifold-valued parameters by the exponential map is generally called trivialization, which has been well-studied by Lezcano Casado (2019, Sec. 4.1).

---

[1] As mentioned by Chen et al. (2024c, Sec. 3.2), $f$ is flexible and could be other operations, such as vector transport and the differential of gyro group translation.

## 5 SPD FULLY CONNECTED AND CONVOLUTIONAL LAYERS

This section instantiates our theoretical FC layer in Thm. 4.2 over the SPD manifold, *i.e.,* $\mathcal{F}(\cdot)$ : $\mathcal{S}_{++}^n \to \mathcal{S}_{++}^m$. The SPD convolution can then be derived by the product of FC layers. We focus on five popular Riemannian metrics, *i.e.,* LEM, AIM, PEM, LCM, and BWM. As the identity matrix is the neutral element under various Lie and gyro group structures (Arsigny et al., 2005; Lin, 2019; Thanwerdas & Pennec, 2022; Nguyen, 2022a), we define the origin on the SPD manifold as the identity matrix. The following theorem presents our results.

**Theorem 5.1** (SPD FC Layers). [↓] *Given an SPD matrix $S \in \mathcal{S}_{++}^n$, the SPD FC layers $\mathcal{F}(\cdot)$ : $\mathcal{S}_{++}^n \to \mathcal{S}_{++}^m$ under different Riemannian metrics are*

$$
\textit{LEM}: Y = \exp\left(V^{\mathrm{LE}}\right), V_{ij}^{\mathrm{LE}} = \begin{cases} \frac{1}{\sqrt{\alpha}} v_{ii}^{\mathrm{LE}}(S) + \mu \sum_{k=1}^m v_{kk}^{\mathrm{LE}}(S), & \textit{if } i = j \\ \frac{1}{\sqrt{2\alpha}} v_{ij}^{\mathrm{LE}}(S), & \textit{if } i > j \\ V_{ji}^{\mathrm{LE}}, & \textit{otherwise} \end{cases} \tag{17}
$$

$$
\textit{AIM}: Y = \exp\left(V^{\mathrm{AI}}\right), V_{ij}^{\mathrm{AI}} = \begin{cases} \frac{1}{\sqrt{\alpha}} v_{ii}^{\mathrm{AI}}(S) + \mu \sum_{k=1}^m v_{kk}^{\mathrm{AI}}(S), & \textit{if } i = j \\ \frac{1}{\sqrt{2\alpha}} v_{ij}^{\mathrm{AI}}(S), & \textit{if } i > j \\ V_{ji}^{\mathrm{AI}}, & \textit{otherwise} \end{cases} \tag{18}
$$

$$
\textit{PEM}: Y = \left(I + V^{\mathrm{PE}}\right)^{\frac{1}{\theta}}, V_{ij}^{\mathrm{PE}} = \begin{cases} \frac{1}{\sqrt{\alpha}} v_{ii}^{\mathrm{PE}}(S) + \mu \sum_{k=1}^m v_{kk}^{\mathrm{PE}}(S), & \textit{if } i = j \\ \frac{1}{\sqrt{2\alpha}} v_{ij}^{\mathrm{PE}}(S), & \textit{if } i > j \\ V_{ji}^{\mathrm{PE}}, & \textit{otherwise} \end{cases} \tag{19}
$$

$$
\textit{LCM}: Y = V^{\mathrm{LC}}(V^{\mathrm{LC}})^\top, V_{ij}^{\mathrm{LC}} = \begin{cases} \exp\left(v_{ii}^{\mathrm{LC}}(S)\right), & \textit{if } i = j \\ v_{ij}^{\mathrm{LC}}(S), & \textit{if } i > j \\ 0, & \textit{otherwise} \end{cases} \tag{20}
$$

$$
\textit{BWM}: Y = \left(I + \frac{1}{2} V^{\mathrm{BW}}\right)^2, V_{ij}^{\mathrm{BW}} = \begin{cases} v_{ii}^{\mathrm{BW}}(S), & \textit{if } i = j \\ \frac{1}{\sqrt{2}} v_{ij}^{\mathrm{BW}}(S), & \textit{if } i > j \\ V_{ji}^{\mathrm{BW}}, & \textit{otherwise} \end{cases} \tag{21}
$$

*Here, $v_{ij}(S)$ under different metrics are given as*

$$
\textit{LEM}: \langle \log(S) - \log(P_{ij}), Z_{ij} \rangle^{(\alpha,\beta)}, \tag{22}
$$

$$
\textit{AIM}: \left\langle \log(P_{ij}^{-\frac{1}{2}} S P_{ij}^{-\frac{1}{2}}), Z_{ij} \right\rangle^{(\alpha,\beta)}, \tag{23}
$$

$$
\textit{PEM}: \left\langle S^\theta - P_{ij}^\theta, Z_{ij} \right\rangle^{(\alpha,\beta)}, \tag{24}
$$

$$
\textit{LCM}: \left\langle \lfloor K \rfloor - \lfloor L_{ij} \rfloor + \mathrm{Dlog}(\mathbb{K}\mathbb{L}_{ij}^{-1}), \lfloor Z_{ij} \rfloor + \frac{1}{2} \mathbb{Z}_{ij} \right\rangle, \tag{25}
$$

$$
\textit{BWM}: \left\langle (P_{ij}S)^{\frac{1}{2}} + (SP_{ij})^{\frac{1}{2}} - 2P_{ij}, \mathcal{L}_{P_{ij}}(L_{ij}Z_{ij}L_{ij}^\top) \right\rangle, \tag{26}
$$

*The above notations are defined in the following.*

- *For $i, j = 1, \cdots, m$ and $i \geq j$, $Z_{ij} \in T_I \mathcal{S}_{++}^n \cong \mathcal{S}^n$ and $P_{ij} \in \mathcal{S}_{++}^n$ are the parameters.*

- $\log(\cdot)$ *is the matrix logarithm.* $\mathrm{Dlog}(\cdot)$ *is the diagonal element-wise logarithm.* $\lfloor \cdot \rfloor$ *is the strictly lower part of a square matrix.* $\mathrm{Chol}(\cdot)$ *is the Cholesky decomposition.* $\mathbb{V}$ *is a diagonal matrix with diagonal elements of the square matrix $V$. $\mathcal{L}_P(V)$ is the solution to the matrix linear system $\mathcal{L}_P[V]P + P\mathcal{L}_P[V] = V$, known as the Lyapunov operator.*

- $\langle \cdot, \cdot \rangle^{(\alpha,\beta)}$ *is the $\mathrm{O}(n)$-invariant inner product defined in Eq. (34) and $\langle \cdot, \cdot \rangle$ is the Frobenius matrix inner product.*

- $\mu = \frac{1}{n}\left(\frac{1}{\sqrt{\alpha+n\beta}} - \frac{1}{\sqrt{\alpha}}\right)$, $K = \mathrm{Chol}(S)$ *and $L_{ij} = \mathrm{Chol}(P_{ij})$.*

- *Due to the incompleteness of PEM and BWM, there are constraints for $V^{\mathrm{PE}}$ and $V^{\mathrm{BW}}$: $I + \theta V^{\mathrm{PE}} \in \mathcal{S}_{++}^m$ and $I + \frac{1}{2}V^{\mathrm{BW}} \in \mathcal{S}_{++}^n$. Both constraints can be solved numerically, such as the regularization of eigenvalues, as detailed in Rmk. F.2.*

The affine transformation $y = Ax + b$ in the Euclidean FC layer incorporates the linear map $y = Ax$, the most natural map between linear spaces. As shown by Arsigny et al. (2005, Sec. 4.4) and Chen et al. (2024d, Thm. 1), the SPD manifold admits two vector space structures w.r.t. LEM and LCM. Similar to the Euclidean FC layer, our SPD FC layer also incorporates linear homomorphisms over these vector structures. Denoting the element addition and scalar product as $\oplus^{\text{LE}}$ ($\oplus^{\text{LC}}$) and $\odot^{\text{LE}}$ ($\odot^{\text{LC}}$), which is detailed in App. K.4, we have the following result.

**Proposition 5.2.** [↓] *The SPD FC layers under LEM and LCM incorporate the linear homomorphisms over the vector spaces $\{\mathcal{S}_{++}^n, \oplus^{\text{LE}}, \odot^{\text{LE}}\}$ and $\{\mathcal{S}_{++}^n, \oplus^{\text{LC}}, \odot^{\text{LC}}\}$, respectively.*

**Difference with gyro SPD FC layers.** We acknowledge that Nguyen et al. (2024, Props. 3.4-3.6) introduced gyro SPD FC layers under the AIM, LEM, and LCM gyro structures. However, gyro structures are not universally applicable across all Riemannian geometries. For example, BWM is agnostic to gyro structures (Chen et al., 2024c, Rmk. 4.3). In contrast, our framework relies solely on Riemannian structures, allowing it to handle a broader range of geometries. For the specific case of SPD FC layers, our Thm. 5.1 incorporates all the gyro SPD FC layers as special cases, which are detailed in App. E. Tab. 1 summarizes the comparison.

Table 1: Comparison with the Gyro SPD FC layers.

| SPD FC Layers | Geometries | Requirements | Incorporated by Ours |
|---|---|---|---|
| Gyro SPD FC layer | AIM, LEM & LCM on $\mathcal{S}_{++}^n$ | Gyro structures | ✓(App. E) |
| Ours | Riemannian manifolds | Riemannian geometries | N/A |

**Parameter manipulation and simplification.** Following the discussion in Sec. 4.3, we model each $P_{ij} \in \mathcal{S}_{++}^n$ by Riemannian exponential at the identity matrix, *i.e.*,$\text{Exp}_I(\gamma_{ij}[Z_{ij}])$. Under this trivialization, the SPD FC layer under LEM, AIM, LCM, and PEM can be further simplified. Please refer to App. F for more details.

**SPD convolution.** As discussed in Sec. 4.2, the SPD convolution is defined as the product of the SPD FC layers, *i.e.*,$\text{Conv}(\cdot) : (\mathcal{S}_{++}^n)^c \to (\mathcal{S}_{++}^m)^k$

$$\text{Conv}(\cdot) = \{\mathcal{F}^1(\cdot), \cdots, \mathcal{F}^k(\cdot)\}, \text{ with } \mathcal{F}^i(\cdot) : (\mathcal{S}_{++}^n)^c \to \mathcal{S}_{++}^m, \forall i = 1, \cdots k, \tag{27}$$

with $\mathcal{F}^i$ as the SPD FC layer under a given metric.

## 6 GRASSMANNIAN FULLY CONNECTED AND CONVOLUTIONAL LAYERS

We first discuss the FC layers over the ONB Grassmannian in Sec. 6.1, followed by the cases under the PP Grassmannian in Sec. 6.2. As the product of the FC layers, the convolutional layer can be derived as before. Finally, Sec. 6.3 compares our Grassmannian convolution (GrConv) with existing popular Grassmannian transformation layers, concluding that our GrConv enables more flexibility in both dimensionality and perspective.

### 6.1 ONB GRASSMANNIAN TRANSFORMATION LAYERS

Under the ONB perspective, each Grassmannian point can be represented as a column-wise orthogonal matrix. We denote $I_{p,n} = \begin{pmatrix} I_p \\ \mathbf{0} \end{pmatrix} \in \mathbb{R}^{n \times p}$, with $I_p$ as the $p \times p$ identity matrix. As $I_{p,n}$ is the identity element of the gyro group on the ONB Grassmannian $\text{Gr}(p, n)$ (Nguyen & Yang, 2023), we define it as the origin. As discussed in Sec. 4.3, we model the FC parameters by parallel transport and Riemannian exponential map at $I_{p,n}$. Under this trivialization, the manifestation of Thm. 4.2 on the ONB Grassmannian can be further simplified.

**Theorem 6.1** (ONB Grassmannian FC Layers). [↓] *Given an ONB Grassmannian feature $U \in \text{Gr}(p, n)$, the ONB Grassmannian FC layer $\mathcal{F}(\cdot) : \text{Gr}(p, n) \to \text{Gr}(q, m)$ is*

$$Y = \begin{pmatrix} R\cos(\Sigma)R^\top \\ O\sin(\Sigma)R^\top \end{pmatrix} \text{ with } B^{\text{ONB}} \stackrel{SVD}{:=} O\Sigma R^\top \in \mathbb{R}^{(m-q) \times q}. \tag{28}$$

*Here, each $(i, j)$ element of $B^{\text{ONB}} \in \mathbb{R}^{(m-q) \times q}$ is defined as $\left\langle \text{Log}_{P_{ij}}^{\text{ONB}}(U), T_{ij}B_{Z_{ij}} \right\rangle$, with*

$$T_{ij} = \begin{pmatrix} -R_{ij}\sin(\Sigma_{ij})O_{ij}^\top \\ O_{ij}\cos(\Sigma_{ij})O_{ij}^\top + I_{n-p} - O_{ij}O_{ij}^\top \end{pmatrix} \tag{29}$$

*where $\gamma_{ij}[B_{Z_{ij}}] \stackrel{SVD}{:=} O_{ij}\Sigma_{ij}R_{ij}^{\top}$ is the SVD decomposition, and $B_{Z_{ij}} \in \mathbb{R}^{(n-p)\times p}$ and $\gamma_{ij} \in \mathbb{R}$ are the FC parameters.*

## 6.2 PP GRASSMANNIAN TRANSFORMATION LAYERS

Under the PP perspective, each Grassmannian point can be represented as a symmetric matrix. We define the PP origin as $\widetilde{I}_{p,n} = \begin{pmatrix} I_p & \mathbf{0} \\ \mathbf{0} & \mathbf{0} \end{pmatrix} \in \mathbb{R}^{n\times n}$, as it is the identity element of the gyro group on the PP Grassmannian $\widetilde{\mathrm{Gr}}(p,n)$ (Nguyen, 2022a). Similarly, we model the FC parameters by parallel transport and Riemannian exponential map at $\widetilde{I}_{p,n}$. Under this trivialization, Thm. 4.2 on the PP Grassmannian can be further simplified. Besides, the Riemannian logarithm under the PP Grassmannian can be calculated by the ONB logarithm to support the auto-differentiation (Nguyen et al., 2024, Prop. 3.12). For more details, please refer to the proof of the following theorem.

**Theorem 6.2** (PP Grassmannian FC Layers). [↓] *Given a PP Grassmannian feature $X \in \widetilde{\mathrm{Gr}}(p,n)$, the PP Grassmannian FC layer $\mathcal{F}(\cdot) : \widetilde{\mathrm{Gr}}(p,n) \to \widetilde{\mathrm{Gr}}(q,m)$ is*

$$Y = \widetilde{U}\widetilde{U}^{\top} \text{ with } \widetilde{U} = \left( \exp\left( \begin{pmatrix} 0 & -(B^{\mathrm{PP}})^T \\ B^{\mathrm{PP}} & 0 \end{pmatrix} \right) \right)_{1:q}, \tag{30}$$

*where $(\cdot)_{1:q}$ returns the first-$q$ columns of the input square matrix. Here, each $(i,j)$ element of $B^{\mathrm{PP}} \in \mathbb{R}^{(m-q)\times q}$ is defined as $\frac{1}{2}\left\langle \pi_{*,\pi(P)}\left( \mathrm{Log}^{\mathrm{ONB}}_{(O_{ij})_{1:p}}(\pi^{-1}(X)) \right), O_{ij}Z_{ij}O_{ij}^{\top} \right\rangle$, with*

$$O_{ij} = \exp\left( \begin{pmatrix} 0 & -(\gamma_{ij}[B_{Z_{ij}}])^T \\ \gamma_{ij}[B_{Z_{ij}}] & 0 \end{pmatrix} \right), \tag{31}$$

*where $\pi(U) = UU^{\top}$, and $\pi_{*,U}(V) = UV^{\top} + VU^{\top}$ is the differential map for all $U \in \mathrm{Gr}(p,n)$ and $V \in T_U\mathrm{Gr}(p,n)$. The FC parameters are $B_{Z_{ij}} \in \mathbb{R}^{(n-p)\times p}$ and $\gamma_{ij} \in \mathbb{R}$ for $i = 1, \cdots, m-q$ and $j = 1, \cdots, q$.*

## 6.3 COMPARISON WITH THE EXISTING GRASSMANNIAN TRANSFORMATION LAYERS

Table 2: Comparison of our GrConv against the existing transformation layers. Unlike existing transformation layers, our GrConv can transform subspace dimension $p$, the ambient dimension $n$, and the channel dimension $c$ across both two perspectives, providing more flexibility.

| Methods | Perspective | Flexible dimensions | | |
|---|---|---|---|---|
| | | Subspace $p$ | Ambient $n$ | Channel |
| FRMap + ReOrth (Huang et al., 2018, Eqs. (2-4)) | ONB | ✗ | ✓ | ✗ |
| PP Scaling (Nguyen, 2022a, Sec. 4.2.2) | PP | ✗ | ✗ | ✗ |
| ONB Scaling (Nguyen & Yang, 2023, Sec. 3.2) | ONB | ✗ | ✗ | ✗ |
| GrTrans (Nguyen & Yang, 2023, Sec. 2.3.2) | ONB + PP | ✗ | ✗ | ✗ |
| GrConv | ONB + PP | ✓ | ✓ | ✓ |

As discussed in Sec. 4.2, The product of the FC layers defines the ONB and PP Grassmannian convolution. For example, the ONB Grassmannian, $\mathrm{Conv}(\cdot) : (\mathrm{Gr}(p,n))^c \to (\mathrm{Gr}(q,m))^k$, is defined as

$$\mathrm{Conv}(\cdot) = \{\mathcal{F}^1(\cdot), \cdots, \mathcal{F}^k(\cdot)\}, \text{ with } \mathcal{F}^i(\cdot) : (\mathrm{Gr}(p,n))^c \to \mathrm{Gr}(q,m), \forall i = 1, \cdots k, \tag{32}$$

with $\mathcal{F}^i$ as the ONB Grassmannian FC layer. The following begins with a brief recap of several popular Grassmannian transformation layers, followed by a comparison with our proposed Grassmannian Convolution (GrConv).

Huang et al. (2018) proposed FRMap + ReOrth layers to perform the transformation over the ONB Grassmannian via left matrix product (FRMap) and QR decomposition (ReOrth). Nguyen (2022a) proposed the matrix scaling for the PP Grassmannian by the tangent space at the identity. Nguyen & Yang (2023) extended the matrix scaling into the ONB Grassmannian. Besides, Nguyen & Yang (2023) used the gyro group left translation (GrTrans) as the transformation. These layers are briefly recapped in App. H. However, all the previous layers lack flexibility regarding dimensions and

perspectives. Given a $c$-channel Grassmannian $\mathrm{Gr}(p, n)$ (or $\widetilde{\mathrm{Gr}}(p, n)$) input, the existing layers can modify only specific aspects of the three dimensions $(c, p, n)$ or operate on a limited perspective. In contrast, our GrConv layer can adjust all dimensions across both perspectives, enabling more flexibility. Tab. 2 compares our GrConv with other Grassmannian transformation layers, highlighting the advantages of our approach.

# 7 MANIFOLD EMBEDDING AND RIEMANNIAN FULLY CONNECTED LAYER

Embedding into non-Euclidean manifolds often yields superior results compared to standard Euclidean spaces (Chami et al., 2019; López et al., 2021; Zhao et al., 2023; Nguyen et al., 2024). A common approach for embedding Euclidean features into manifolds involves mapping the Euclidean vector to the tangent space at the origin via a linear layer, followed by applying the exponential map at the origin. This method has been widely adopted in various embeddings, including hyperbolic (Chami et al., 2019; Fu et al., 2024), SPD (Zhao et al., 2023), and Grassmannian spaces (Nguyen et al., 2024, Sec. 3.4.2). While this process appears extrinsic due to its dependence on the tangent space, our framework offers a novel intrinsic interpretation. The following proposition shows that this operation is, in essence, a Riemannian FC layer between the Euclidean space and the target manifold.

**Proposition 7.1** (Manifold Embeddings & Riemannian FC layers). [↓] *The Riemannian FC layer from a standard Euclidean space $\mathbb{R}^n$ to an $m$-dimensional target manifold $\mathcal{M}$, namely $\mathcal{F}(\cdot) : \mathbb{R}^n \to \mathcal{M}$, takes the following form*

$$\mathcal{F}(x) = \mathrm{Exp}_E(Ax + b),\qquad(33)$$

*where $A \in \mathbb{R}^{n \times m}$ and $b \in \mathbb{R}^m$ are the transformation matrix and biasing vector, respectively.*

# 8 EXPERIMENTS

We use the proposed Riemannian convolutional layers to construct Riemannian Convolutional Neural Networks (RCNNs) on the SPD and Grassmannian manifolds, referred to as SPDConvNets and GrConvNets, respectively. Following previous work (Huang et al., 2017; Brooks et al., 2019; Wang et al., 2024a), we evaluate our method on radar signal classification and human action recognition tasks. More details of the datasets and implementation are exposed in App. I.

Table 3: Comparison of the SPDConvNets under different metrics against other SPD networks on all three datasets. The best three results are highlighted with **red**, **blue**, and **cyan**.

| Methods | Radar | | HDM05 | | FPHA | |
|---|---|---|---|---|---|---|
| | Mean±STD | Max | Mean±STD | Max | Mean±STD | Max |
| SPDNet | $93.25 \pm 1.10$ | 94.4 | $64.57 \pm 0.61$ | 65.14 | $85.59 \pm 0.72$ | 86 |
| SPDNetBN | $94.85 \pm 0.99$ | 96.13 | $71.28 \pm 0.79$ | 72.7 | $89.33 \pm 0.49$ | 90.17 |
| RResNet-AIM | $95.71 \pm 0.37$ | 96.4 | $64.95 \pm 0.82$ | 66.19 | $86.63 \pm 0.55$ | 87.33 |
| RResNet-LEM | $95.89 \pm 0.86$ | 97.07 | $70.12 \pm 2.45$ | 71.92 | $85.07 \pm 0.99$ | 86.17 |
| SPDNetLieBN-AIM | $95.47 \pm 0.90$ | 96.27 | $71.83 \pm 0.69$ | 72.51 | $90.39 \pm 0.66$ | **92.17** |
| SPDNetLieBN-LCM | $94.80 \pm 0.71$ | 95.73 | $71.78 \pm 0.44$ | 72.61 | $86.33 \pm 0.43$ | 87 |
| SPDNetMLR | $95.64 \pm 0.83$ | 97.33 | $65.90 \pm 0.93$ | 66.98 | $85.67 \pm 0.69$ | 86.33 |
| SPDConvNet-LEM | $\mathbf{98.27 \pm 0.48}$ | **98.93** | $81.16 \pm 0.93$ | 82.44 | $91.83 \pm 0.41$ | 92.5 |
| SPDConvNet-AIM | $97.63 \pm 0.50$ | **98.4** | $80.12 \pm 0.78$ | 81.55 | $91.57 \pm 0.40$ | 92.17 |
| SPDConvNet-PEM | $98.43 \pm 0.44$ | 99.07 | $78.77 \pm 0.45$ | 79.19 | $90.33 \pm 0.37$ | 90.67 |
| SPDConvNet-LCM | $97.65 \pm 0.75$ | **98.93** | $75.42 \pm 0.95$ | 76.74 | $91.33 \pm 0.24$ | 91.67 |
| SPDConvNet-BWM | $96.40 \pm 0.91$ | 97.87 | $74.34 \pm 0.86$ | 75.85 | $90.03 \pm 0.55$ | 90.83 |

## 8.1 EXPERIMENTS ON SPD GEOMETRIES

**Datasets.** Following previous SPD methods (Huang et al., 2017; Brooks et al., 2019; Chen et al., 2024b), we use the Radar dataset (Brooks et al., 2019) for radar classification, and the HDM05 (Müller et al., 2007) and FPHA (Garcia-Hernando et al., 2018) datasets for human action recognition. In line with Wang et al. (2024a); Nguyen et al. (2024), we model each input feature as a multi-channel SPD tensor of covariance matrices, shaped as $[c, n, n]$.

**SPDConvNets.** We construct SPDConvNets based on convolutional layers induced by five Riemannian metrics, *i.e.*, LEM, AIM, PEM, LCM, and BWM. We employ a single convolutional layer,

followed by an SPD MLR (Chen et al., 2024c). We denote SPDConvNet-[Metric] as the SPDConvNet using convolution under the specified metric. For SPDConvNet-LEM, -PEM, and -LCM, the MLR is based on the same metric as the convolution, *i.e.,*, LEM, PEM, and LCM, respectively. Since the MLR for AIM and BWM is less efficient (Chen et al., 2024c), we apply LEM MLR for SPDConvNet-AIM and -BWM to facilitate training. Besides, we trivialize the SPD parameter in the MLR as Sec. 4.3, which are detailed in App. G. Consequently, all parameters in the SPDConvNets can be directly optimized using a Euclidean optimizer.

**Results.** We compare our SPDConvNets with various SPD baseline networks, including SPDNet (Huang et al., 2017), SPDNetBN (Brooks et al., 2019), LieBN (Chen et al., 2024b), RResNet (Katsman et al., 2024), and MLR (Chen et al., 2024c). The 5-fold average and maximum results are shown in Tab. 3. For RResNet, due to significant fluctuations in its training dynamics on the radar dataset, the test performance over the last several epochs varies by up to 20%. Therefore, we select the maximum accuracy from the last 10 epochs as its final scoring metric. Our findings are as follows. Firstly, our SPDConvNets consistently outperform other SPD-based models regarding both average and maximum accuracy. Specifically, our SPDConvNets surpass the classic SPDNet by up to **5.02%, 16.59%, and 6.24%** on the Radar, HDM05, and FPHA datasets, respectively. Notably, the best performance of our SPDConvNets on the Radar dataset even reaches 99.07%. These results demonstrate the effectiveness of our framework. Additionally, the variation in optimal metrics across datasets highlights the flexibility of our methods.

## 8.2 EXPERIMENTS ON GRASSMANNIAN GEOMETRIES

We compare our Grassmannian Convolutional (GrConv) layer against previous transformation layers, such as FRMap + ReOrth, GrTrans, and scaling under the GrNet backbone. In our experiments, we replace the vanilla FRMap + ReOrth in the GrNet backbone with GrTrans, ONB scaling, and our ONB & PP convolutional layers, respectively. Each model includes one transformation layer followed by a classification layer. The corresponding models are denoted as GyroGr, GyroGr-Scaling, GrConvNetONB, and GrConvNetPP, respectively. As shown in Tab. 2, our GrConv allows for more flexible manipulation of dimensionality. Therefore,

Table 4: Comparison of the ONB and PP GrConvNets under different settings against other Grassmannian networks on the Radar dataset. The best three results are highlighted with red, blue, and cyan.

| Methods | Subspace dims | Ambient dims | Mean±Std | Max |
|---------|---------------|--------------|----------|-----|
| GrNet | 4 | 20->16 | 90.48 ± 0.76 | 91.73 |
| GyroGr | 4 | 20->20 | 90.64 ± 0.57 | 91.47 |
| GyroGr-Scaling | 4 | 20->20 | 88.88 ± 1.52 | 91.07 |
| GrConvNetONB | 4->4 | 20->16 | 93.92 ± 0.74 | 94.93 |
|  |  | 20->20 | 92.83 ± 0.66 | 93.73 |
|  | 4->8 | 20->16 | 94.77 ± 0.81 | 96.13 |
|  | 4->6 | 20->16 | 95.23 ± 0.96 | 96.67 |
| GrConvNetPP | 4->4 | 20->16 | 94.35 ± 0.42 | 94.8 |
|  |  | 20->20 | 94.56 ± 0.58 | 95.2 |
|  | 4->8 | 20->16 | 94.11 ± 0.58 | 95.07 |
|  | 4->6 | 20->16 | 94.51 ± 0.53 | 95.47 |

we also perform ablation studies on different subspace and ambient dimension settings. The experiments are conducted on the Radar dataset. Following Wang et al. (2024a), we model each radar signal as a multi-channel Grassmannian tensor, i.e., $[c, n, p]$ for the ONB and $[c, n, n]$ for the PP. The 5-fold average and maximum results are presented in Tab. 4, demonstrating that our GrConv significantly outperforms other Grassmannian transformation layers. Furthermore, varying the subspace dimension proves to be potentially beneficial, as our GrConv achieves the top two results under varying subspace dimensions. These observations highlight the effectiveness and flexibility of our GrConv.

## 9 CONCLUSION

This paper extends basic transformation layers, such as FC and convolutional layers, to operate on general manifolds. Our approach provides a natural, Riemannian-oriented generalization applicable more broadly than previous manifold-specific transformation layers. Empirically, we demonstrate our framework on five SPD geometries and two Grassmannian perspectives. Extensive experiments on radar and human action recognition tasks highlight the effectiveness and flexibility of our approach. We hope that our work will facilitate the development of deep networks for data with nontrivial geometries in machine learning.

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

APPENDIX CONTENTS

## A  GLOSSARY OF SYMBOLS

Tab. 5 summarizes all the notations in the main paper.

Table 5: Summary of notations.

| Notation | Explanation |
|---|---|
| $\{\mathcal{N}, g^{\mathcal{N}}\}$ | Riemannian manifold $\mathcal{N}$ with Riemannian metric $g^{\mathcal{N}}$ |
| $\{\mathcal{M}, g^{\mathcal{M}}\}$ | Riemannian manifold $\mathcal{M}$ with Riemannian metric $g^{\mathcal{M}}$ |
| $E$ | Origin of the interested manifold |
| $T_P\mathcal{M}$ | Tangent space at $P \in \mathcal{M}$ |
| $g_p(\cdot, \cdot)$ or $\langle \cdot, \cdot \rangle_P$ | Riemannian metric at $P$ |
| $\|\cdot\|_P$ | The norm induced by $\langle \cdot, \cdot \rangle_P$ on $T_P\mathcal{M}$ |
| $\mathrm{d}(\cdot, \cdot)$ | Geodesic distance |
| $\mathrm{Log}_P$ | Riemannian logarithm at $P$ |
| $\mathrm{Exp}_P$ | Riemannian exponentiation at $P$ |
| $\Gamma_{P \to Q}$ | Parallel transportation from $P$ to $Q$ along the geodesic |
| $f_{*,P}$ | Differential map of the smooth map $f$ at $P \in \mathcal{M}$ |
| $\{B_i\}_{i=1}^m$ | Standard orthonormal bases over $m$-dimensional $T_E\mathcal{M}$ |
| $\mathcal{S}_{++}^n$ | Space of $n \times n$ SPD matrices |
| $\mathcal{S}^n$ | Euclidean space of $n \times n$ symmetric matrices |
| $\mathcal{L}^n$ | Euclidean space of $n \times n$ lower triangular matrices |
| $\langle \cdot, \cdot \rangle$ | Standard Frobenius inner product |
| $\langle \cdot, \cdot \rangle^{(\alpha, \beta)}$ | O$(n)$-invariant Euclidean metric on $\mathcal{S}^n$ s.t. $\min(\alpha, \alpha + n\beta) > 0$ |
| $\|\cdot\|_{\mathrm{F}}$ | Frobenius Norm |
| $\log$ | Matrix logarithm |
| $\exp$ | Matrix exponentiation |
| $P^\theta$ | Matrix power for SPD matrix $P$ |
| $\mathcal{L}_P[\cdot]$ | Lyapunov operator by $P \in \mathcal{S}_{++}^n$ |
| $\mathscr{L}$ | Cholesky decomposition |
| $\mathrm{Dlog}$ | Diagonal element-wise logarithm |
| $\lfloor \cdot \rfloor$ | Strictly lower triangular part of a square matrix |
| $\mathbb{D}(\cdot)$ | A diagonal matrix with diagonal elements from a square matrix |
| $\mathrm{Gr}(p, n)$ | Grassmannian under the ONB perspective |
| $\widetilde{\mathrm{Gr}}(p, n)$ | Grassmannian under the projector perspective |
| $\mathcal{Q}(\cdot)$ | Return an orthogonal matrix by QR decomposition |
| $[\cdot, \cdot]$ | Matrix commutator |
| $I_{p,n}$ | Grassmannian identity under the ONB perspective |
| $\widetilde{I}_{p,n}$ | Grassmannian identity under the projector perspective |
| $I_n$ | $n \times n$ identity matrix |
| $\pi$ | Riemannian isometry from $\mathrm{Gr}(p, n)$ onto $\widetilde{\mathrm{Gr}}(p, n)$ |
| $\overline{(\cdot)}$ | $\overline{(\cdot)} = \widetilde{\mathrm{Log}}_{\widetilde{I}_{p,n}}(\cdot)$ with $\widetilde{\mathrm{Log}}$ as the Riemannian logarithm on $\widetilde{\mathrm{Gr}}(p, n)$ |
| $\mathbf{0}$ | Zero matrix with all the entities as zero |
| $\mathrm{St}(p, n)$ | Stiefel manifold of $n \times p$ column-wise orthogonal matrices |
| $\mathrm{GL}(n)$ | General linear group of $n \times n$ invertible matrices |
| $\mathrm{O}(n)$ | Orthogonal group of $n \times n$ orthogonal matrices |
| $\mathbb{R}^n$ | Euclidean space of $n$-dimensional vectors |

## B  RIEMANNIAN OPERATORS ON THE SPD AND GRASSMANNIAN MANIFOLDS

### B.1  RIEMANNIAN OPERATORS ON THE SPD MANIFOLD

Tabs. 6 and 7 summarizes the associated Riemannian operators and properties. Following Tab. 5, we further make the following notations. Given any SPD points $P, Q \in \mathcal{S}_{++}^n$ and tangent vectors $V, W \in T_P\mathcal{S}_{++}^n$, we denote $\widetilde{V} = \mathrm{Chol}_{*,P}(V), \widetilde{W} = \mathrm{Chol}_{*,P}(W), L = \mathrm{Chol}\, P$, and $K = \mathrm{Chol}\, Q$. The corresponding diagonal matrix with their diagonal elements are denoted as $\widetilde{\mathbb{V}}, \widetilde{\mathbb{W}}, \mathbb{L}$, and $\mathbb{K}$,

respectively. For the parallel transport under the BWM, we only present the case where $P, Q$ are commuting matrices, *i.e.,* $P = U\Sigma U^\top$ and $Q = U\Delta U^\top$.

The $O(n)$-invariant Euclidean metric on $\mathcal{S}^n$ (Thanwerdas & Pennec, 2023) is

$$\langle V, W\rangle^{(\alpha,\beta)} = \alpha\langle V, W\rangle + \beta\operatorname{tr}(V)\operatorname{tr}(W), \quad \text{with } \min(\alpha, \alpha + n\beta) > 0. \tag{34}$$

*Remark* B.1. We make the following remarks w.r.t. the geometries on the SPD manifold.

- **PEM & EM.** When the power equals 1, the associated PEM is reduced to the Euclidean Metric (EM) (Thanwerdas & Pennec, 2023, Sec. 3.1).

- **Incompleteness & Riemannian exponentiation.** As PEM and BWM are incomplete, their Riemannian exponential maps are locally defined. As shown by (Malagò et al., 2018, Prop. 9) and implied by Chen et al. (2024c); Thanwerdas & Pennec (2023), the restricted domains are

$$\begin{aligned} \text{PEM: } & P^\theta + P_{\theta*,P}(V) \in \mathcal{S}^n_{++}, \\ \text{BWM: } & \mathcal{L}_P[V] + I \in \mathcal{S}^n_{++}. \end{aligned} \tag{35}$$

The above restriction can be solved numerically, such as ReEig (Huang et al., 2017):

$$\widetilde{S} = U\max(\epsilon I, \Sigma)U^\top, \tag{36}$$

where $S \overset{\text{Eig}}{:=} U\Sigma U^\top$ is the Eigendecomposition.

Table 6: The Riemannian operators under LEM, AIM, and PEM on the SPD manifold.

| Operators | LEM | AIM | PEM |
|---|---|---|---|
| $g_P(V,W)$ | $\langle\log_{*,P}(V), \log_{*,P}(W)\rangle^{(\alpha,\beta)}$ | $\langle P^{-1}V, WP^{-1}\rangle^{(\alpha,\beta)}$ | $\frac{1}{\theta^2}\langle P_{\theta*,P}(V), P_{\theta*,P}(W)\rangle^{(\alpha,\beta)}$ |
| $\operatorname{Log}_P Q$ | $(\log_{*,P})^{-1}\left[\log(Q) - \log(P)\right]$ | $P^{\frac{1}{2}}\log\left(P^{-\frac{1}{2}}QP^{-\frac{1}{2}}\right)P^{\frac{1}{2}}$ | $(P_{\theta*,P})^{-1}\left(Q^\theta - P^\theta\right)$ |
| $\Gamma_{P\to Q}(V)$ | $(\log_{*,Q})^{-1} \circ \log_{*,P}(V)$ | $(QP^{-1})^{\frac{1}{2}}V(P^{-1}Q)^{\frac{1}{2}}$ | $(P_{\theta*,Q})^{-1} \circ P_{\theta*,P}(V)$ |
| $\operatorname{Exp}_P(V)$ | $\exp\left(\log(P) + \log_{*,P}(V)\right)$ | $P^{\frac{1}{2}}\exp\left(P^{-\frac{1}{2}}VP^{-\frac{1}{2}}\right)P^{\frac{1}{2}}$ | $\left(P^\theta + P_{\theta*,P}(V)\right)^{\frac{1}{\theta}}$ |
| Invariance | Lie group bi-invariance $O(n)$-invariance | Lie group left-invariance $GL(n)$-invariance | $O(n)$-invariance |
| References | Arsigny et al. (2005) Thanwerdas & Pennec (2023) | Pennec et al. (2006) Thanwerdas & Pennec (2019) | Dryden et al. (2010) Thanwerdas & Pennec (2023) Chen et al. (2024c) |

Table 7: The Riemannian operators under BWM and LCM on the SPD manifold.

| Operators | LCM | BWM |
|---|---|---|
| $g_P(V,W)$ | $\langle\lfloor\widetilde{V}\rfloor, \lfloor\widetilde{W}\rfloor\rangle + \langle\widetilde{\mathbb{V}}\mathbb{L}^{-1}, \widetilde{\mathbb{W}}\mathbb{L}^{-1}\rangle$ | $\frac{1}{2}\langle\mathcal{L}_P[V], W\rangle$ |
| $\operatorname{Log}_P Q$ | $(\text{Chol}^{-1})_{*,L}\left[\lfloor K\rfloor - \lfloor L\rfloor + \mathbb{L}\operatorname{Dlog}(\mathbb{L}^{-1}\mathbb{K})\right]$ | $(PQ)^{\frac{1}{2}} + (QP)^{\frac{1}{2}} - 2P$ |
| $\Gamma_{P\to Q}(V)$ | $(\text{Chol}^{-1})_{*,K}\left[\lfloor\widetilde{V}\rfloor + \mathbb{K}\mathbb{L}^{-1}\widetilde{\mathbb{V}}\right]$ | $U\left[\sqrt{\frac{\delta_i+\delta_j}{\sigma_i+\sigma_j}}\left[U^\top VU\right]_{ij}\right]U^\top$ |
| $\operatorname{Exp}_P(V)$ | $\text{Chol}^{-1}\left[\lfloor L\rfloor + \lfloor\widetilde{V}\rfloor + \mathbb{L}\operatorname{Dexp}(\mathbb{L}^{-1}\widetilde{V})\right]$ | $P + V + \mathcal{L}_P[V]P\mathcal{L}_P[V]$ |
| Invariance | Lie group bi-invariance | $O(n)$-invariance |
| References | Lin (2019) | Bhatia et al. (2019) Thanwerdas & Pennec (2023) |

## B.2 RIEMANNIAN OPERATORS ON THE GRASSMANNIAN

As the set of linear subspaces, the Grassmannian can naturally be represented by any of the orthonormal bases, which is called the OrthoNormal Basis (ONB) perspective. Under this perspective, the Grassmannian is the quotient of the Stiefel manifold (Bendokat et al., 2024), denoted as

Table 8: Riemannian operators on the Grassmannian.

| Operators | $\mathrm{Gr}(p,n)$ | $\widetilde{\mathrm{Gr}}(p,n)$ |
|---|---|---|
| $g_P(V,W)$ | $\langle V, W \rangle$ | $\frac{1}{2}\langle V, W \rangle$ |
| $\mathrm{Log}_P Q$ | $O \arctan(\Sigma) R^\top$ 
 $(I_n - PP^\top)Q(P^\top Q)^{-1} \overset{\mathrm{SVD}}{:=} O\Sigma R^\top$ | $\frac{1}{2}[\log\left((I_n - 2Q)(I_n - 2P)\right), P]$ |
| $\Gamma_{P \to Q}(V)$ | $\left(\begin{pmatrix} PR & O \end{pmatrix}\begin{pmatrix} -\sin(\Sigma) \\ \cos(\Sigma) \end{pmatrix}O^T + (I - OO^T)\right)V$ 
 $\mathrm{Log}_P(Q) \overset{\mathrm{SVD}}{:=} O\Sigma R^\top$ | $\exp([\log_P(Q), P])V\exp(-[\log_P(Q), P])$ |
| $\mathrm{Exp}_P V$ | $\begin{pmatrix} PR & O \end{pmatrix}\begin{pmatrix} \cos(\Sigma) \\ \sin(\Sigma) \end{pmatrix}R^\top$ 
 $V \overset{\mathrm{SVD}}{:=} O\Sigma R^\top$ | $\exp([V,P])P\exp(-[V,P])$ |
| References | Edelman et al. (1998) 
 Bendokat et al. (2024) | Batzies et al. (2015) 
 Bendokat et al. (2024) |

$\mathrm{Gr}(p,n) \cong \mathrm{St}(p,n)/\mathrm{O}(p)$. Each point is an equivalence class:

$$\mathrm{Gr}(p,n) = \{[U] : [U] := \{\widetilde{U} \in \mathrm{St}(p,n) \mid \widetilde{U} = UR, R \in \mathrm{O}(p)\}\}. \tag{37}$$

By abuse of notations, we use $[U]$ and $U$ interchangeably for elements of $\mathrm{Gr}(p,n)$. Each tangent space can be identified as a subspace of a corresponding tangent space on the Stiefel manifold, which is called horizontal space. Therefore, every tangent vector can be identified with a tangent vector in the horizontal space, called horizontal lift[2]. Under this identification, each tangent vector $V \in T_P\mathrm{Gr}(p,n)$ can be represented as

$$V = P_\perp B, \text{ with } B \in \mathbb{R}^{(n-p)\times p}, \tag{38}$$

where $P_\perp \in \mathrm{St}(n-p,n)$ is the orthogonal complement of $P$.

Another perspective is called the Projector Perspective (PP). As shown by Bendokat et al. (2024), the Grassmannian is an embedded submanifold of $\mathcal{S}^n$:

$$\widetilde{\mathrm{Gr}}(p,n) = \{P \in \mathcal{S}^n : P^2 = P, \mathrm{rank}(P) = p\}. \tag{39}$$

Therefore, each point can be represented as an $n \times n$ symmetric matrix. Under this perspective, any tangent vector $V \in T_P\widetilde{\mathrm{Gr}}(p,n)$ at $P \in \widetilde{\mathrm{Gr}}(p,n)$ can be represented as

$$V = Q\begin{pmatrix} 0 & B^T \\ B & 0 \end{pmatrix}Q^T, \text{ with } B \in \mathbb{R}^{(n-p)\times p}, \tag{40}$$

where $Q\widetilde{I}_{p,n}Q^\top = P$.

Supposing $P$ and $Q$ are the points on the Grassmannian $\mathrm{Gr}(p,n)$ ($\widetilde{\mathrm{Gr}}(p,n)$), and $V$ and $W$ are the tangent vectors over $T_P\mathrm{Gr}(p,n)$ ($T_P\widetilde{\mathrm{Gr}}(p,n)$), Tab. 8 summarizes the associated Riemannian operators following the notations in Tab. 5.

*Remark* B.2. We make the following remarks w.r.t. the Riemannian operators over the Grassmannian.

- **Cut locus & logarithm.** The Grassmannian Riemannian logarithm does not exists for any pair of $P$ and $Q$. As shwon by (Bendokat et al., 2024, Sec. 5), $\mathrm{Log}_P(Q)$ exists only if $P$ and $Q$ are not in each other's cut locus. However, this can be numerically solved, such as (Bendokat et al., 2024, Alg. 5.3) or using Moore–Penrose inverse for the inverse in the ONB logarithm (Nguyen, 2022a).

- **PP & ONB logarithm.** The matrix logarithm shown in the PP logarithm does not support backpropagation, as it can not be calculated by the SVD like the SPD matrix. However, the PP logarithm can be calculated via the ONB logarithm (Nguyen et al., 2024, Prop. 3.12). The latter can be backpropagated by the SVD. In this way, the PP logarithm can be integrated into the Pytorch deep learning framework.

---

[2]In this paper, the tangent vector under the ONB perspective is always considered as the horizontal lift.

## C    ADDITION DISCUSSIONS ON THE ORTHOGONAL BASIS

When the inner product $g_E$ on $T_E\mathcal{M}$ is the standard inner product, we use familiar $\{e_i\}_{i=1}^m$ the orthonormal basis. However, when $g_E$ is not standard, $\{e_i\}_{i=1}^m$ might not be orthonormal. In this case, we can always find one associated to $\{e_i\}_{i=1}^m$ by a linear isometry. We rewrite the inner product $g_E$ as

$$g_E(V, W) = \langle f(V), f(W)\rangle = f(V)^\top f(W), \forall V, W \in T_E\mathcal{M} \cong \mathbb{R}^m, \tag{41}$$

where $f$ is the linear isometry that pulls back the standard inner product $\langle \cdot, \cdot \rangle$ to $g_E$. Then, $\{B_i\}_{i=1}^m = \{f^{-1}(e_i)\}_{i=1}^m$ is the standard orthonormal bases over $\{T_E\mathcal{M}, g_E\}$.

## D    RIEMANNIAN FC LAYERS UNDER ISOMETRIES

The following theorem demonstrates that a Riemannian FC layer under isometric metrics can be computed by the following procedure: mapping, applying the Riemannian FC layer, and remapping.

**Theorem D.1** (Isometric FC Layers). *Given $n$-dimensional Riemannian manifolds $\left\{\widetilde{\mathcal{N}}, g^{\widetilde{\mathcal{N}}}\right\}$ and $\left\{\mathcal{N}, g^{\mathcal{N}}\right\}$ with a Riemannian isometry $\phi^{\mathcal{N}}: \widetilde{\mathcal{N}} \to \mathcal{N}$, and $m$-dimensional Riemannian manifolds $\left\{\widetilde{\mathcal{M}}, g^{\widetilde{\mathcal{M}}}\right\}$ and $\left\{\mathcal{M}, g^{\mathcal{M}}\right\}$ with $\phi^{\mathcal{M}}: \widetilde{\mathcal{M}} \to \mathcal{M}$ as a Riemannian isometry mapping origin $E^{\widetilde{\mathcal{M}}} \in \widetilde{\mathcal{M}}$ into the origin $E \in \mathcal{M}$, the Riemannian FC layer $\widetilde{\mathcal{F}}: \widetilde{\mathcal{N}} \to \widetilde{\mathcal{M}}$ can be calculated by $\mathcal{F}: \mathcal{N} \to \mathcal{M}$:*

$$\widetilde{\mathcal{F}}\left(\widetilde{X}; \widetilde{\mathbf{P}}, \widetilde{\mathbf{A}}\right) = \left(\phi^{\mathcal{M}}\right)^{-1}\left(\mathcal{F}\left(\phi^{\mathcal{N}}(\widetilde{X}); \mathbf{P}, \mathbf{A}\right)\right), \tag{42}$$

*where $\widetilde{\mathbf{P}} = \left\{\widetilde{P}_i \in \widetilde{\mathcal{N}}\right\}_{i=1}^m$ and $\widetilde{\mathbf{A}} = \left\{\widetilde{A}_i \in T_{\widetilde{P}_i}\widetilde{\mathcal{N}}\right\}_{i=1}^m$ are the FC parameters of $\widetilde{\mathcal{F}}$, while $\mathbf{P} = \left\{\phi^{\mathcal{N}}(\widetilde{P}_i)\right\}_{i=1}^m$ and $\mathbf{A} = \left\{\phi^{\mathcal{N}}_{*,\widetilde{P}_i}(\widetilde{A}_i)\right\}_{i=1}^m$ are the FC parameters of $\mathcal{F}$.*

*Proof.* First we show the correspondence between the standard orthonormal bases $\{\widetilde{B}_i \in \widetilde{\mathcal{M}}\}$ and $\{B_i \in \mathcal{M}\}$. Obviously, $\{\widetilde{B}_i \in \widetilde{\mathcal{M}}\}$ is orthonormal iff $\{B_i \in \mathcal{M}\}$ is orthonormal. We only need to show the standardness. The Riemannian metric $g^{\widetilde{\mathcal{M}}}$ has the following:

$$\begin{aligned}
g^{\widetilde{\mathcal{M}}}_{\widetilde{E}}(V, W) &\overset{(1)}{=} g^{\mathcal{M}}_E\left(\phi^{\mathcal{M}}_{*,\widetilde{E}}(V), \phi^{\mathcal{M}}_{*,\widetilde{E}}(V)\right) \\
&= \left\langle f \circ \phi^{\mathcal{M}}_{*,\widetilde{E}}(V), f \circ \phi^{\mathcal{M}}_{*,\widetilde{E}}(V)\right\rangle,
\end{aligned} \tag{43}$$

where $f$ is the linear isomorphism that pulls back the standard Frobenius inner product to $g^{\mathcal{M}}_E$. Here, (1) comes from the isometry. Therefore, for each $i$, we have the following

$$\begin{aligned}
\widetilde{B}_i &= (f \circ \phi^{\mathcal{M}}_{*,\widetilde{E}})^{-1}(E_i) \\
&\overset{(1)}{=} \left(\phi^{\mathcal{M}}_{*,\widetilde{E}}\right)^{-1}(B_i),
\end{aligned} \tag{44}$$

where (1) comes from $B_i = f^{-1}(E_i), \forall i = 1, \cdots, n$.

We now demonstrate the correspondence between the FC layers as follows:

$$\begin{aligned}
Y &= \mathrm{Exp}^{\widetilde{\mathcal{M}}}_{\widetilde{E}}\left(\sum_{i=1}^m \left(\langle \mathrm{Log}^{\widetilde{\mathcal{N}}}_{\widetilde{P}_i}(\widetilde{X}), \widetilde{A}_i\rangle^{\widetilde{\mathcal{N}}}_{\widetilde{P}_i} \widetilde{B}_i\right)\right) \\
&\overset{(1)}{=} \left(\phi^{\mathcal{M}}\right)^{-1}\left(\mathrm{Exp}^{\mathcal{M}}_E\left(\phi^{\mathcal{M}}_{*,\widetilde{E}}\left[\sum_{i=1}^m \left(\langle \mathrm{Log}^{\mathcal{N}}_{P_i}(X), A_i\rangle^{\mathcal{N}}_{P_i} \widetilde{B}_i\right)\right]\right)\right) \\
&\overset{(2)}{=} \left(\phi^{\mathcal{M}}\right)^{-1}\left(\mathrm{Exp}^{\mathcal{M}}_E\left(\sum_{i=1}^m \left(\langle \mathrm{Log}^{\mathcal{N}}_{P_i}(X), A_i\rangle^{\mathcal{N}}_{P_i} B_i\right)\right)\right),
\end{aligned} \tag{45}$$

where $B_i = \phi^{\mathcal{M}}_{*,\widetilde{E}}(\widetilde{B}_i)$, $A_i = \phi^{\mathcal{N}}_{*,\widetilde{P}_i}(\widetilde{A}_i)$, $X = \phi^{\mathcal{N}}(\widetilde{X})$, and $P_i = \phi^{\mathcal{N}}(\widetilde{P}_i)$. The above derivation comes from the following.

(1) The isometry of $\phi^{\mathcal{M}}$ and $\phi^{\mathcal{N}}$;

(2) The linearity of $\phi^{\mathcal{M}}_{*,\widetilde{E}}$.

$\square$

# E  RELATION WITH THE GYRO SPD FULLY CONNECTED LAYERS

We first review some related SPD gyro structures (Nguyen & Yang, 2023). Given $P, Q$ in $\{\mathcal{S}^n_{++}, g\}$ with $g$ as AIM, LEM or LCM, and $t \in \mathbb{R}$, the gyro structures induced by $g$ are defined as follows:

$$\text{Gyro addition: } P \oplus Q = \text{Exp}_P \left( \Gamma_{I \to P} \left( \text{Log}_I(Q) \right) \right), \tag{46}$$

$$\text{Gyro scalar product: } t \otimes P = \text{Exp}_I \left( t \, \text{Log}_I(P) \right), \tag{47}$$

$$\text{Gyro inverse: } \ominus P = -1 \otimes P = \text{Exp}_I \left( -\text{Log}_I(P) \right), \tag{48}$$

$$\text{Gyro inner product: } \langle P, Q \rangle_{\text{gr}} = \langle \text{Log}_I(P), \text{Log}_I(Q) \rangle_I, \tag{49}$$

where $\text{Log}_I$ and $\langle \cdot, \cdot \rangle_I$ is the Riemannian logarithm and metric at the identity matrix $I$. As shown by Nguyen (2022a), the gyro addition and scalar product under AIM, LEM, and LCM form gyrovector spaces.

Based on these gyro structures, Nguyen et al. (2024) introduces the gyro SPD FC layers under AIM, LEM, and LCM, respectively. We review their results in the following.

**Theorem E.1** (Gyro SPD FC Layers (Nguyen et al., 2024)). *The gyro SPD FC layers under standard LEM, AIM, and LCM are*

$$LEM : Y = \exp\left(V^{\text{LE}}\right), V^{\text{LE}}_{ij} = \begin{cases} v^{\text{LE}}_{ii}(S), & \text{if } i = j \\ \frac{1}{\sqrt{2}} v^{\text{LE}}_{ij}(S), & \text{if } i > j \\ V^{\text{LE}}_{ji}, & \text{otherwise} \end{cases} \tag{50}$$

$$AIM : Y = \exp\left(V^{\text{AI}}\right), V^{\text{AI}}_{ij} = \begin{cases} v^{\text{AI}}_{ii}(S) + \eta \sum_{k=1}^m v^{\text{AI}}_{kk}(S), & \text{if } i = j \\ \frac{1}{\sqrt{2}} v^{\text{AI}}_{ij}(S), & \text{if } i > j \\ V^{\text{AI}}_{ji}, & \text{otherwise} \end{cases} \tag{51}$$

$$LCM : Y = V^{\text{LC}}(V^{\text{LC}})^\top, V^{\text{LC}}_{ij} = \begin{cases} \exp\left(v^{\text{LC}}_{ii}(S)\right), & \text{if } i = j \\ v^{\text{LC}}_{ij}(S), & \text{if } i > j \\ 0, & \text{otherwise} \end{cases} \tag{52}$$

*where $\eta = \frac{1}{n} \left( \frac{1}{\sqrt{1+n\beta}} - 1 \right)$, and $v^g_{ij} = \langle \ominus P_{ij} \oplus S, W_{ij} \rangle_{\text{gr}}$ with $g$ as LEM, AIM, or LCM. Here, $P_{ij}, W_{ij} \in \mathcal{S}^n_{++}, \forall i \geq j, i, j = 1, \cdots, m$.*

**Proposition E.2.** *Our LEM ($(\alpha, \beta) = (1, 0)$), AIM ($(\alpha, \beta) = (1, \beta)$), and LCM SPD FC layers incorporate the LEM, AIM, and LCM gyro SPD FC layers, respectively.*

*Proof.* Comparing Thm. E.1 with our Thm. 5.1, we only need to show the equality of $v_{ij}$ in the gyro and our framework.

$$\begin{aligned} v^g_{ij} &= \langle \ominus P_{ij} \oplus S, W_{ij} \rangle_{\text{gr}} \\ &\overset{(1)}{=} \left\langle \text{Exp}_I \left( \Gamma_{P_{ij} \to I} \left( \text{Log}_{P_{ij}}(S) \right) \right), W_{ij} \right\rangle_{\text{gr}} \\ &\overset{(2)}{=} \left\langle \Gamma_{P_{ij} \to I} \left( \text{Log}_{P_{ij}}(S) \right), \text{Log}_I(W_{ij}) \right\rangle_I \\ &\overset{(3)}{=} \left\langle \text{Log}_{P_{ij}}(S), \Gamma_{I \to P_{ij}} \left( \text{Log}_I(W_{ij}) \right) \right\rangle_{P_{ij}} \end{aligned} \tag{53}$$

The above derivation comes from the following.

(1) $\ominus P_{ij} \oplus S = \text{Exp}_I \left( \Gamma_{P_{ij} \to I} \left( \text{Log}_{P_{ij}}(S) \right) \right)$ (Nguyen et al., 2024, Eq. (6));

(2) Eq. (49);

(3) Norm preservation of parallel transport (Do Carmo & Flaherty Francis, 1992, Def. 3.1).

Setting $A_{ij} = \Gamma_{I \to P} \left( \text{Log}_I(W_{ij}) \right) \in T_{P_{ij}} \mathcal{S}^n_{++}$, we recover Eqs. (92), (93) and (95) for each metric.

$\square$

# F    TRIVIALIZED SPD FULLY CONNECTED LAYERS

**Theorem F.1** (Trivialized SPD FC Layers). *Trivializing each $P_{ij}$ in Thm. 5.1 as $\mathrm{Exp}_I(\gamma_{ij}[Z_{ij}])$, $v_{ij}(S)$ under different metrics can be further simplified:*

$$LEM : \langle \log(S), Z_{ij} \rangle^{(\alpha,\beta)} - \gamma_{ij} \|Z_{ij}\|^{(\alpha,\beta)}, \tag{54}$$

$$AIM : \left\langle \log\left(\exp\left(-\frac{\gamma_{ij}}{2}[Z_{ij}]\right) S \exp\left(-\frac{\gamma_{ij}}{2}[Z_{ij}]\right)\right), Z_{ij} \right\rangle^{(\alpha,\beta)}, \tag{55}$$

$$PEM : \left\langle S^\theta - (I + \theta\gamma_{ij}[Z_{ij}]), Z_{ij} \right\rangle^{(\alpha,\beta)}, \tag{56}$$

$$LCM : \left\langle \lfloor K \rfloor + \mathrm{Dlog}(\mathbb{K}) - \left(\gamma_{ij}\lfloor [Z_{ij}] \rfloor + \frac{1}{2}\gamma_{ij}\mathbb{D}([Z_{ij}])\right), \lfloor Z_{ij} \rfloor + \frac{1}{2}\mathbb{Z}_{ij} \right\rangle, \tag{57}$$

*where $\|\cdot\|^{(\alpha,\beta)}$ is the norm induced by $\langle \cdot, \cdot \rangle^{(\alpha,\beta)}$, and $\mathbb{D}(\cdot)$ returns a diagonal matrix with diagonal elements from the input square matrix.*

*Proof.* **LEM:**

$$\langle \log(S) - \log(P_{ij}), Z_{ij} \rangle^{(\alpha,\beta)} \overset{(1)}{=} \langle \log(S) - \gamma_{ij}[Z_{ij}], Z_{ij} \rangle^{(\alpha,\beta)}$$
$$\overset{(2)}{=} \langle \log(S), Z_{ij} \rangle^{(\alpha,\beta)} - \gamma_{ij} \|Z_{ij}\|^{(\alpha,\beta)}, \tag{58}$$

The above comes from the following.

(1) Eq. (108);

(2) $[Z_{ij}] = \frac{Z_{ij}}{\|Z_{ij}\|^{(\alpha,\beta)}}$.

**AIM:** This can be obtained by the following:

$$\exp\left(\gamma_{ij}[Z_{ij}]\right)^{-\frac{1}{2}} = \exp\left(-\frac{\gamma_{ij}}{2}[Z_{ij}]\right). \tag{59}$$

**PEM:** This can be obtained by Eq. (109).

**LCM:**

$$\left\langle \lfloor K \rfloor - \lfloor L_{ij} \rfloor + \mathrm{Dlog}(\mathbb{K}\mathbb{L}_{ij}^{-1}), \lfloor Z_{ij} \rfloor + \frac{1}{2}\mathbb{Z}_{ij} \right\rangle$$
$$= \left\langle \lfloor K \rfloor + \mathrm{Dlog}(\mathbb{K}) - (\lfloor L_{ij} \rfloor + \mathrm{Dlog}(\mathbb{L}_{ij})), \lfloor Z_{ij} \rfloor + \frac{1}{2}\mathbb{Z}_{ij} \right\rangle \tag{60}$$
$$\overset{(1)}{=} \left\langle \lfloor K \rfloor + \mathrm{Dlog}(\mathbb{K}) - \left(\gamma_{ij}\lfloor [Z_{ij}] \rfloor + \frac{1}{2}\gamma_{ij}\mathbb{D}([Z_{ij}])\right), \lfloor Z_{ij} \rfloor + \frac{1}{2}\mathbb{Z}_{ij} \right\rangle,$$

where (2) comes from Eq. (110).    $\square$

*Remark F.2.* Due to the incompleteness of PEM and BWM, their exponential maps at $I$, $\mathrm{Exp}_I(V)$, are well-defined locally:

$$PEM: I + \theta V \in \mathcal{S}_{++}^n,$$
$$BWM: I + \frac{1}{2}V \in \mathcal{S}_{++}^n. \tag{61}$$

The above restriction can be solved numerically, such as ReEig (Huang et al., 2017):

$$\widetilde{S} = U \max(\epsilon I, \Sigma)U^\top, \tag{62}$$

where $S \overset{\mathrm{Eig}}{:=} U\Sigma U^\top$ is the eigendecomposition.

## G  Trivialized SPD Multinomial Logistic Regression

In our implementation, we trivialize the SPD parameters in the SPD MLR as Sec. 4.3. The SPD MLRs proposed in Chen et al. (2024c) under five geometries can be further simplified. For simplicity, we do not involve the power deformation (Chen et al., 2024c).

**Theorem G.1** (Trivialized SPD MLRs). [↓] *Given $C$ classes and an SPD feature $S$, the SPD MLRs, $p(y = k \mid S \in \mathcal{S}_{++}^n)$, are proportional to*

$$LEM : \exp\left[\langle \log(S), Z_k \rangle^{(\alpha,\beta)} - \gamma_k \left\| Z_k \right\|^{(\alpha,\beta)} \right], \tag{63}$$

$$AIM : \left[\exp\left\langle \log\left(\exp\left(-\frac{\gamma_k}{2}[Z_k]\right) S \exp\left(-\frac{\gamma_k}{2}[Z_k]\right)\right), Z_k \right\rangle^{(\alpha,\beta)}\right], \tag{64}$$

$$PEM : \frac{1}{\theta} \exp\left[\left\langle S^\theta - (I + \theta\gamma_k[Z_k]), Z_k \right\rangle^{(\alpha,\beta)}\right], \tag{65}$$

$$LCM : \exp\left[\left\langle \lfloor K \rfloor + \mathrm{Dlog}(\mathbb{K}) - \left(\gamma_k \lfloor [Z_k] \rfloor + \frac{1}{2}\gamma_k \mathbb{D}([Z_k])\right), \lfloor Z_k \rfloor + \frac{1}{2}\mathbb{Z}_k \right\rangle\right], \tag{66}$$

$$BWM : \exp\left[\frac{1}{2}\left\langle (P_k S)^{\frac{1}{2}} + (S P_k)^{\frac{1}{2}} - 2P_k, \mathcal{L}_{P_k}(L_k Z_k L_k^\top) \right\rangle\right], \tag{67}$$

*where $Z_k \in T_I \mathcal{S}_{++}^n \backslash \{0\}$ is a symmetric matrix, $L_k = \mathrm{Chol}(P_k)$ is the Cholesky factor of $P_k$ with $P_k = (I + \frac{1}{2}\gamma_k[Z_k])^2$. Here $\{Z_k \in \mathcal{S}^n\}_{k=1}^C$ and $\{\gamma_k \in \mathbb{R}\}_{k=1}^C$ are the MLR parameters.*

*Proof.* For each class $k$, the expression of $v_k$ in the SPD MLR (Chen et al., 2024c, Thm. 4.2) has been reviewed in App. K.3. For MLR under each metric $g$, we parameterize the each parameter $P_k \in \mathcal{S}_{++}^n$ by $Z_k$ and $\gamma_k$ by

$$P_k = \mathrm{Exp}_I^g(\gamma_k[Z_k]), \tag{68}$$

with $[Z_k]$ as the unit vector of $Z_k$. Under this parameterization, the MLRs under LEM, AIM, PEM, and LCM can be further simplified, which has been implied by Thm. F.1. □

*Remark* G.2. Similar to the SPD FC layer, due to the incompleteness of PEM and BWM, the associated parameterization should follow

$$PEM: I + \theta\gamma_k[Z_k] \in \mathcal{S}_{++}^n, \tag{69}$$

$$BWM: I + \frac{1}{2}\gamma_k[Z_k] \in \mathcal{S}_{++}^n. \tag{70}$$

## H  Review of previous Grassmannian transformation layers

This section briefly reviews several popular Grassmannian transformation layers.

**FRMap + ReOrth.** Given input Grassmannian $X \in \mathrm{Gr}(p, q)$, Huang et al. (2018) first used Full Rank Map (FRMap) to first transform the input orthonormal matrices of subspaces to new matrices by a linear mapping function, and then applied QR decomposition to recover the orthogonality:

$$Y = \mathcal{Q}(WX), \tag{71}$$

where $W \in \mathbb{R}^{m \times n}$ is a row-wisely orthogonal parameter, and $\mathcal{Q}(\cdot)$ returns the orthogonal matrix in the QR decomposition.

**PP & ONB Scaling.** Nguyen (2022a); Nguyen & Yang (2023) proposed matrix scaling for the PP and ONB Grassmannian, respectively. Given $P = XX^\top \in \widetilde{\mathrm{Gr}}(p, n)$ with $X \in \mathrm{Gr}(p, n)$, the operations are defined as

$$\mathbf{PP:}\ Y = \exp\left(\begin{bmatrix} 0 & W * B \\ -(W * B)^T & 0 \end{bmatrix}\right) \widetilde{I}_{p,n} \exp\left(-\begin{bmatrix} 0 & W * B \\ -(W * B)^T & 0 \end{bmatrix}\right), \tag{72}$$

$$\mathbf{ONB:}\ Y = \exp\left(\begin{bmatrix} 0 & W * B \\ -(W * B)^T & 0 \end{bmatrix}\right) I_{p,n}, \tag{73}$$

where $*$ denotes the Hadamard product and $B \in \mathbb{R}^{(n-p) \times p}$ is a Euclidean parameter. Here, $X = \exp\left(\begin{bmatrix} 0 & B \\ -B^T & 0 \end{bmatrix}\right) I_{p,n}$.

**GrTrans.** Nguyen & Yang (2023) adopted the Grassmannian Gyro group translation (GrTrans) to transform the ONB and PP Grassmannian features. Given $X \in \widetilde{\mathrm{Gr}}(p, n)$ ( or $X \in \mathrm{Gr}(p, n)$), the operation is defined as

$$Y = W \oplus X, \tag{74}$$

where $\oplus$ is the Grassmannian PP (ONB) gyro addition (Nguyen & Yang, 2023, Sec. 2.3), and $W \in \widetilde{\mathrm{Gr}}(p, n)$ (or $W \in \mathrm{Gr}(p, n)$) is a Grassmannian parameter.

## I  EXPERIMENTAL DETAILS

### I.1  DETAILS OF THE EXPERIMENTS ON THE SPD MANIFOLD

#### I.1.1  DATASETS

**Radar**[3] (**Brooks et al., 2019**).  It consists of 3,000 synthetic radar signals equally distributed in 3 classes.

**HDM05**[4] (**Müller et al., 2007**).  It consists of 2,273 skeleton-based motion capture sequences executed by different actors. Each frame consists of 3D coordinates of 31 joints. We remove the under-represented clips, trimming the dataset down to 2086 instances scattered throughout 117 classes. We randomly select 50% of the samples from each category for training and the remaining 50% for testing.

**FPHA**[5] (**Garcia-Hernando et al., 2018**).  It includes 1,175 skeleton-based first-person hand gesture videos of 45 different categories with 600 clips for training and 575 for testing. Each frame contains the 3D coordinates of 21 hand joints.

For the HDM05 and FPHA datasets, we preprocess each sequence using the code[6] provided by Vemulapalli et al. (2014) to normalize body part lengths and ensure invariance to scale and view.

#### I.1.2  SPD MODELLING

For our SPDConvNets, we follow Wang et al. (2024a); Nguyen et al. (2024) to model each sample into a multi-channel SPD tensor. For the Radar dataset, we follow Wang et al. (2024a) to use the temporal convolution followed by a covariance pooling layer to obtain a multi-channel covariance $[c, 20, 20]$ tensor. For the HDM05 and FPHA datasets, we follow Nguyen et al. (2024, Sec. D.2.2) to model each skeleton sequence into a multi-channel covariance tensor $[c, n, n]$. Specifically, we first identify a closest left (right) neighbor of every joint based on their distance to the hip (wrist) joint, and then combine the 3D coordinates of each joint and those of its left (right) neighbor to create a feature vector for the joint. For a given frame $t$, we compute its Gaussian embedding (Lovrić et al., 2000):

$$Y_t = (\det \Sigma_t)^{-\frac{1}{n+1}} \left[ \begin{array}{cc} \Sigma_t + \mu_t \left(\mu_t\right)^T & \mu_t \\ \left(\mu_t\right)^T & 1 \end{array} \right], \tag{75}$$

where $\mu_t$ and $\Sigma_t$ are the mean vector and covariance matrix computed from the set of feature vectors within the frame. The lower part of matrix $\log\left(Y_t\right)$ is flattened to obtain a vector $\tilde{v}_t$. All vectors $\tilde{v}_t$ within a time window $[t, t + c - 1]$, where $c$ is determined from a temporal pyramid representation of the sequence (the number of temporal pyramids is set to 2 in our experiments), are used to compute a covariance matrix as

$$Z_t = \frac{1}{c} \sum_{i=t}^{t+c-1} \left(\tilde{v}_i - \overline{v}_t\right) \left(\tilde{v}_i - \overline{v}_t\right)^T, \tag{76}$$

where $\overline{v}_t = \frac{1}{c} \sum_{i=t}^{t+c-1} \tilde{v}_i$. The resulting $\{Z_t\}$ is the input covariance tensor. On the FPHA dataset, we generate the covariance based on three sets of neighbors: left, right, and vertical (bottom) neighbors.

For other SPD baselines, such as SPDNet, SPDNetBN, LieBN, MLR, and RResNet, each sequence is represented by a global covariance representation (Huang & Van Gool, 2017; Brooks et al., 2019). The sizes of the covariance matrices are $20 \times 20$, $93 \times 93$, and $63 \times 63$ for Radar, HDM05, and FPHA datasets, respectively.

---

[3] https://www.dropbox.com/s/dfnlx2bnyh3kjwy/data.zip?dl=0
[4] https://resources.mpi-inf.mpg.de/HDM05/
[5] https://github.com/guiggh/hand_pose_action
[6] https://ravitejav.weebly.com/kbac.html

## I.2 IMPLEMENTATION DETAILS

**Comparative methods.** We follow the official Pytorch code of SPDNetBN[7] to implement SPDNet and SPDNetBN. For LieBN[8], we focus on the instantiation under AIM and LCM, while for RResNet[9], we implement the ones induced by LEM and AIM. For SPD MLR[10], we use LCM on the HDM05 datasets, and AIM for the rest two datasets.

**SPDConvNets.** The output dimensions of the SPD convolutional layer are $8 \times 8$, $34 \times 34$, and $22 \times 22$ for the Radar, HDM05, and FPHA datasets, respectively. We primarily use the AMSGrad (Reddi et al., 2019) optimizer, except for SPDConvNet-LEM and SPDConvNet-AIM on the HDM05 dataset, where SGD (Robbins & Monro, 1951) is employed. Weight decay is set to zero, except for SPDConvNet-PEM on the FPHA dataset, where it is $5e^{-4}$. The matrix power in SPDConvNet-PEM is set as 0.5, 0.25, and 0.25 for the three datasets. Since matrix power can deform the latent Riemannian metric (Chen et al., 2024c, Fig. 1), we also apply matrix power $(\cdot)^\theta$ before the convolutional layer in SPDConvNet-AIM, -LCM, and -BWM to activate the latent geometries. The batch size is set to 30 with a training epoch of 150. Tab. 9 summarizes the training hyper-parameters.

Table 9: Training hyer-parameters in SPDConvNets

| Dataset | Model | $\theta$ | Optimizer | Learning Rate |
|---|---|---|---|---|
| Radar | SPDConvNet-LEM | N/A | AMSGrad | $5e^{-3}$ |
| | SPDConvNet-AIM | 0.25 | AMSGrad | $5e^{-4}$ |
| | SPDConvNet-PEM | N/A | AMSGrad | $1e^{-2}$ |
| | SPDConvNet-LCM | 0.25 | AMSGrad | $5e^{-4}$ |
| | SPDConvNet-BWM | N/A | AMSGrad | $5e^{-4}$ |
| HDM05 | SPDConvNet-LEM | N/A | SGD | $5e^{-3}$ |
| | SPDConvNet-AIM | N/A | SGD | $5e^{-3}$ |
| | SPDConvNet-PEM | N/A | AMSGrad | $1e^{-3}$ |
| | SPDConvNet-LCM | N/A | AMSGrad | $1e^{-3}$ |
| | SPDConvNet-BWM | N/A | AMSGrad | $1e^{-3}$ |
| FPHA | SPDConvNet-LEM | N/A | AMSGrad | $1e^{-4}$ |
| | SPDConvNet-AIM | N/A | AMSGrad | $1e^{-4}$ |
| | SPDConvNet-PEM | N/A | AMSGrad | $1e^{-3}$ |
| | SPDConvNet-LCM | -0.25 | AMSGrad | $1e^{-3}$ |
| | SPDConvNet-BWM | -0.25 | AMSGrad | $1e^{-4}$ |

## I.3 DETAILS OF THE EXPERIMENTS ON THE GRASSMANNIAN

**Grassmannian Modelling.** As Grassmannian descriptors can be derived by the SVD of the covariance (Huang et al., 2018; Nguyen & Yang, 2023), we map the multi-channel Radar covariance into a $[c, n, p]$ ONB Grassmannian tensor via the SVD decomposition. The PP Grassmannian features can be derived from the ONB Grassmannian features via the isometry $\pi(\cdot): \mathrm{Gr}(p,n) \to \widetilde{\mathrm{Gr}}(p,n)$:

$$\pi(U) = UU^\top, \forall U \in \mathrm{Gr}(p,n). \tag{77}$$

**Implementation details.** Since GrNet is officially implemented by Matlab, we carefully re-implemented it using PyTorch. Additionally, as both GryroGr and GryroGr-Scaling do not release official code, we re-implemented them based on the original papers (Nguyen, 2022a; Nguyen & Yang, 2023). For all comparative methods, we use SGD with a learning rate of $5e^{-2}$. For training our ONB and PP GrConvNets, we use AMSGrad with a learning rate of $5e^{-3}$. The batch size is set to 30 with a training epoch of 150.

## I.4 TRAINING EFFICIENCY

[7]https://proceedings.neurips.cc/paper_files/paper/2019/file/6e69ebbfad976d4637bb4b39de261bf7-Supplemental.zip
[8]https://github.com/GitZH-Chen/LieBN
[9]https://github.com/CUAI/Riemannian-Residual-Neural-Networks
[10]https://github.com/GitZH-Chen/SPDMLR

Table 10: Training efficiency (second / epoch).

| Method | Radar | HDM05 | FPHA |
|---|---|---|---|
| SPDNet | 0.66 | 0.50 | 0.28 |
| SPDNetBN | 1.25 | 0.94 | 0.58 |
| SPDResNet-AIM | 0.96 | 1.23 | 0.69 |
| SPDResNet-LEM | 0.77 | 0.55 | 0.25 |
| SPDNetLieBN-AIM | 1.21 | 1.15 | 0.97 |
| SPDNetLieBN-LCM | 1.10 | 1.11 | 0.59 |
| SPDNetMLR | 0.96 | 5.46 | 6.36 |
| SPDConvNet-LEM | 0.86 | 0.74 | 0.74 |
| SPDConvNet-AIM | 5.09 | 101.80 | 51.14 |
| SPDConvNet-PEM | 1.09 | 7.10 | 1.57 |
| SPDConvNet-LCM | 0.65 | 0.59 | 0.53 |
| SPDConvNet-BWM | 6.07 | 110.51 | 56.07 |

Tab. 10 presents the average training time per epoch of each SPD network. On the HDM05 and FPHA datasets, all baseline methods involve SVD on relatively large matrices, which are more efficiently executed on a CPU. Consequently, these methods are run on a CPU, while all other cases are executed on a single A6000 GPU. We have the following observations:

- **The efficiency of SPDConvNet varies across metrics.** The most efficient metric is LCM, where our model even achieves comparable efficiency to the vanilla SPDNet. However, AIM and BWM demonstrate significant computational burden, primarily due to their complex Riemannian computations.

- **Our trivialization improves efficiency.** On the HDM05 dataset, SPDNetMLR is implemented under LCM. Similarly, our SPDNetMLR-LCM also employs LCM-based MLR. However, SPDNetMLR-LCM achieves substantially lower training time. This improvement can be attributed to our trivialization, which simplifies the final expression (App. G).

## J    APPLICATIONS TO HYPERBOLIC SPACES

Hyperbolic Neural Networks (HNNs) have recently shown success in different applications (Ganea et al., 2018; Shimizu et al., 2020; Chami et al., 2019; Skopek et al., 2020; Bdeir et al., 2024; Fu et al., 2024). This section applies our Riemannian FC (Thm. 4.2) into the hyperbolic space.

### J.1    GEOMETRIES OF THE HYPERBOLIC SPACE

There are five models over the hyperbolic space (Cannon et al., 1997). We focus on the Poincaré ball and hyperboloid models:

$$\text{Poincaré ball: } \mathbb{P}_K^n = \left\{ x \in \mathbb{R}^n \mid \|x\|^2 < -\frac{1}{K} \right\} \tag{78}$$

$$\text{Hyperboloid: } \mathbb{H}_K^n = \left\{ x \in \mathbb{R}^{n+1} \mid \|x\|_{\mathcal{L}}^2 = \frac{1}{K} \right\}, \tag{79}$$

where $\|x\|_{\mathcal{L}}^2 = \sum_{i=2}^{n+1} x_i^2 - x_1^2$ is the Lorentz inner product, and $\|\cdot\|$ is the standard $L_2$ norm induced by the standard inner product $\langle \cdot, \cdot \rangle$. Here, $K < 0$ is the constant sectional curvature.

As shown by Ungar (2022), the Poincaré ball model admits a gyrovector space structure, which is a natural generalization of vector space in the manifold. The gyro addition, known as Möbius addition, is defined as

$$x \oplus_K y = \frac{\left(1 - 2K\langle x, y \rangle - K\|y\|^2\right) x + \left(1 + K\|x\|^2\right) y}{1 - 2K\langle x, y \rangle_2 + K^2\|x\|^2\|y\|^2}, \tag{80}$$

For parallel transport over the Poincaré ball, we further need the notion of gyration (Ungar, 2022):

$$\text{gyr}[x, y]z = \ominus_K (x \oplus_K y) \oplus_K (x \oplus_K (y \oplus_K z)), \forall x, y, z \in \mathbb{P}_K^n. \tag{81}$$

All Riemannian operators on Poincaré ball and hyperboloid models are relatively simple and have close-form expressions, which are summarized in Tab. 11.

Table 11: Riemannian operators on the hyperbolic space ($K < 0$).

| Operators | $\mathbb{P}^n_K = \left\{ x \in \mathbb{R}^n \mid \|x\|^2 < -\frac{1}{K} \right\}$ | $\mathbb{H}^n_K = \left\{ x \in \mathbb{R}^{n+1} \mid \|x\|^2_{\mathcal{L}} = \frac{1}{K} \right\}$, with $\|x\|^2_{\mathcal{L}} = \sum_{i=2}^{n+1} x_i^2 - x_1^2$ |
|---|---|---|
| $g_x(v, w)$ | $(\lambda_x^K)^2 \langle v, w \rangle$ $\lambda_x^K = \frac{2}{(1+K\|x\|^2)}$ | $\langle v, w \rangle_{\mathcal{L}} = \sum_{i=2}^{n+1} v_i w_i - v_1 w_1$ |
| $\mathrm{Log}_x(y)$ | $\frac{2}{\sqrt{\|K\|}\lambda_x^K} \tanh^{-1}\left( \sqrt{\|K\|} \|-x \oplus_K y\| \right) \frac{-x \oplus_K y}{\|-x \oplus_K y\|}$ | $\frac{\cosh^{-1}(K\langle x,y\rangle_{\mathcal{L}})}{\sinh\left(\cosh^{-1}(K\langle x,y\rangle_{\mathcal{L}})\right)} (y - K\langle x,y\rangle_{\mathcal{L}} x)$ |
| $\Gamma_{x \to y}(v)$ | $\frac{\lambda_x^K}{\lambda_y^K} \mathrm{gyr}[y, -x]v$ | $v - \frac{K\langle y,v\rangle_{\mathcal{L}}}{1+K\langle x,y\rangle_{\mathcal{L}}}(x + y)$ |
| $\mathrm{Exp}_x(v)$ | $x \oplus_K \left( \tanh\left( \sqrt{\|K\|}\frac{\lambda_x^K\|v\|}{2} \right) \frac{v}{\sqrt{\|K\|}\|v\|} \right)$ | $\cosh\left( \sqrt{\|K\|}\|v\|_{\mathcal{L}} \right) x + \sinh\left( \sqrt{\|K\|}\|v\|_{\mathcal{L}} \right) \frac{v}{\sqrt{\|K\|}\|v\|_{\mathcal{L}}}$ |
| References | Ganea et al. (2018) Skopek et al. (2020) Ungar (2022) | Petersen (2006) Skopek et al. (2020) |

## J.2 RIEMANNIAN FC LAYERS: MANIFESTATIONS IN HYPERBOLIC SPACES

As Riemannian computations over the hyperbolic space are much simpler than the matrix manifold, Thm. 4.2 can manifest in a plug-in-manner. This subsection introduces the concrete formulations.

The origin of the Poincaré ball is defined as the zero vector $\mathbf{0}$, as it is the identity element in the gyrovector space. Besides, due to the gyro structure of the Poincaré ball, Thm. 4.2 under this geometry can be further simplified.

**Theorem J.1** (RiemFC-P layer). [↓] *Given* $x \in \mathbb{P}^n_K$, *the Riemannian FC transformation* $\mathcal{F}(\cdot)$ : $\mathbb{P}^n_K \to \mathbb{P}^m_K$ *is*

$$y == \mathrm{Exp}_{\mathbf{0}}\left( \sum_{i=1}^m \left( \langle \mathrm{Log}_{\mathbf{0}}(-p_i \oplus_K x), z_i \rangle e_i \right) \right) \tag{82}$$

*where* $p_i = \mathrm{Exp}_{\mathbf{0}}(\gamma_i[z_i])$. *Here,* $\{\gamma_i \in \mathbb{R}\}_{i=1}^m$ *and* $\{z_i \in \mathbb{R}^n\}_{i=1}^m$ *are the FC parameters. Each* $e_i \in \mathbb{R}^m$ *is a vector with its* $i$-th *element equal to 1 and all other elements equal to 0. The Riemannian exponentiation and logarithm at* $\mathbf{0}$ *are*

$$\mathrm{Exp}_{\mathbf{0}}(v) = \tanh(\sqrt{\|K\|}\|v\|) \frac{v}{\sqrt{\|K\|}\|v\|}, \quad \forall v \in T_{\mathbf{0}}\mathbb{P}^n_K, \tag{83}$$

$$\mathrm{Log}_{\mathbf{0}}(y) = \tanh^{-1}(\sqrt{\|K\|}\|y\|) \frac{y}{\sqrt{\|K\|}\|y\|}, \quad \forall y \in \mathbb{P}^n_K. \tag{84}$$

**Theorem J.2** (RiemFC-H FC layer). [↓] *Following the notation of Thm. J.1, the Riemannian FC transformation* $\mathcal{F}(\cdot) : \mathbb{H}^n_K \to \mathbb{H}^m_K$ *for the input* $x \in \mathbb{H}^n_K$ *is*

$$y = \mathrm{Exp}_e\left( (0, v_1(x), \cdots, v_m(x))^\top \right) \tag{85}$$

*where* $e = \left( \frac{1}{\sqrt{\|K\|}}, 0\cdots, 0 \right)^\top$, $v_i(x) = \left\langle \mathrm{Log}_{p_i}(x), \Gamma_{e \to p_i}(z_i) \right\rangle$, *and* $p_i = \mathrm{Exp}_e(\gamma_i[(0, z_i^\top)^\top])$. *Here,* $\gamma_i \in \mathbb{R}$ *and* $z_i \in \mathbb{R}^n$ *are parameters for* $i = 1, \cdots, m$.

## J.3 EXPERIMENTS

We validate our hyperbolic FC layers on three graph datasets for the link prediction task, including the Cora (Sen et al., 2008), Disease (Anderson & May, 1991), and Airport (Zhang & Chen, 2018) datasets. We also compared our hyperbolic FC layer with the transformation layer in HNN (Ganea et al., 2018, Sec. 3.2) and HNN++ (Shimizu et al., 2020, Sec. 3.2), named Möbius transformation and the hyperbolic Poincaré FC layer, which are all based on the Poincaré model.

### J.3.1 DATASETS

**Cora.** It is a citation network where nodes represent scientific papers in the area of machine learning, edges are citations between them, and node labels are academic (sub)areas.

**Disease.** It represents a disease propagation tree, simulating the SIR disease transmission model, with each node representing either an infection or a non-infection state.

**Airport.** It is a transductive dataset where nodes represent airports and edges represent the airline routes as from OpenFlights.org.

### J.3.2 IMPLEMENTATION DETAILS

We follow the official implementations of HNN[11], and HNN++[12] to conduct the experiments. We follow the settings as HGCN[13] (Chami et al., 2019) for the link prediction task. Specifically, the baseline encoder consists of two transformation layers: the first maps the input feature dimension to 16, and the second maps 16 to 16. The transformation layers could be our hyperbolic FC layer or the ones in HNN and HNN++. We use the Adam optimizer (Kingma, 2014), with a learning rate of $1e^{-2}$. We fine-tune each model w.r.t. dropout of transformation weight and weight decay.

### J.3.3 RESULTS

Table 12: Comparison of different transformation layers on link prediction task. The graph hyperbolicity is denoted as $\delta$ (lower is more hyperbolic).

| Method | Geometry | Disease $\delta = 0$ | Airport $\delta = 1$ | Cora $\delta = 11$ |
|---|---|---|---|---|
| Möbius | Poincaré Ball | 75.1 ± 0.3 | 90.8 ± 0.2 | 89.0 ± 0.1 |
| Poincaré FC | Poincaré Ball | 77.8 ± 1.4 | **94.0 ± 0.4** | 88.1 ± 0.3 |
| RiemFC-P | Poincaré Ball | **79.2 ± 1.2** | 93.1 ± 0.7 | 89.2 ± 0.6 |
| RiemFC-H | Hyperboloid | 71.2 ± 0.6 | 84.3 ± 1.7 | **92.8 ± 0.4** |

Tab. 12 presents the 5-fold average AUC results across three datasets, revealing the following key insights:

- **Effectiveness:** Our RiemFC achieves either superior or comparable performance to the prior Möbius and Poincaré transformations.

- **Hyperbolicity & Riemannian transformation:** On datasets with high hyperbolicity, RiemFC, and Poincaré FC transformations consistently outperform Möbius transformations. Conversely, on the Cora dataset with the lowest hyperbolicity, all three Poincaré transformations perform similarly. This suggests that for highly hyperbolic data, intrinsic Riemannian transformations are more effective, as tangent Möbius transformations may distort the geometry.

- **Metric & representation power:** On the dataset with the lowest hyperbolicity, hyperboloid-based RiemFC outperforms other Poincaré-based layers, highlighting the importance of the underlying metric in Riemannian networks. Unlike the prior Poincaré FC layer, which is designed specifically for the Poincaré ball model, our Riemannian FC layer in Thm. 4.2 can adapt to various metrics in a plug-and-play manner. This adaptability enhances the representation power of HNNs, making them more versatile for diverse applications.

## K PROOFS

### K.1 PROOF OF THM. 4.2

*Proof.* By Thm. 3.1, the Riemannian signed distance from a point $Y \in \mathcal{M}$ to a Riemannian hyperplane over $\mathcal{M}$ is

$$\bar{\mathrm{d}}(Y, \widetilde{H}_{A,P}) = \frac{\langle \mathrm{Log}_P^{\mathcal{M}} Y, A \rangle_P^{\mathcal{M}}}{\|A\|_P^{\mathcal{M}}}, \tag{86}$$

---

[11] https://github.com/dalab/hyperbolic_nn
[12] https://github.com/mil-tokyo/hyperbolic_nn_plusplus
[13] https://github.com/HazyResearch/hgcn

where $\widetilde{H}_{A,P}$ is a Riemannian hyperplane parameterized by $P \in \mathcal{M}$ and $A \in T_P\mathcal{M}$. Therefore, the signed distance from $Y$ to $\widetilde{H}_{B_i,E}$ is

$$
\begin{aligned}
\widetilde{\mathrm{d}}(Y, \widetilde{H}_{B_i,E}) &= \frac{\langle \mathrm{Log}_E^{\mathcal{M}}(Y), B_i \rangle_E^{\mathcal{M}}}{\|B_i\|_E^{\mathcal{M}}} \\
&\stackrel{(1)}{=} \langle \mathrm{Log}_E^{\mathcal{M}}(Y), B_i \rangle_E^{\mathcal{M}}
\end{aligned}
\tag{87}
$$

where (1) comes from the orthonormality of $B_i$.

Setting Eq. (87) equal to $v_i(X)$, we have

$$
\langle \mathrm{Log}_E^{\mathcal{M}}(Y), B_i \rangle_E^{\mathcal{M}} = \langle \mathrm{Log}_{P_i}^{\mathcal{N}}(X), A_i \rangle_{P_i}^{\mathcal{N}}.
\tag{88}
$$

The above equation indicates

$$
\mathrm{Log}_E^{\mathcal{M}}(Y) = \sum_{i=1}^m \left( \langle \mathrm{Log}_{P_i}^{\mathcal{N}}(X), A_i \rangle_{P_i}^{\mathcal{N}} B_i \right).
\tag{89}
$$

$\square$

## K.2 PROOF OF PROP. 4.4

*Proof.* Given the FC parameters $\{p_i \in \mathbb{R}^n\}_{i=1}^m$ and $\{a_i \in \mathbb{R}^n\}_{i=1}^m$, and input vector $x \in \mathbb{R}^n$, Eq. (12) becomes

$$
\begin{aligned}
Y &\stackrel{(1)}{=} \mathrm{Exp}_0 \left( \sum_{i=1}^m \left( \langle \mathrm{Log}_{p_i}(x), a_i \rangle_{p_i} e_i \right) \right) \\
&\stackrel{(2)}{=} \sum_{i=1}^m \left( \langle x - p_i, a_i \rangle e_i \right),
\end{aligned}
\tag{90}
$$

The above comes from the following.

(1) The standard orthonormal bases over the standard inner product space $T_0\mathbb{R}^m \cong \mathbb{R}^m$ are $\{e_i\}_{i=1}^m$, with the $k-$th element defined as

$$
(e_i)_k = \begin{cases} 1 & \text{if } k = i \\ 0 & \text{otherwise.} \end{cases}
\tag{91}
$$

(2) $\mathrm{Exp}_0(x) = x$, $\langle \cdot, \cdot \rangle_{p_i} = \langle \cdot, \cdot \rangle$, and $\mathrm{Log}_{p_i}(x) = x - p_i$.

$\square$

## K.3 PROOF OF THM. 5.1

*Proof.* In the following proof, we first present the expressions of several operators under different metrics, including $v_{ij}(S)$, standard orthonormal bases, and Riemannian exponentiation at the origin. Then, we begin to prove the theorem. In this proof, we follow all the notations as the theorem.

$v_{ij}(S)$ **under different metrics:** The expressions are implied by Chen et al. (2024c, Thm. 4.2):

$$
\mathrm{LEM}: \langle \log(S) - \log(P_{ij}), Z_{ij} \rangle^{(\alpha,\beta)},
\tag{92}
$$

$$
\mathrm{AIM}: \left\langle \log(P_{ij}^{-\frac{1}{2}} S P_{ij}^{-\frac{1}{2}}), Z_{ij} \right\rangle^{(\alpha,\beta)},
\tag{93}
$$

$$
\mathrm{PEM}: \frac{1}{\theta} \left\langle S^\theta - P_{ij}^\theta, Z_{ij} \right\rangle^{(\alpha,\beta)},
\tag{94}
$$

$$
\mathrm{LCM}: \left\langle \lfloor K \rfloor - \lfloor L_{ij} \rfloor + \mathrm{Dlog}(\mathbb{K}\mathbb{L}_{ij}^{-1}), \lfloor Z_{ij} \rfloor + \frac{1}{2}\mathbb{Z}_{ij} \right\rangle,
\tag{95}
$$

$$
\mathrm{BWM}: \frac{1}{2} \left\langle (P_{ij}S)^{\frac{1}{2}} + (SP_{ij})^{\frac{1}{2}} - 2P_{ij}, \mathcal{L}_{P_{ij}}(L_{ij}Z_{ij}L_{ij}^\top) \right\rangle.
\tag{96}
$$

**Standard orthonormal bases:** Next, we show the standard orthonormal bases over $T_I \mathcal{S}^n_{++}$ under different metrics. As indicated by Tabs. 6 and 7, the inner products for any $V, W \in T_I \mathcal{S}^n_{++}$ are

$$\text{LEM, AIM, and PEM} : \langle V, W \rangle^{(\alpha, \beta)}, \tag{97}$$

$$\text{LCM} : \langle \lfloor V \rfloor + \frac{1}{2}\mathbb{V}, \lfloor W \rfloor + \frac{1}{2}\mathbb{W} \rangle, \tag{98}$$

$$\text{BWM} : \frac{1}{4} \langle V, W \rangle \tag{99}$$

The above comes from the following.

(1) Eq. (97) comes from $\log_{*, I}(V) = V$ and $\mathrm{P}_{\theta*, I}(V) = \theta V$;

(2) Eq. (98) comes from $\mathrm{Chol}_{*, I}(V) = \lfloor V \rfloor + \frac{1}{2}\mathbb{V}$;

(3) Eq. (99) comes from $\mathcal{L}_I[V] = \frac{1}{2}V$.

As shown by Thanwerdas & Pennec (2023, Thm.2.1), $F_{\sqrt{\alpha+n\beta},\sqrt{\alpha}} : \{\mathcal{S}^n, \langle \cdot, \cdot \rangle^{(\alpha,\beta)}\} \to \{\mathcal{S}^n, \langle \cdot, \cdot \rangle\}$ is the linear isometry pulling the standard inner product back to the $\mathrm{O}(n)$-invariant one:

$$F_{\sqrt{\alpha+n\beta},\sqrt{\alpha}}(X) = \sqrt{\alpha}X + \frac{\sqrt{\alpha+n\beta} - \sqrt{\alpha}}{n}\operatorname{tr}(X)I_n, \forall X \in \mathcal{S}^n. \tag{100}$$

Given any $Y \in \mathcal{S}^n$, its inverse map is

$$
\begin{aligned}
\left(F_{\sqrt{\alpha+n\beta},\sqrt{\alpha}}\right)^{-1}(Y) &= \frac{1}{\sqrt{\alpha}}\left\{Y - \left(\frac{\sqrt{1 + n\frac{\beta}{\alpha}} - 1}{n}\frac{1}{\sqrt{1 + n\frac{\beta}{\alpha}}}\right)\operatorname{tr}(Y)I\right\} \\
&= \frac{1}{\sqrt{\alpha}}\left\{Y - \frac{1}{n}\left(1 - \frac{1}{\sqrt{1 + n\frac{\beta}{\alpha}}}\right)\operatorname{tr}(Y)I\right\} \\
&= \frac{1}{\sqrt{\alpha}}Y - \frac{1}{n}\left(\frac{1}{\sqrt{\alpha}} - \frac{1}{\sqrt{\alpha+n\beta}}\right)\operatorname{tr}(Y)I.
\end{aligned} \tag{101}
$$

The standard orthonormal bases over the Euclidean spaces $\{\mathcal{S}^n, \langle \cdot, \cdot \rangle\}$ and $\{\mathcal{L}^n, \langle \cdot, \cdot \rangle\}$ are

$$\{\mathcal{S}^n, \langle \cdot, \cdot \rangle\} : U^{\mathrm{sym}}_{ij} = \begin{cases} E_{ii}, & \text{if } i = j, \\ \frac{E_{ij} + E_{ji}}{\sqrt{2}}, & \text{if } i > j. \end{cases} \tag{102}$$

$$\{\mathcal{L}^n, \langle \cdot, \cdot \rangle\} : U^{\mathrm{tril}}_{ij} = E_{ij}, \forall i \geq j \tag{103}$$

where $i \geq j, i, j = 1, \cdots, n$, and $\{E_{ij}\}^n_{i,j=1}$ are standard basis matrices, with the $(k, l)$ element defined as

$$(E_{ij})_{kl} = \begin{cases} 1 & \text{if } k = i \text{ and } l = j, \\ 0 & \text{otherwise.} \end{cases} \tag{104}$$

The standard orthonormal bases w.r.t. Eqs. (97) to (99) are

$$\text{LEM, AIM, PEM} : U^{(\alpha,\beta)}_{ij} \stackrel{(1)}{=} \begin{cases} \frac{1}{\sqrt{\alpha}}E_{ii} - \frac{1}{n}\left(\frac{1}{\sqrt{\alpha}} - \frac{1}{\sqrt{\alpha+n\beta}}\right)I, & \text{if } i = j, \\ \frac{E_{ij} + E_{ji}}{\sqrt{2\alpha}}, & \text{if } i > j. \end{cases} \tag{105}$$

$$\text{LCM} : U^{\mathrm{LC}}_{ij} \stackrel{(2)}{=} \begin{cases} 2E_{ii}, & \text{if } i = j, \\ E_{ij}, & \text{if } i > j. \end{cases} \tag{106}$$

$$\text{BWM} : U^{\mathrm{BW}}_{ij} \stackrel{(3)}{=} \begin{cases} 2E_{ii}, & \text{if } i = j, \\ \sqrt{2}(E_{ij} + E_{ji}), & \text{if } i > j. \end{cases} \tag{107}$$

Here, $i \geq j, i, j = 1, \cdots, n$. The above comes from the following.

(1) $U^{(\alpha,\beta)}_{ij} = \left(F_{\sqrt{\alpha+n\beta},\sqrt{\alpha}}\right)^{-1}(U^{\mathrm{sym}}_{ij})$, with $F_{\sqrt{\alpha+n\beta},\sqrt{\alpha}} : \mathcal{S}^n \to \mathcal{S}^n$ as the linear isometry pulling back the Frobenius inner product to the $\mathrm{O}(n)$-invariant inner product;

(2) $f^{\mathrm{LC}}(V) = \lfloor V \rfloor + \frac{1}{2}\mathbb{V} : \mathcal{L}^n \to \mathcal{L}^n$ is the linear isometry pulling the Frobenius inner product to Eq. (98);

(3) $f^{\mathrm{BW}}(V) = \frac{1}{2}V : \mathcal{S}^n \to \mathcal{S}^n$ is the linear isometry pulling the Frobenius inner product back to Eq. (99);

**Riemannian exponentiation:** Next, we show $\mathrm{Exp}_I$ under different metrics

$$\text{LEM and AIM} : \mathrm{Exp}_I(V) \overset{(1)}{=} \exp(V), \tag{108}$$

$$\text{PEM} : \mathrm{Exp}_I(V) \overset{(2)}{=} (I + \theta V)^{\frac{1}{\theta}}, \tag{109}$$

$$\text{LCM} : \mathrm{Exp}_I(V) \overset{(3)}{=} \left( \lfloor V \rfloor + \mathrm{Dexp}\left(\frac{1}{2}\mathbb{V}\right) \right) \left( \lfloor V \rfloor + \mathrm{Dexp}\left(\frac{1}{2}\mathbb{V}\right) \right)^{\top}, \tag{110}$$

$$\text{BWM} : \mathrm{Exp}_I(V) \overset{(4)}{=} I + V + \frac{1}{4}V^2 = \left( I + \frac{1}{2}V \right)^2, \tag{111}$$

The above comes from the following.

(1) $\log_{*,I}(V) = V$ and $\log I = \mathbf{0}$;

(2) $\mathrm{P}_{\theta*,I}(V) = \theta V$;

(3) $\mathrm{Chol}_{*,I}(V) = \lfloor V \rfloor + \frac{1}{2}\mathbb{V}$;

(4) $\mathcal{L}_I[V] = \frac{1}{2}V$.

Now, we can prove the results metric by metric.

**LEM:**

$$\mathrm{Exp}_I\left( \sum_{i,j=1,i\geq j}^{m} v_{ij}^{\mathrm{LE}}(S)U_{ij}^{(\alpha,\beta)} \right)$$

$$= \exp\left( \sum_{i,j=1,i\geq j}^{m} \left( \log(S) - \log(P_{ij}), Z_{ij}\rangle^{(\alpha,\beta)}U_{ij}^{(\alpha,\beta)} \right) \right). \tag{112}$$

**AIM:**

$$\mathrm{Exp}_I\left( \sum_{i,j=1,i\geq j}^{m} v_{ij}^{\mathrm{AI}}(S)U_{ij}^{(\alpha,\beta)} \right)$$

$$= \exp\left( \sum_{i,j=1,i\geq j}^{m} \left( \langle \log(P_{ij}^{-\frac{1}{2}}SP_{ij}^{-\frac{1}{2}}), Z_{ij}\rangle^{(\alpha,\beta)}U_{ij}^{(\alpha,\beta)} \right) \right). \tag{113}$$

**PEM:**

$$\mathrm{Exp}_I\left( \sum_{i,j=1,i\geq j}^{m} v_{ij}^{\mathrm{PE}}(S)U_{ij}^{(\alpha,\beta)} \right)$$

$$= \left( I + \theta \sum_{i,j=1,i\geq j}^{m} \left( \frac{1}{\theta}\langle S^\theta - P_{ij}^\theta, Z_{ij}\rangle^{(\alpha,\beta)}U_{ij}^{(\alpha,\beta)} \right) \right)^{\frac{1}{\theta}} \tag{114}$$

$$= \left( I + \sum_{i,j=1,i\geq j}^{m} \left( \langle S^\theta - P_{ij}^\theta, Z_{ij}\rangle^{(\alpha,\beta)}U_{ij}^{(\alpha,\beta)} \right) \right)^{\frac{1}{\theta}}.$$

**LCM:**

$$\mathrm{Exp}_I\left(\sum_{i,j=1,i\geq j}^m v_{ij}^{\mathrm{LC}}(S)U_{ij}^{\mathrm{LC}}\right)$$

$$= \left(\lfloor V^{\mathrm{LC}}\rfloor + \mathrm{Dexp}\left(\frac{1}{2}\mathbb{V}^{\mathrm{LC}}\right)\right)\left(\lfloor V^{\mathrm{LC}}\rfloor + \mathrm{Dexp}\left(\frac{1}{2}\mathbb{V}^{\mathrm{LC}}\right)\right)^\top, \tag{115}$$

with

$$V^{\mathrm{LC}} = \sum_{i,j=1,i\geq j}^m v_{ij}^{\mathrm{LC}}(S)U_{ij}^{\mathrm{LC}}$$

$$= \sum_{i,j=1,i\geq j}^m \left(\left\langle \lfloor K\rfloor - \lfloor L_{ij}\rfloor + \mathrm{Dlog}(\mathbb{K}\mathbb{L}_{ij}^{-1}), \lfloor Z_{ij}\rfloor + \frac{1}{2}\mathbb{Z}_{ij}\right\rangle\right)U_{ij}^{\mathrm{LC}} \tag{116}$$

**BWM:**

$$\mathrm{Exp}_I\left(\sum_{i,j=1,i\geq j}^m v_{ij}^{\mathrm{BW}}(S)U_{ij}^{\mathrm{BW}}\right)$$

$$= \left(I + \frac{1}{2}V^{\mathrm{BW}}\right)^2, \tag{117}$$

with $V^{\mathrm{BW}}$ defined as

$$V^{\mathrm{BW}} = \sum_{i,j=1,i\geq j}^m \left\{\frac{1}{2}\left\langle (P_{ij}S)^{\frac{1}{2}} + (SP_{ij})^{\frac{1}{2}} - 2P_{ij}, \mathcal{L}_{P_{ij}}(L_{ij}Z_{ij}L_{ij}^\top)\right\rangle U_{ij}^{\mathrm{BW}}\right\}. \tag{118}$$

$\square$

### K.4 PROOF OF PROP. 5.2

We begin by recalling two vector structures on the SPD manifold. Next, we identify the expression for the linear homomorphisms. Finally, we present our proof.

We define a map $\phi(\cdot) : \mathcal{S}_{++}^n \to \mathcal{L}^n$ as

$$\phi(S) = \lfloor L\rfloor + \mathrm{Dlog}(\mathbb{L}), \tag{119}$$

where $P = LL^\top$ is the Cholesky decomposition. For any $P, Q \in \mathcal{S}_{++}^n$ and $t \in \mathbb{R}$, the vector structures over the SPD manifold are defined as

$$P \oplus^{\mathrm{LE}} Q = \exp(\log(P) + \log(Q)) \tag{120}$$

$$t \odot^{\mathrm{LE}} P = \exp(t\log(P)) = P^t \tag{121}$$

$$P \oplus^{\mathrm{LC}} Q = \phi^{-1}(\phi(P) + \phi(Q)) \tag{122}$$

$$t \odot^{\mathrm{LC}} P = \phi^{-1}(t\phi(P)) = P^t \tag{123}$$

As shown by Arsigny et al. (2005); Chen et al. (2024d), $\{\mathcal{S}_{++}^n, \oplus^{\mathrm{LE}}, \odot^{\mathrm{LE}}\}$ and $\{\mathcal{S}_{++}^n, \oplus^{\mathrm{LC}}, \odot^{\mathrm{LC}}\}$ forms vector spaces. We further present the associated linear homomorphisms.

**Lemma K.1** (SPD Homomorphisms). *Given any homomorphisms*

$$\zeta^{\mathrm{LE}}(\cdot) : \{\mathcal{S}_{++}^n, \oplus^{\mathrm{LE}}, \odot^{\mathrm{LE}}\} \to \{\mathcal{S}_{++}^m, \oplus^{\mathrm{LE}}, \odot^{\mathrm{LE}}\}, \tag{124}$$

$$\zeta^{\mathrm{LC}}(\cdot) : \{\mathcal{S}_{++}^n, \oplus^{\mathrm{LC}}, \odot^{\mathrm{LC}}\} \to \{\mathcal{S}_{++}^m, \oplus^{\mathrm{LC}}, \odot^{\mathrm{LC}}\}, \tag{125}$$

*they can be expressed as*

$$\zeta^{\mathrm{LE}} = \exp \circ g \circ \log, \tag{126}$$

$$\zeta^{\mathrm{LC}} = \phi^{-1} \circ f \circ \phi, \tag{127}$$

*where $f : \mathcal{L}^n \to \mathcal{L}^m$ and $g : \mathcal{S}^n \to \mathcal{S}^m$ are linear homomorphisms over the Euclidean space $\mathcal{L}^n$ and $\mathcal{S}^n$, respectively.*

*Proof.* As shown by Chen et al. (2024d), $\log(\cdot)$ is the linear isomorphism from $\{\mathcal{S}^n_{++}, \oplus^{\mathrm{LE}}, \odot^{\mathrm{LE}}\}$ to the Euclidean space $\mathcal{S}^n$ and $\phi$ is the linear isomorphism from $\{\mathcal{S}^n_{++}, \oplus^{\mathrm{LC}}, \odot^{\mathrm{LC}}\}$ to the Euclidean space $\mathcal{L}^n$. Therefore, any linear homomorphisms over these two linear spaces have the following forms:

$$\zeta^{\mathrm{LE}} = \log^{-1} f \circ \log, \tag{128}$$

$$\zeta^{\mathrm{LC}} = \phi^{-1} g \circ \phi, \tag{129}$$

where $f : \mathcal{S}^n \to \mathcal{S}^m$ and $g : \mathcal{L}^n \to \mathcal{L}^m$ are linear homomorphisms over the Euclidean space $\mathcal{S}^n$ and $\mathcal{L}^n$, respectively. $\qquad\square$

With all the above theoretical preparation, we begin to present our proof.

*Proof.* Given an SPD matrix $S \in \mathcal{S}^n_{++}$, Eq. (128) can be rewritten as

$$\zeta^{\mathrm{LE}}(S) \overset{(1)}{=} \exp\left( \sum_{i,j=1,i\geq j}^m \langle \log(S), A_{ij} \rangle U_{ij}^{\mathrm{sym}} \right)$$

$$\overset{(2)}{=} \exp\left( \sum_{i,j=1,i\geq j}^m \langle \log(S), A_{ij} \rangle U_{ij}^{(1,0)} \right) \tag{130}$$

$$\overset{(3)}{=} \mathcal{F}^{\mathrm{LE}}(S; \mathbf{A}, \mathbf{I})$$

where $\mathbf{A} = \{A_{ij} \in \mathcal{S}^n\}_{i,j=1,i\geq j}^m$ and $\mathbf{I} = \{I, \cdots, I\}$. The above comes from the following.

(1) The linear map $f$ can be represented by $\{A_{ij} \in \mathcal{S}^n\}_{i,j=1,i\geq j}^m$ under the bases $\{U_{ij}^{\mathrm{sym}}\}_{i,j=1,i\geq j}^n$ over $\mathcal{S}^n$ and $\{U_{ij}^{\mathrm{sym}}\}_{i,j=1,i\geq j}^m$ over $\mathcal{S}^m$;

(2) $\{U_{ij}^{\mathrm{sym}}\}_{i,j=1,i\geq j}^m = \{U_{ij}^{(1,0)}\}_{i,j=1,i\geq j}^m$;

(3) $\mathrm{Exp}_I = \exp$ under LEM.

Following the above logic, we have the following for $\{\mathcal{S}^n_{++}, \oplus^{\mathrm{LC}}, \odot^{\mathrm{LC}}\}$:

$$\zeta^{\mathrm{LC}}(S) \overset{(1)}{=} \phi^{-1}\left( \sum_{i,j=1,i\geq j}^m \langle \phi(S), A_{ij} \rangle U_{ij}^{\mathrm{tril}} \right) \tag{131}$$

$$\overset{(2)}{=} \mathcal{F}^{\mathrm{LC}}(S; \mathbf{Z}, \mathbf{I}),$$

where $A_{ij} \in \mathcal{L}^n$ for $i, j = 1, \cdots, m, i \geq j$, $\mathbf{Z} = \{Z_{ij} = A_{ij} + \mathbb{D}(A_{ij}) \in \mathcal{L}^n\}_{i,j=1,i\geq j}^m$ and $\mathbf{I} = \{I, \cdots, I\}$. The above comes from the following.

(1) The linear map $g$ can be represented by $\{A_{ij}\}_{i,j=1,i\geq j}^m$;

(2) Eqs. (20) and (25).

$\qquad\square$

### K.5 PROOF OF THM. 6.1

Before presenting our proof, we first discuss some basic facts about the ONB Grassmannian FC layer.

As implied by Eq. (38), any tangent vector $V \in T_{I_{p,n}} \mathrm{Gr}(p, n)$ can be expressed as

$$V = \begin{pmatrix} \mathbf{0} \\ I_{n-p} \end{pmatrix} B_V = \begin{pmatrix} \mathbf{0} \\ B_V \end{pmatrix}, \text{ with } B_V \in \mathbb{R}^{(n-p)\times p}. \tag{132}$$

According to Thm. 4.2 and Eq. (132), the ONB Grassmannian FC layer $\mathcal{F}(\cdot) : \mathrm{Gr}(p, n) \to \mathrm{Gr}(q, m)$ has the following form:

$$Y = \mathrm{Exp}_{I_{q,m}}\left( \sum_{\substack{i=1,\cdots,m-q \\ j=1,\cdots,m}} \left( \langle \mathrm{Log}_{P_{ij}}(X), A_{ij} \rangle_{P_{ij}} U_{ij} \right) \right), \tag{133}$$

where $\{U_{ij}\}$ are the orthonormal bases over $T_{I_{q,m}}\mathrm{Gr}(q,m)$. As discussed in Sec. 4.3, we model the FC parameters by parallel transport and Riemannian exponential map:

$$A_{ij} = \Gamma_{I_{p,n}\to P_{ij}}(Z_{ij}), \tag{134}$$

$$P_{ij} = \mathrm{Exp}_{I_{p,n}}(\gamma_{ij}[Z_{ij}]), \tag{135}$$

where $Z_{ij} = \begin{pmatrix} \mathbf{0} \\ B_{Z_{ij}} \end{pmatrix} \in T_{I_{p,n}}\mathrm{Gr}(p,n)$. Therefore, we can model each $P_{ij}$ and $A_{ij}$ by $B_{Z_{ij}} \in \mathbb{R}^{(n-p)\times p}$ and $\gamma_{ij} \in \mathbb{R}$. With the above ingredient, we present the proof in the following.

*Proof.* **The standard orthonormal basis:** As the inner product over $T_{I_{q,m}}\mathrm{Gr}(q,m)$ is the Frobenius matrix inner product (Bendokat et al., 2024, Eq. 3.2), the standard orthonormal basis over $T_{I_{q,m}}\mathrm{Gr}(q,m)$ is

$$U_{ij} = \begin{pmatrix} \mathbf{0} \\ E_{ij} \end{pmatrix}, 1 \le i \le m-q \wedge 1 \le j \le q, \tag{136}$$

where $\{E_{ij}\}$ are standard basis matrices over $\mathbb{R}^{(m-q)\times q}$

**The Riemannian exponential map at the origin:** The SVD of $V \in T_{I_{p,n}}\mathrm{Gr}(p,n)$ can be calculated via the SVD of $B_V$:

$$V = \begin{pmatrix} \mathbf{0} \\ B_V \end{pmatrix} = \begin{pmatrix} \mathbf{0} \\ O \end{pmatrix}\Sigma R^\top = \begin{pmatrix} \mathbf{0} \\ O\Sigma R^\top \end{pmatrix}, \tag{137}$$

where $B_V \overset{\mathrm{SVD}}{:=} O\Sigma R^\top$. Therefore, the Riemannian exponential map at $I_{p,n}$ can be simplified as

$$\begin{aligned}
\mathrm{Exp}_{I_{p,n}}(V) &= \begin{pmatrix} I_p \\ \mathbf{0} \end{pmatrix} R\cos(\Sigma)R^T + \begin{pmatrix} \mathbf{0} \\ O \end{pmatrix}\sin(\Sigma)R^T \\
&= \begin{pmatrix} R\cos(\Sigma)R^T \\ O\sin(\Sigma)R^T \end{pmatrix}
\end{aligned} \tag{138}$$

$v_{ij}(U)$ **under the ONB perspective:** The ONB parallel transport can be further simplified. Given $P \in \mathrm{Gr}(p,n)$, we have the following for the Riemannian logarithm

$$\mathrm{Log}_{I_{p,n}}(P) = \begin{pmatrix} \mathbf{0} \\ B_P \end{pmatrix} \overset{\mathrm{SVD}}{:=} \begin{pmatrix} \mathbf{0} \\ O_P\Sigma_P R_P^\top \end{pmatrix}, \tag{139}$$

with $B_P \overset{\mathrm{SVD}}{:=} O_P\Sigma_P R_P^\top$. For $P \in \mathrm{Gr}(p,n)$ and $Z \in T_{I_{p,n}}\mathrm{Gr}(p,n)$, the parallel transport can be further simplified:

$$\Gamma_{I_{p,n}\to P}(Z)$$

$$= \left(\left(\begin{pmatrix} I_{p,n}R_P & \begin{pmatrix} \mathbf{0} \\ O_P \end{pmatrix} \end{pmatrix}\right)\begin{pmatrix} -\sin(\Sigma_P) \\ \cos(\Sigma_P) \end{pmatrix}\begin{pmatrix} \mathbf{0} \\ O_P \end{pmatrix}^T + \left(I - \begin{pmatrix} \mathbf{0} \\ O_P \end{pmatrix}\begin{pmatrix} \mathbf{0} \\ O_P \end{pmatrix}^T\right)\right)Z$$

$$= \left(\left(-\begin{pmatrix} I_p \\ \mathbf{0} \end{pmatrix}R_P\sin(\Sigma_P) + \begin{pmatrix} \mathbf{0} \\ O_P \end{pmatrix}\cos(\Sigma_P)\right)\begin{pmatrix} \mathbf{0} \\ O_P \end{pmatrix}^T + \begin{pmatrix} I_p & \mathbf{0} \\ \mathbf{0} & I_{n-p}-O_PO_P^\top \end{pmatrix}\right)Z$$

$$= \left(\begin{pmatrix} -R_P\sin(\Sigma_P) \\ O_P\cos(\Sigma_P) \end{pmatrix}\begin{pmatrix} \mathbf{0} & O_P^\top \end{pmatrix} + \begin{pmatrix} I_p & \mathbf{0} \\ \mathbf{0} & I_{n-p}-O_PO_P^\top \end{pmatrix}\right)Z$$

$$= \left(\begin{pmatrix} \mathbf{0} & -R_P\sin(\Sigma_P)O_P^\top \\ \mathbf{0} & O_P\cos(\Sigma_P)O_P^\top \end{pmatrix} + \begin{pmatrix} I_p & \mathbf{0} \\ \mathbf{0} & I_{n-p}-O_PO_P^\top \end{pmatrix}\right)Z$$

$$= \begin{pmatrix} I_p & -R_P\sin(\Sigma_P)O_P^\top \\ \mathbf{0} & I_{n-p}+O_P\cos(\Sigma_P)O_P^\top - O_PO_P^\top \end{pmatrix}Z$$

$$= \begin{pmatrix} I_p & -R_P\sin(\Sigma_P)O_P^\top \\ \mathbf{0} & I_{n-p}+O_P\cos(\Sigma_P)O_P^\top - O_PO_P^\top \end{pmatrix}\begin{pmatrix} \mathbf{0} \\ B_Z \end{pmatrix}$$

$$= \begin{pmatrix} -R_P\sin(\Sigma_P)O_P^\top B_Z \\ \left(O_P\cos(\Sigma_P)O_P^\top + I_{n-p} - O_PO_P^\top\right)B_Z \end{pmatrix}.$$

Combining all the above results, one can directly obtain the results. $\qquad\square$

### K.6    PROOF OF THM. 6.2

*Proof.* Firstly, $v_{ij}(X)$ over the Grassmannian $\widetilde{\mathrm{Gr}}(p,n)$ takes the following form:

$$
\begin{aligned}
v_{ij}(X) &= \left\langle \mathrm{Log}_{P_{ij}}(X), \Gamma_{\widetilde{I}_{p,n} \to P_{ij}}(Z_{ij}) \right\rangle_{P_{ij}} \\
&\overset{(1)}{=} \frac{1}{2} \left\langle \mathrm{Log}_{P_{ij}}(X), \Gamma_{\widetilde{I}_{p,n} \to P_{ij}}(Z_{ij}) \right\rangle
\end{aligned}
\tag{140}
$$

where (1) comes from Tab. 8. Here, each $Z_{ij} \in T_{\widetilde{I}_{p,n}} \widetilde{\mathrm{Gr}}(p,n)$ and $P_{ij} \in \widetilde{\mathrm{Gr}}(p,n)$.

**Riemannian logarithm.** As shown by Nguyen et al. (2024, Prop. 3.12), the PP Grassmannian logarithm can be calculated by the ONB logarithm:

$$
\mathrm{Log}_P^{\mathrm{PP}}(X) = \pi_{*,\pi(P)}\left(\mathrm{Log}_{\pi^{-1}(P)}^{\mathrm{ONB}}(\pi^{-1}(X))\right),
\tag{141}
$$

where $\pi(U) = UU^\top : \mathrm{Gr}(p,n) \to \widetilde{\mathrm{Gr}}(p,n)$ is the Riemannian isometry, and $\pi_{*,U}(V) = UV^\top + VU^\top$ is the differential map for all $U \in \mathrm{Gr}(p,n)$ and $V \in T_U\mathrm{Gr}(p,n)$.

**Tangent vector and Riemannian exponential map at the identity.** As implied by Eq. (40), any tangent vector at the identity has the following form:

$$
V = \begin{pmatrix} 0 & B^T \\ B & 0 \end{pmatrix} \in T_{\widetilde{I}_{p,n}} \widetilde{\mathrm{Gr}}(p,n) \text{ with } B \in \mathbb{R}^{(n-p)\times p}.
\tag{142}
$$

The Riemannian exponential at the identity can also be simplified:

$$
\begin{aligned}
\mathrm{Exp}_{\widetilde{I}_{p,n}}(V) &= \exp([V, \widetilde{I}_{p,n}])\widetilde{I}_{p,n} \exp(-[V, \widetilde{I}_{p,n}]) \\
&= \exp\left(\begin{pmatrix} 0 & -B^T \\ B & 0 \end{pmatrix}\right) \widetilde{I}_{p,n} \exp\left(\begin{pmatrix} 0 & -B^T \\ B & 0 \end{pmatrix}\right)^\top \\
&= \left(\exp\left(\begin{pmatrix} 0 & -B^T \\ B & 0 \end{pmatrix}\right)\right)_{1:p} \left(\left(\exp\left(\begin{pmatrix} 0 & -B^T \\ B & 0 \end{pmatrix}\right)\right)_{1:p}\right)^\top
\end{aligned}
\tag{143}
$$

with $(\cdot)_{1:p}$ as the first-$p$ columns of the input square matrix.

**Parallel transport starting at the identity.** The parallel transport along geodesic from $\widetilde{I}_{p,n}$ to $P \in \widetilde{\mathrm{Gr}}(p,n)$ can also be simplified. For any $V \in T_{\widetilde{I}_{p,n}} \widetilde{\mathrm{Gr}}(p,n)$, denoting $\bar{P} = \mathrm{Log}_{\widetilde{I}_{p,n}}(P)$, we have the following:

$$
\begin{aligned}
\Gamma_{\widetilde{I}_{p,n} \to P}(V) &\overset{(1)}{=} \exp\left(\left[\bar{P}, \widetilde{I}_{p,n}\right]\right) V \exp\left(-\left[\bar{P}, \widetilde{I}_{p,n}\right]\right) \\
&\overset{(2)}{=} \exp\left(\begin{pmatrix} 0 & -B_P^T \\ B_P & 0 \end{pmatrix}\right) V \exp\left(\begin{pmatrix} 0 & -B_P^T \\ B_P & 0 \end{pmatrix}\right)^\top
\end{aligned}
\tag{144}
$$

The above derivation comes from the following.

(1) Tab. 8;

(2) $\bar{P} = \begin{pmatrix} 0 & B_P^T \\ B_P & 0 \end{pmatrix}$

**Trivialization and simplification** Combining Eqs. (140) and (142) to (144), we model each $P_{ij}$ such that

$$
P_{ij} = \exp\left(\begin{pmatrix} 0 & -B_{P_{ij}}^T \\ B_{P_{ij}} & 0 \end{pmatrix}\right) \widetilde{I}_{p,n} \exp\left(\begin{pmatrix} 0 & -B_{P_{ij}}^T \\ B_{P_{ij}} & 0 \end{pmatrix}\right)^\top
\tag{145}
$$

where $B_{P_{ij}} = \gamma_{ij}[B_{Z_{ij}}]$ with $Z_{ij} = \begin{pmatrix} 0 & B_{Z_{ij}}^T \\ B_{Z_{ij}} & 0 \end{pmatrix}$ and $B_{Z_{ij}} \in \mathbb{R}^{(n-p)\times p}$.

Denoting $O_{ij} = \exp\left(\begin{pmatrix} 0 & -B_{P_{ij}}^T \\ B_{P_{ij}} & 0 \end{pmatrix}\right)$, $v_{ij}(X)$ can be simplified as

$$
v_{ij}(X) = \frac{1}{2} \left\langle \pi_{*,\pi(P)}\left(\mathrm{Log}_{(O_{ij})_{1:p}}^{\mathrm{ONB}}(\pi^{-1}(X))\right), O_{ij}Z_{ij}O_{ij}^\top \right\rangle
\tag{146}
$$

**Orthonormal bases.** Finally, let us deal with the orthonormal bases over $T_{\widetilde{I}_{q,m}} \widetilde{\mathrm{Gr}}(q,m)$. For any tangent vector $V_1, V_2 \in T_{\widetilde{I}_{q,m}} \widetilde{\mathrm{Gr}}(q,m)$, we have the following:

$$
\begin{aligned}
\langle V_1, V_2 \rangle_{\widetilde{I}_{p,n}} &= \frac{1}{2} \langle V_1, V_2 \rangle \\
&= \frac{1}{2} \left\langle \begin{pmatrix} 0 & B_{V_1}^T \\ B_{V_1} & 0 \end{pmatrix}, \begin{pmatrix} 0 & B_{V_2}^T \\ B_{V_2} & 0 \end{pmatrix} \right\rangle \\
&= \langle B_{V_1}, B_{V_2} \rangle
\end{aligned}
\tag{147}
$$

Therefore, the orthonormal bases are

$$
U_{ij} = \begin{pmatrix} 0 & E_{ij}^\top \\ E_{ij} & 0 \end{pmatrix}, \forall i = 1, \cdots, m-q \wedge j = 1, \cdots, q
\tag{148}
$$

where $E_{ij} \in \mathbb{R}^{(m-q) \times q}$ is the standard basis matrix.

Combining Eqs. (143), (146) and (148), one can readily obtain the results. $\qquad\square$

### K.7 PROOF OF PROP. 7.1

*Proof.* By Thm. 4.2, we have the following

$$
\begin{aligned}
Y &\overset{(1)}{=} \mathrm{Exp}_E^{\mathcal{M}} \left( \sum_{i=1}^m \left( \langle \mathrm{Log}_{p_i}^{\mathrm{Euc}}(x), a_i \rangle_{p_i}^{\mathrm{Euc}} B_i \right) \right), \\
&\overset{(2)}{=} \mathrm{Exp}_E^{\mathcal{M}} \left( \sum_{i=1}^m \left( \langle x - p_i, a_i \rangle B_i \right) \right), \\
&\overset{(3)}{=} \mathrm{Exp}_E^{\mathcal{M}} \left( \sum_{i=1}^m \left( \langle x - p_i, a_i \rangle f^{-1}(e_i) \right) \right), \\
&\overset{(4)}{=} \mathrm{Exp}_E^{\mathcal{M}} \left( f^{-1} \left( \sum_{i=1}^m \langle x - p_i, a_i \rangle e_i \right) \right), \\
&\overset{(5)}{=} \mathrm{Exp}_E^{\mathcal{M}} \left( f^{-1} \left( \bar{A}x + \bar{b} \right) \right), \\
&\overset{(6)}{=} \mathrm{Exp}_E^{\mathcal{M}} \left( Ax + b \right).
\end{aligned}
\tag{149}
$$

The above comes from the following,

(1) $p_i, a_i \in \mathbb{R}^n$, and $\{B_i\}$ are the orthonormal bases over $\{T_E \mathcal{M}, g_E\}$;

(2) The Euclidean logarithm and metric become the familiar vector operation:
$$
\begin{aligned}
\mathrm{Log}_{p_i}^{\mathrm{Euc}}(x) &= x - p_i \\
\langle v, w \rangle_p^{\mathrm{Euc}} &= \langle v, w \rangle, \forall p \in \mathbb{R}^n, \forall v, w \in T_p \mathbb{R}^n;
\end{aligned}
$$

(3) $f$ is the linear isomorphism pulling the standard inner product back to $g_E$; $\{e_i\}$ are the standard orthonormal bases over the standard inner product;

(4) Linearity of $f^{-1}$;

(5) $\sum_{i=1}^m \langle x - p_i, a_i \rangle e_i$ has the form of affine transformation;

(6) As $f^{-1}$ has matrix representation, $f^{-1}(x) = \tilde{A}x$, we have
$$
\begin{aligned}
f^{-1} \left( \bar{A}x + \bar{b} \right) &= \tilde{A} \left( \bar{A}x + \bar{b} \right) \\
&= \tilde{A}\bar{A}x + \tilde{A}\bar{b}.
\end{aligned}
\tag{150}
$$

Setting $A = \tilde{A}\bar{A}$ and $b = \tilde{A}\bar{b}$, one can obtain the result.

$\qquad\square$

### K.8 PROOF OF THM. J.1

We first prove a useful lemma.

**Lemma K.2.** *We assume that the manifold $\mathcal{M}$ admits a gyrogroup (Nguyen, 2022a, Def. 2.2) defined by*[14]

$$x \oplus y = \mathrm{Exp}_x \left( \Gamma_{e \to x} \left( \mathrm{Log}_e (y) \right) \right), \forall p, q \in \mathcal{M}, \tag{151}$$

*where $e \in \mathcal{M}$ is the origin of the manifold. Then, we have the following*

$$\left\langle \mathrm{Log}_p(x), a \right\rangle_p = \left\langle \mathrm{Log}_e (\ominus p \oplus x), \Gamma_{p \to e}(a) \right\rangle_e, \quad \forall x, p \in \mathcal{M} \text{ and } \forall a \in T_p \mathcal{M}. \tag{152}$$

*Proof.* **Credit of the proof:** Eq. (151) comes from Nguyen & Yang (2023, Eq. (1)), who demonstrated that several geometries admit gyrogroups based on this definition. The prototype of Eq. (152) comes from App. I by Nguyen et al. (2024), which only deals with SPD matrices. Here, we further extend the result into general gyrogroups.

Denoting $\ominus p$ as the gyro inverse of $p$ ($\ominus p \oplus p = e$), we have

$$x \overset{(1)}{=} p \oplus (\ominus p \oplus x) \overset{(2)}{=} \mathrm{Exp}_p \left( \Gamma_{e \to p} \left( \mathrm{Log}_e (\ominus p \oplus x) \right) \right)$$

$$\overset{(3)}{\Rightarrow} \mathrm{Log}_p(x) = \Gamma_{e \to p} \left( \mathrm{Log}_e (\ominus p \oplus x) \right). \tag{153}$$

The above comes from the following,

(1) Left cancellation law of the gyrogroup (Ungar, 2022, Thms. 1.13).

(2) Definition of gyro addition.

(3) Applying both sides with $\mathrm{Log}_p(\cdot)$.

By the last equation, we have

$$\left\langle \mathrm{Log}_p(x), a \right\rangle_p = \left\langle \Gamma_{e \to p} \left( \mathrm{Log}_e (\ominus p \oplus x) \right), a \right\rangle_p$$

$$\overset{(1)}{=} \left\langle \mathrm{Log}_e (\ominus p \oplus x), \Gamma_{p \to e}(a) \right\rangle_e, \tag{154}$$

where (1) comes from

- Parallel transport preserving the norm (Do Carmo & Flaherty Francis, 1992, Sec. 3.1)

- $\Gamma_{p \to e} \circ \Gamma_{e \to p}(v) = v, \forall v \in T_e \mathcal{M}$.

$\square$

Now we begin to prove Thm. J.1.

*Proof of Thm. J.1.* The Riemannian metric at the identity element is

$$\langle v, w \rangle_{\mathbf{0}} = 4 \langle v, w \rangle, \forall v, w \in T_{\mathbf{0}} \mathbb{P}_K^m. \tag{155}$$

Obviously, $\{\frac{1}{4} e_i\}_{i=1}^m$ is an orthonormal basis.

By Lem. K.2, we have

$$\left\langle \mathrm{Log}_{p_i}(x), a_i \right\rangle_{p_i} \frac{1}{4} e_i \overset{(1)}{=} \left\langle \mathrm{Log}_{\mathbf{0}}(-p_i \oplus_K x), \Gamma_{p_i \to \mathbf{0}}(a_i) \right\rangle_{\mathbf{0}} \frac{1}{4} e_i$$

$$\overset{(2)}{=} \left\langle \mathrm{Log}_{\mathbf{0}}(-p_i \oplus_K x), \Gamma_{p_i \to \mathbf{0}}(a_i) \right\rangle e_i \tag{156}$$

$$\overset{(3)}{=} \left\langle \mathrm{Log}_{\mathbf{0}}(-p_i \oplus_K x), z_i \right\rangle e_i.$$

The above comes from the following,

(1) Lem. K.2 and $\ominus_K p = -p \forall p \in \mathbb{P}_K^n$.

(2) Eq. (155).

(3) $a_i = \Gamma_{\mathbf{0} \to p_i}(z_i)$.

$\square$

---

[14]We assume all the involved Riemannian operators are well-defined.

## K.9 PROOF OF THM. J.2

*Proof.* We only need to show the origin, the tangent space at the origin, and the inner product and an orthonormal basis over the tangent space at the origin.

The hyperboloid is isometric to the Poincaré ball by the following diffeomorphism (Lee, 2006):

$$\pi_{\mathbb{P}_K^n \to \mathbb{H}_K^n}(x) = \left( \frac{1}{\sqrt{|K|}} \frac{1 - K\|x\|^2}{1 + K\|x\|^2}; \frac{2x^T}{1 + K\|x\|^2} \right)^\top. \tag{157}$$

The origin of hyperboloid is therefore defined as

$$e := \pi_{\mathbb{P}_K^n \to \mathbb{H}_K^n}(\mathbf{0}) = \left( \frac{1}{\sqrt{|K|}}, 0 \cdots, 0 \right)^\top. \tag{158}$$

The Riemannian metric and tangent space at $e$ are

$$T_e\mathbb{H}_K^n = \{(0, v^\top)^\top | v \in \mathbb{R}^n\}, \tag{159}$$

$$\langle (0, v^\top)^\top, (0, w^\top)^\top \rangle_e = \langle v, w \rangle, \quad \forall (0, v^\top)^\top, (0, w^\top)^\top \in T_e\mathbb{H}_K^n. \tag{160}$$

Therefore, $\{(0, e_i^\top)^\top\}_{i=1}^m$ is an orthonormal basis of $T_e\mathbb{H}_K^n$ with $e_i \in \mathbb{R}^n$.

Putting the above with Tab. 11, we can manifest Thm. 4.2 in the hyperboloid geometry. $\square$

