# OpenReview forum: "Riemannian Transformation Layers for General Geometries"
_ICLR.cc/2025/Conference — Submitted to ICLR 2025_

### Official Review · Reviewer_piKA · 2024-10-15

**Soundness:** 3
**Presentation:** 3
**Contribution:** 2
**Rating:** 5
**Confidence:** 4

**Summary:**

The paper focuses on designing deep neural networks operating on general Riemannian manifolds, and in particular generalizing fully connected and convolutional layers to Riemannian geometries. Previous work in this domain has often been limited to specific manifolds or has relied heavily on the properties of certain geometric structures, limiting flexibility. The paper proposes a more general and flexible framework for constructing FC and convolutional layers that operate on general geometries, allowing broader applicability across different types of Riemannian manifolds. The paper demonstrates their framework primarily over the Symmetric Positive Definite (SPD) and Grassmannian manifolds.

**Strengths:**

The groundwork for defining neural network architectures over general geometries is both theoretically interesting and provides a practical blueprint for future work designing such architectures. For manifolds where we cannot afford to (or where it may not make sense to) leave the manifold, such purely intrinsic formulations are crucial. Although components like the exponential map (let alone the logarithmic map) can often be intractable to compute, this work sets a valuable precedent for future endeavors in designing neural network architectures over specific manifolds in practice. The presented experiments also offer a broad array of baselines, with notable improvements over prior work, particularly on HDM05. Additionally, the framework's flexibility to vary the latent geometry could have significant implications for understanding and comparing geometric representations in neural networks.

**Weaknesses:**

(roughly ordered by importance)

- **Theorem statements should be more self-contained**. For example, when reading Theorem 4.1, it appears to be merely defining the Riemannian FC layer, which leads to confusion about what is actually being proven. The lack of clarity here makes the theoretical contributions less accessible, reducing the impact of the formalism.

- **Flexibility claim for Grassmannian networks**. The main example manifold used to demonstrate the framework is the Grassmannian, where the authors claim that their layer uniquely allows the flexibility typical of MLP and CNN layers (i.e., modifying output dimensions like subspace dimension, ambient dimension, and channel count). While this is technically true, it is unclear why this flexibility cannot be achieved with simple modifications to previous architectures (e.g. basic lifting/copying maps and truncation), and there is no empirical evidence directly demonstrating the practical benefit of this added flexibility. This weakens the argument that the proposed method offers a clear advantage over prior approaches.

- **Marginal improvements over baselines**. While there is a non-negligible improvement on HDM05, the gains over other baselines are mostly marginal. Given that all of the listed manifolds have been extensively studied in their own right and with respect to neural network architectures (as well-documented by the authors), it remains unclear why practitioners should adopt the authors' framework rather than leveraging the extensive existing literature that exploits the structure of these highly structured manifolds. Furthermore, since this method relies on Riemannian logarithmic and exponential maps (which are often intractable or computationally prohibitive for general Riemannian manifolds) there is insufficient justification for the viability of this framework for designing neural network architectures over general manifolds. It would be nice to see an example manifold where NN architectures haven't been developed yet, or extensive ablations (with significant performance improvement) on one of the demonstrated manifolds to demonstrate the practical use of the flexibility in this framework.

**Questions:**

- Could you clarify why this flexibility on the Grassmannian cannot be achieved through simple modifications of existing architectures (or why they would be ineffective)? Furthermore, do you have any empirical evidence demonstrating that this added flexibility provides a significant practical benefit compared to previous approaches?

- Are there any manifolds where (a) there is a practical benefit for learning features on this manifold in the broader ICLR community, (b) your framework provides a concrete guideline to designing a NN architecture on this manifold, and (c) NN architectures have not been studied before on this manifold?

---

> ### Author Response · Authors · 2024-11-29
> **Response to Reviewer piKA (Part 1)**
>
> We thank reviewer $\textcolor{red}{piKA}$ for the careful review. Below is our detailed response. 😄
>
> ***
> ## **1. Thm. 4.1 --> Def. 4.1 \& Thm. 4.2**
>
> Thank you for the suggestive comments. We have revised Sec. 4.1 in our updated manuscript. In the revised paper, we first extend the reformulation of the Euclidean FC layer [a, Sec. 3.2] to general manifolds and provide the general definition in Def. 4.1. Subsequently, we demonstrate that Def. 4.1 can be solved using our proposed method, which is presented in Thm. 4.2 (corresponding to the previous Thm. 4.1).
>
> ***
> ## **2. Flexibility: Comparison against Modified Previous Grassmannian Layers**
>
> As shown in Tab. 2, previous Grassmannian layers inherently allow changes at most to the ambient dimension. We further conduct additional ablations on modifying previous layers, including channel aggregation and truncation \& lifting for the subspace dimension. However, these modifications do not result in significant improvements for the previous layers and, in most cases, lead to even worse performance.  The primary reason is that these modifications fail to respect the latent geometry, potentially leading to unsatisfactory results.
>
> ### **Modification 1: Aggregation of the Channel Dimension**
>
> To enable the previous layers to reduce the channel dimension, we implemented an aggregation layer after these layers. The aggregation was performed using the Fréchet mean or the extrinsic mean [b] along the channel dimension.
>
> The 5-fold average results are presented in **Tab. A**. We observe that none of the three previous networks—GrNet, GyroGr, and GyroGr-Scaling—benefit from aggregation. The aggregation often degrades overall performance. Furthermore, we find that using the Fréchet mean for aggregation causes GrNet to produce NaN values. Consequently, we report N/A for this outcome.
>
> **Table A:** Comparison against previous layers using channel aggregation on the Radar dataset. Here, $C_{\text{in}}$ and $C_{\text{out}}$ represent the input and output channel dimensions, respectively.
> |     Method     |   Aggregation  | $[C _{in}, C _{out}]$ | Acc  |
> |:--------------:|:--------------:|:-------------:|:------------:|
> |      GrNet     |      None      |     [8,8]     | 90.48 ± 0.76 |
> |                |  Fréchet Mean  |     [8,1]     |      N/A     |
> |                | Extrinsic Mean |     [8,1]     | 86.40 ± 1.11 |
> |     GyroGr     |      None      |     [8,8]     | 90.64 ± 0.57 |
> |                |  Fréchet Mean  |     [8,1]     | 85.44 ± 0.95 |
> |                | Extrinsic Mean |     [8,1]     | 86.94 ± 1.04 |
> | GyroGr-Scaling |      None      |     [8,8]     | 88.88 ± 1.52 |
> |                |  Fréchet Mean  |     [8,1]     | 78.86 ± 1.32 |
> |                | Extrinsic Mean |     [8,1]     | 78.77 ± 1.29 |
> |  GrConvNetONB  |       N/A      |     [8,1]     | **95.23 ± 0.96** |
> |   GrConvNetPP  |       N/A      |     [8,1]     | 94.56 ± 0.58 |
>
> (To be continued)

---

> > ### Author Response · Authors · 2024-11-29
> > **Response to Reviewer piKA (Part 2)**
> >
> > ### **Modification 2: Truncation and Lifting for the Subspace Dimension**
> >
> > We apply truncation to reduce the subspace dimension by dropping the last $k$ columns. Since copying does not preserve the Grassmannian structure (as it contradicts orthogonality), we use the following to achieve lifting:
> > \begin{equation}
> > Y = \mathcal{Q}(W _1 X W _2 ^\top),
> > \end{equation}
> > where $W _1$ and $W _2$ are row-wise orthogonal matrices, and $\mathcal{Q}(\cdot)$ returns the column-wise orthogonal matrix from the QR decomposition.
> >
> > We focus on the GrNet backbone and present the 5-fold results on the Radar dataset in Tab. B. We observe that numerical tricks such as truncation and lifting do not improve the performance of the GrNet backbone network. The primary reason is that these numerical tricks fail to respect the complex latent Grassmannian geometry. In contrast, our proposed layers naturally respect the latent geometry as they are directly induced by Riemannian operators.
> >
> >
> > **Table B:** Comparison against truncation and lifting on the Radar dataset.
> > |     Method    | Subspace dims | Ambient dims |      Acc     |
> > |:-------------:|:-------------:|:------------:|:------------:|
> > |  GrNet  |     [4,4]     |    [20,16]   | 90.48 ± 0.76 |
> > | GrNet-Lifting |     [4,4]     |    [20,16]   | 90.38 ± 0.98 |
> > |               |     [4,6]     |    [20,16]   | 88.45 ± 0.88 |
> > |               |     [4,8]     |    [20,16]   | 87.91 ± 0.63 |
> > |  GrNet-Trunc  |     [4,2]     |    [20,16]   | 87.14 ± 1.03 |
> > |               |     [4,3]     |    [20,16]   | 88.39 ± 0.32 |
> > |               |     [6,4]     |    [20,16]   | 88.42 ± 0.60 |
> > |               |     [8,4]     |    [20,16]   | 88.53 ± 0.56 |
> > |  GrConvNetONB |     [4,4]     |    [20,16]   | 93.92 ± 0.74 |
> > |               |     [4,6]     |    [20,16]   | **95.23 ± 0.96** |
> > |               |     [4,8]     |    [20,16]   | 94.77 ± 0.81 |
> > |  GrConvNetPP  |     [4,4]     |    [20,16]   | 94.35 ± 0.42 |
> > |               |     [4,6]     |    [20,16]   | 94.51 ± 0.53 |
> > |               |     [4,8]     |    [20,16]   | 94.11 ± 0.58 |
> >
> > ### **Empirical Evidence for the Flexibility in Grassmannian FC Layers**
> >
> > **Channel dimension:** Reducing the channel dimension can significantly decrease the number of parameters. Besides, as shown in Tab. A, our transformation layer outperforms the previous layer under the setting of $[C_{\text{in}}, C_{\text{out}}] = [8, 1]$ by up to **8.29\%**. Even when comparing our layers under $[8, 1]$ against the previous layers under $[8, 8]$, the improvement can still reach up to **4.59\%**.
> >
> > **Subspace and ambient dimensions:** As shown in Tab. B (or Tab. 4 in the main paper), the flexibility in these two dimensions can further improve the performance of GrConvNet. The best results are achieved by ONB GrConvNet under the settings of [8,1], [4,6], and [20,16] for the channel, subspace, and ambient dimensions, respectively, where all three dimensions are changed.
> >
> > ***
> > ## **3. Our Riemannian Transformation Layer Enjoys Generality**
> >
> > Our FC layer in Thm. 4.2 only requires the operators such as Riemannian log and exp. Given a Riemannian manifold, one only needs to put the required operators into Thm. 4.2, and then can obtain the FC layer for the specific metric. If closed-form expressions are unavailable, we can rely on approximation algorithms for computation, such as [c] for Lie groups.
> >
> > However, most matrix or vector manifolds in machine learning typically have closed-form formulations for Riemannian computations. A plethora of examples can be found in [d]. In our implementation, we have showcased our FC layers under **nine concrete geometries**: five for the SPD manifold, two for Grassmannian, and two for the hyperbolic paces (App. J in the revised paper).

---

> > > ### Author Response · Authors · 2024-11-29
> > > **Response to Reviewer piKA (Part 3)**
> > >
> > > ***
> > > ## **4. Potential Applications to Other Manifolds**
> > >
> > > **General guideline:** Various network architectures consist of three basic layers: transformation, activation, and classification. By incorporating manifold counterparts for these three layers, various network architectures can be generalized to manifolds. Among these counterparts, activation is the simplest. Since the manifold itself is already highly non-linear, activation can either be omitted or implemented via the tangent space. For the other two more challenging layers, classification can be constructed using Thm. 3.1, which comes from [e, Thm. 2]. Transformation can be implemented using our Riemannian FC layer (Thm. 4.2) or convolutional layers (Sec. 4.2).
> > >
> > > **Potentially unexplored geometries:**
> > > - **Correlation manifolds:** Alongside the SPD manifold, the correlation matrix is a natural alternative to the covariance matrix in statistical analysis. Correlation matrices are compact representations of covariance matrices that focus on scale-invariant information. Recently, correlation matrices have been identified with various Riemannian structures [f]. More importantly, many of these geometries offer closed-form expressions for Riemannian operators, such as the Riemannian exponential and logarithm maps. Therefore, our Riemannian FC layer in Thm. 4.2 can be readily applied to the correlation manifold to construct networks.
> > >
> > > - **Hyperbolic spaces:** Although hyperbolic spaces have been widely explored, most existing work focuses on the Poincaré ball and Hyperboloid models. However, there are five models over the hyperbolic space [g]. Our method enables the direct construction of Riemannian networks under other hyperbolic geometries, extending beyond the commonly used ones.
> > >
> > > - **Other possible manifolds:** A plethora of geometries is summarized in [d], many of which have primarily been applied to optimization rather than network design. Most of these manifolds are well-studied and feature closed-form expressions for the involved Riemannian operators. Consequently, our Riemannian FC layer in Thm. 4.2 can also be readily constructed on these manifolds.
> > >
> > > ## References
> > >
> > > > [a] Hyperbolic neural networks++
> > > >
> > > > [b] Bayesian and geometric subspace tracking
> > > >
> > > > [c] A reduced parallel transport equation on Lie groups with a left-invariant
> > > metric
> > > >
> > > > [d] https://www.manopt.org/manifolds.html
> > > >
> > > > [e] RMLR: Extending Multinomial Logistic Regression into General Geometries
> > > >
> > > > [f] Theoretically and computationally convenient geometries on full-rank correlation matrices
> > > >
> > > > [g] Hyperbolic geometry

---

### Official Review · Reviewer_qFyX · 2024-10-21

**Soundness:** 4
**Presentation:** 3
**Contribution:** 3
**Rating:** 6
**Confidence:** 3

**Summary:**

The authors introduce a framework for constructing fully connected and convolutional layers in contexts where one wants to construct a deep network with manifold-supported data and/or outputs. The framework for deriving such layers is general and only requires that the input and output spaces are Riemannian manifolds. For experiments, the authors specialize to several different geometries on the symmetric positive definite cone and the Grassmanian manifold, at which point they show improvements over previous constructions of such layers.

**Strengths:**

- The main constructions of the layers are relatively intuitive, make sense, and natural to arrive at when starting at the analogous objects in Euclidean spaces.
- The cookbook looks sufficiently general as to be useful to others in the community as a starting point for formulating their own architectures.
- The experimental results on different geometries are comprehensive relative to similar previous work, and show improvement over prior work.
- The paper is nicely written; the main intuitions are conveyed through clear writing and useful examples.

**Weaknesses:**

- The Riemannian convolution may not be efficient. Part of why (Euclidean) convolutions are popular is that they can be efficiently computed by a GPU. Is there any possibility that the convolution may share this property (say for the SPD/Grassmanian examples)?
- More generally, it seems difficult efficiently compute/approximate a forward pass of a Riemannian feed-forward/conv layer on more general manifolds, since the forward pass uses exponential maps and logarithms. It seems that to get efficiently computable solutions one needs to do more ad-hoc/specialized work on top of the presented framework.

**Questions:**

- Is there any intuition about why the layers formulated in the paper perform better than previous ad-hoc approaches? What practical benefits are gained by approaching this problem more generally?
- Is there any remedy when the exponential map is only locally defined? There seems to be an example in the appendix but is there any more general solution?

---

> ### Author Response · Authors · 2024-11-29
> **Response to Reviewer qFyX (Part 1)**
>
> We thank Reviewer $\textcolor{purple}{qFyX}$ for valuable comments. Below, we address the comments in detail. 😄
>
> ***
> ## **1. Riemannian Convolution and GPUs Acceleration**
>
> ### **Matrix Manifolds: GPUs \& Dimensions**
>
> As the Riemannian computations on matrix manifolds require SVD-based matrix functions, CPUs could normally outperform GPUs. However, we found that our convolution benefits significantly from GPUs acceleration when the input matrices have relatively small dimensions ($\leq 30$), which is the case in all our implementations.
>
> **SPDConvNets:** The input dimensions of the SPD matrices for our SPDConvNets are [20,20], [28,28], and [28,28] on the Radar, HDM05, and FPHA datasets, respectively. Under these small dimensions, we observed that GPUs can greatly accelerate training, reducing the average training time by up to four times.
>
> **GrConvNet:** The input dimensions of the Grassmannian matrices for our GrConvNet are [20,p] on the Radar dataset, with $p = 4, 8, 6$. Similarly, we found that GPU acceleration significantly reduces training time.
>
> However, for relatively large dimensions, CPUs are generally faster than GPUs. For example, previous SPD networks modeled each raw signal into a relatively large global covariance matrix ([93,93] for the HDM05 and [63,63] for the FPHA datasets). In such cases, Riemannian computations are more efficient on CPUs.
>
> ### **Traning Efficiency**
>
> **Tab. A** summarizes the average training time for each SPD network on the most efficient device. Due to the fast and simple computation under LEM, PEM, and LCM, our SPDConvNet models based on these three metrics are significantly more efficient than those based on AIM and BWM, whose computations are more complex. Particularly, SPDConvNet-LCM is the most efficient one among the five SPDConvNets. It outperforms several prior SPD baselines in terms of training time and even achieves comparable efficiency against the simplest SPDNet. However, SPDConvNet delivers noticeable performance improvements, surpassing SPDNet by up to **5.02%**, **16.59%**, and **6.24%** on the Radar, HDM05, and FPHA datasets, respectively.
>
> **Table A**: Training efficiency (seconds/epoch) of different SPD networks.
> |      Method     | Radar |  HDM05  |  FPHA  |
> |:---------------:|:-----:|:-------:|:------:|
> |      SPDNet     | 0.66  |  0.50   |  0.28  |
> |     SPDNetBN    | 1.25  |  0.94   |  0.58  |
> |  SPDResNet-AIM  | 0.96  |  1.23   |  0.69  |
> |  SPDResNet-LEM  | 0.77  |  0.55   |  0.25  |
> | SPDNetLieBN-AIM | 1.21  |  1.15   |  0.97  |
> | SPDNetLieBN-LCM | 1.10  |  1.11   |  0.59  |
> |    SPDNetMLR    | 0.96  |  5.46   |  6.36  |
> |  SPDConvNet-LEM | 0.86  |  0.74   |  0.74  |
> |  SPDConvNet-AIM | 5.09  | 101.80  | 51.14  |
> |  SPDConvNet-PEM | 1.09  |  7.10   |  1.57  |
> |  SPDConvNet-LCM | 0.65  |  0.59   |  0.53  |
> |  SPDConvNet-BWM | 6.07  | 110.51  | 56.07  |
>
> ### **Vector Manifolds and GPU Acceleration**
>
> When it comes to vector manifolds, such as the hyperbolic manifold, their Riemannian computations are decomposition-free, allowing them to benefit significantly from GPU acceleration. During the rebuttal, we further implemented our Riemannian transformation under two hyperbolic geometries and found that GPUs are significantly faster than CPUs. For more details, please refer to **CQ#1** in the common response.
>
> ***
> ## **2. Riemannian FC Layers on Manifolds with Implicit Structures**
>
> Thank you for the insightful comments! Our FC layer in Thm. 4.2 requires the operators such as Riemannian log and exp. If a Riemannian metric has explicit expressions for these operators, the final expression can often be further simplified, as demonstrated in App. F for the SPD manifolds, Thms. 6.1 and 6.2 for the Grassmannian, and Thms. J.1–J.2 for hyperbolic spaces. On the other hand, if closed-form expressions are unavailable, we can rely on approximation algorithms for computation, such as [a] for Lie groups. This could bring extra computational burden, as it is not easy to do further simplification.
>
> However, most matrix or vector manifolds in machine learning typically have closed-form formulations for Riemannian computations. A plethora of examples can be found in [b].
>
> ***
> ## **3. Advantages: Intrinsicness, Generality, and Flexibility**
>
> Compared with previous transformation layers, our Riemannian transformation offers three major advantages: it more faithfully respects the latent geometry, provides generality in handling different geometries, and offers flexibility in changing dimensionality. Please refer to **CQ#2** in the common response for more details.

---

> ### Author Response · Authors · 2024-11-29
> **Response to Reviewer qFyX (Part 2)**
>
> ***
> ## **4. Numerical Tricks for Incompleteness**
>
> In this paper, incompleteness (locally defined exponential map) occurs only for PEM and BWM on the SPD manifold. This issue can be resolved using numerical tricks, such as regularizing the eigenvalues of the input tangent vector, as discussed in Rmk. B.1 in the appendix.
>
> For the general case, we can regularize the norm of the tangent vector to ensure it lies within the domain of the exponential map. Specifically, for a given manifold $\mathcal{M}$ and $p \in \mathcal{M}$, we assume that the exponential map is well-defined over the ball $B_p(r)$ in the tangent space $T_p \mathcal{M}$:
> \begin{equation}
> B_p(r) = \lbrace v \in T_p \mathcal{M} \mid \Vert v \Vert _p < r \rbrace,
> \end{equation}
> where $\Vert  \cdot \Vert _p$ denotes the norm over $T_p \mathcal{M}$. To regularize a tangent vector $v$ for $\mathrm{Exp} _p(v)$, we define:
> \begin{equation}
> v =
> \begin{cases}
> v & \text{if } v \in B_p(r), \\\\
> \epsilon \frac{v}{\Vert v \Vert _p} & \text{otherwise,}
> \end{cases}
> \end{equation}
> where $\epsilon$ is a small positive value.
>
> ***
> ## **References**
>
> > [a] A reduced parallel transport equation on Lie groups with a left-invariant
> metric
>
> > [b] https://www.manopt.org/manifolds.html

---

> > ### Comment · Reviewer_qFyX · 2024-11-30
> > **Reply to Author Response**
> >
> > Thanks for the detailed response. In particular, your remarks here and the common questions in the general response are convincing about how the presented framework in this work is more general and faithful, a definite plus.
> >
> > However, by your own admission, the techniques used here are difficult to scale using GPUs, and the performance gains over previous methods are modest. This presents an obstacle to practical adoption of the framework, albeit one which may exist in previous manifold layers. For this reason, I keep my score recommending acceptance.

---

> > > ### Author Response · Authors · 2024-12-01
> > > **Thanks for the instant feedback!**
> > >
> > > Thank you very much for your instant feedback! Regarding GPU acceleration, I would like to clarify the following kindly:
> > >
> > > - **Vector manifolds**, such as hyperbolic ones, benefit significantly from GPU acceleration as Riemannian computations are free from matrix decompositions.
> > >
> > > - **Matrix manifolds** benefit from GPU acceleration under relatively small dimensions, as in our method and other applications such as Electroencephalography (EEG) [A-C]. The bottleneck arises because matrix decompositions in PyTorch are faster on CPUs for larger dimensions.
> > >
> > > **References**
> > >
> > > > [A] SPD Domain-Specific Batch Normalization to Crack Interpretable Unsupervised Domain Adaptation in EEG
> > > >
> > > > [B] Deep Geodesic Canonical Correlation Analysis for Covariance-Based Neuroimaging Data
> > > >
> > > > [C] MAtt: A Manifold Attention Network for EEG Decoding

---

### Official Review · Reviewer_48Wz · 2024-10-28

**Soundness:** 3
**Presentation:** 2
**Contribution:** 2
**Rating:** 5
**Confidence:** 4

**Summary:**

The paper proposes a technique for building fully-connected and convolution neural networks on Riemannian manifolds. The proposed network architecture builds on linear layers derived from Riemannian MLR [Chen et al., 2024c], and illustrates its manifestations on SPD and Grassmannian manifolds. To validate their method, the authors present experiments on matrix classification tasks.

**Strengths:**

The proposed method can work generally on Riemannian manifolds, and does not rely on special structure such as gyrovector calculus. On the quantitative benchmarks, the method outperforms prior work on matrix classification.

**Weaknesses:**

**Lack of Analysis**
Although the method seems to numerically out-perform prior work on matrix classification tasks, the paper should provide some explanation of why this may be true. The chosen baselines are known to be very sensitive to initialization and hyperparameters chosen, which make the numerical results difficult to interpret at face value. I would have liked if paper analyzed what exactly the model is learning. For example, it is surprising to see that Log-Euclidean metric seems to outperform the other metrics, even though it endows the SPD manifold with a flat structure. Does this mean that a flat structure is best for SPD matrix classification?

Is there a theoretical explanation of why the proposed method may outperform prior work which use gyrovector calculus?

**Other Riemannian Manifolds**
It would strengthen the paper to add experiments on other manifolds such as hyperbolic and spherical spaces. While the authors acknowledge their method’s inspiration from HNN++ [Shimizu et al., 2020], there are no experiments on hyperbolic space. It would have been interesting to see how the proposed method compares to the gyrovector calculus used in HNN++. One way to extend Riemannian projection layers to hyperbolic space is with horocycle projections. Prior work [A] have observed that horocycles can be used as an efficient feature map just like the Riemannian hyperplane projections proposed in this work.

**Computational Cost**
The differences in computational cost seems significant with respect to the metrics used, due to the requirement of computing log and exponential maps. What is the compute time required for each method in Table 3? Do the methods vary much in parameters used and time required? For the proposed method, do the benefits of the added computational costs outweigh the drawbacks?

**Presentation**
It is not clear to me what the purpose of Section 7 is.

[A] Yu & De Sa., 2023. Random Laplacian Features for Learning with Hyperbolic Space. ICLR 2023.

**Questions:**

- How does the computational cost compare between the different metrics used for the SPD tasks?
- How do you ensure that the input features are in a radius close enough to each layer’s base point so that the Log map is well-defined?
- Eq. 11 defines an inner product for a manifold via a linear isometry. Is there an example of this for SPD and Grassmannian spaces? Is there a canonical way to compute this isometry for general Riemannian manifolds?

---

> ### Author Response · Authors · 2024-11-29
> **Response to Reviewer 48Wz (Part 1)**
>
> We thank Reviewer $\textcolor{green}{48Wz}$ for the careful review and suggestive comments. Below, we address the comments in detail. 😄
>
> ***
> ## **1. Optimal Metric Could Vary across Datasets and Network Structures**
>
> **Matrix classification:** Although LEM performs better on all three datasets under the current method, it is difficult to conclude that LEM is the universally optimal metric for SPD matrix learning. The optimal metric could vary across datasets and tasks. Supporting evidence can be found in [a, Tabs. 1-2], [b, Tabs. 3-9], [c, Tabs. 4-5], and [d, Tab. 2].
>
> **Hyperbolic manifold:** Unlike matrix manifolds, the hyperbolic space has the concept of hyperbolicity to describe the dataset. In CQ#1 in the common response, we observe an interesting phenomenon in hyperbolic geometries: **When the hyperbolicity is high, the Poincaré ball model tends to be more effective, while the hyperboloid model is more effective in lower-hyperbolicity cases.**
>
> **Metric as a hyperparameter:** The above discussion indicates that metric is a crucial hyperparameter for Riemannian computations. Therefore, this requires the ability of the proposed methods to handle different geometries, enabling comparisons across geometries within a consistent architecture. Fortunately, our methods possess this flexibility and generality.
>
> ***
> ## **2. Advantages over the Gyrovector Calculus**
>
> **Broader applicability than the gyro framework:** Although gyro calculus has demonstrated success in various applications [e-i], it does not generally exist on all manifolds. In contrast, our Riemannian FC layer can be implemented across diverse manifolds, extending beyond those that admit gyro structures. Furthermore, as shown in Tab. 1, the three previous gyro SPD FC layers [i] are incorporated as special cases within our framework.
>
> ***
> ## **3. Theoretical Advantages over Previous Layers: Intrinsicness, Generality, and Flexibility**
>
> Compared with previous transformation layers, our Riemannian transformation offers three major advantages: it more faithfully respects the latent geometry, provides generality in handling different geometries, and offers flexibility in changing dimensionality. Please refer to **CQ#2** in the common response for more details.
>
> ***
> ## **4. Experiments on the Hyperbolic Space**
>
> We have further implemented our Riemannian FC layer over the Poincaré ball and hyperboloid models. Please refer to **CQ#1** in the common response. We also thank the reviewer for mentioning horocycles. However, it is likely to be non-trivial to generalize horocycles from Poincaré ball into general manifolds. In contrast, our methods enjoy great applicability.
>
> ***
> ## **5. Training Efficiency**
>
> **Tab. A** summarizes the average training time for each SPD network. Our SPDConvNet models based on LEM, PEM, and LCM are significantly more efficient than those based on AIM and BWM. This efficiency stems from the fast and simple Riemannian computations under LEM, PEM, and LCM, in contrast to the more burdensome computations involved in AIM and BWM. Additionally, our parameter trivialization, as discussed in Sec. 4.3, not only eliminates the need for complex Riemannian optimization but also simplifies the final FC layer expression (see App. F). As a result, SPDConvNets based on LCM, LEM, and PEM achieve training efficiency even comparable to the vanilla SPDNet while outperforming several prior SPD baselines in terms of training time.
>
> Moreover, SPDConvNet delivers noticeable performance improvements, surpassing SPDNet by up to **5.02%**, **16.59%**, and **6.24%** on the Radar, HDM05, and FPHA datasets, respectively. This highlights that SPDConvNet not only retains computational efficiency but also achieves a noticeable improvement in accuracy, offering a compelling balance between efficiency and performance.
>
> **Table A**: Training efficiency (seconds/epoch) of different SPD networks.
> |      Method     | Radar |  HDM05  |  FPHA  |
> |:---------------:|:-----:|:-------:|:------:|
> |      SPDNet     | 0.66  |  0.50   |  0.28  |
> |     SPDNetBN    | 1.25  |  0.94   |  0.58  |
> |  SPDResNet-AIM  | 0.96  |  1.23   |  0.69  |
> |  SPDResNet-LEM  | 0.77  |  0.55   |  0.25  |
> | SPDNetLieBN-AIM | 1.21  |  1.15   |  0.97  |
> | SPDNetLieBN-LCM | 1.10  |  1.11   |  0.59  |
> |    SPDNetMLR    | 0.96  |  5.46   |  6.36  |
> |  SPDConvNet-LEM | 0.86  |  0.74   |  0.74  |
> |  SPDConvNet-AIM | 5.09  | 101.80  | 51.14  |
> |  SPDConvNet-PEM | 1.09  |  7.10   |  1.57  |
> |  SPDConvNet-LCM | 0.65  |  0.59   |  0.53  |
> |  SPDConvNet-BWM | 6.07  | 110.51  | 56.07  |
>
> ***Remark:** The above discussion has been added into App. I. 4.*

---

> ### Author Response · Authors · 2024-11-29
> **Response to Reviewer 48Wz (Part 2)**
>
> ***
> ## **6. Sec. 7: Manifold Embedding as the Riemannian FC Layer**
>
> In several applications [e,i-k], a common approach for embedding Euclidean features into a manifold involves mapping the Euclidean vector to the tangent space at the origin via a linear layer, followed by the exponential map at the origin:
> \begin{equation}
> \operatorname{Exp}_E(Ax + b): \mathbb{R}^n \rightarrow \mathcal{M}, \quad \textbf{Eq.(A)}
> \end{equation}
> where $Ax + b$ denotes the linear function mapping $\mathbb{R}^n$ into $\mathbb{R}^m \cong T_E\mathcal{M}$.
>
> However, our framework provides a novel intrinsic insight. Eq. (A) is, in fact, a special case of our Riemannian FC layer (Thm. 4.2) between $\mathbb{R}^n$ and $\mathcal{M}$. By setting $\mathcal{N} = \mathbb{R}^n$ in Thm. 4.2, the Riemannian FC layer in Eq. (12) reduces precisely to Eq. (A). Therefore, beyond the existing tangent-space interpretation, Eq. (A) serves as an FC layer from the Euclidean space $\mathbb{R}^n$ to the manifold $\mathcal{M}$. This could be a natural choice when mapping Euclidean features into the manifold.
>
> ***
> ## **7. Numerical Tricks for Ill-defined Riemannian Logarithm**
>
> In this paper, the ill-definedness of the Riemannian logarithm arises only on the Grassmannian. As discussed Rmk. B. 2, the Grassmannian logarithm $\mathrm{Log}_P(Q)$ exists only if $P$ and $Q$ are not in each other's cut locus. However, this can be numerically solved, such as the extended Grassmannian logarithm [l, Alg. 1] or replaces the inverse with the Moore–Penrose inverse $(\cdot) ^\dagger$ [h]:
> \begin{gathered}
> \mathrm{Log} _ P (Q) = O \arctan (\Sigma) R^{\top} \\\\
> \left(I_n-P P^{\top}\right) Q\left(P^{\top} Q\right)^\dagger \stackrel{\text { SVD }}{:=} O \Sigma R^{\top}.
> \end{gathered}
> In our implementation, we use the latter one, as it is already successfully implemented in several applications [h,i].
>
> ***
> ## **8. On the Linear Isometry**
>
> Denote the $m$-dimensional manifold as $\mathcal{M}$, the Riemannian metric at the origin $e$ as $g _{e}$, and the standard orthonormal basis as $\lbrace e _i = (\delta _{1i}, \cdots, \delta _{mi}) ^\top \in \mathbb{R} ^m \rbrace _{i=1} ^m$. If $g _{e}$ is the standard Euclidean inner product, then $\lbrace e _i \rbrace _{i=1} ^m$ forms an orthonormal basis. Otherwise, an orthonormal basis can be obtained via a linear isometry.
>
> **Concrete examples:** Below is a summary of nine geometries across three manifolds:
> - **SPD manifolds:** Lines 1618–1624 in App. K.3 provides the linear isometries for all five metrics.
> - **Grassmannian manifolds:** As the Riemannian metric at the origin is the standard inner product, the standard orthonormal basis can be directly obtained, as presented in Eq. (136).
> - **Hyperbolic manifolds:** In App. J of the revised manuscript, we further manifest our FC layer on two hyperbolic models. As the Riemannian metrics at the origin are relatively simple, their orthonormal bases can be directly obtained, as discussed in Apps. K.8 and K.9, respectively.
>
> **Canonical method:** The general case can be solved by performing a congruence decomposition of an SPD matrix. Since the inner product $g _e$ can be identified as an SPD matrix, we can perform congruence decomposition on the SPD matrix to transform it into the identity matrix (corresponding to the standard inner product). For any tangent vectors $v, u \in T _e \mathcal{M} \cong \mathbb{R} ^m$, we have:
> \begin{equation}
> g _e(v, u) = v^\top S u = (Cv) ^\top (Cu),
> \end{equation}
> where $S = C ^\top C$ is the congruence decomposition of the SPD matrix $S$ representing $g _e$. Here, $C$ is ensured to be invertible. Then, an orthonormal basis can be obtained as $\lbrace C ^{-1} e _i \rbrace _{i=1} ^m$, with $\lbrace e _i \rbrace _{i=1} ^m$ as the familiar standard orthonormal basis.
>
> ***Remark:** All nine geometries, including five for SPD manifolds, two for the Grassmannian, and two for hyperbolic spaces, discussed in this paper does not need the canonical method, as each of them has a much simpler way to identify the orthonormal basis.*
>
> ***
> ## **References**
>
> > [a] Building neural networks on matrix manifolds: A Gyrovector space approach
> >
> > [b] RMLR: Extending multinomial logistic regression into general geometries
> >
> > [c] A Lie group approach to Riemannian batch normalization
> >
> > [d] Riemannian residual neural networks
> >
> > [e] Vector-valued distance and gyrocalculus on the space of symmetric positive definite matrices
> >
> > [f] Hyperbolic neural networks
> >
> > [g] Hyperbolic neural networks++
> >
> > [h] The Gyro-structure of some matrix manifolds
> >
> > [i] Matrix Manifold Neural Networks++
> >
> > [j] Hyperbolic graph convolutional neural networks
> >
> > [k] Modeling graphs beyond hyperbolic: Graph neural networks in symmetric positive definite matrices
> >
> > [l] A Grassmann manifold handbook: basic geometry and computational aspects

---

> ### Comment · Reviewer_48Wz · 2024-12-03
> **Author Response**
>
> Thank you for your response. The architecture design seems well-motivated, and addresses short-falls of HNN such as how it reduces to an Exp-Log chain. However, I am afraid that the experiments on hyperbolic space do not seem complete. The table should include experiments on the PubMed dataset for completeness, and a comparison to Riemannian ResNets [a], one of the few prior works that also propose a "general Riemannian" neural network architecture.
>
> [a] also allows one to use different metrics, is "intrinsic" to the underlying manifold, and can theoretically work for general manifolds.
> To me, it seems that the major difference between [a] and the proposed work is that [a] only uses Exp maps, while the paper proposes to use both Exp and Log maps. It would strengthen the paper to add this comparison for hyperbolic space, and justify the benefits of using an additional Log map.
>
> [a] Katsman, et al., 2023. Riemannian Residual Neural Networks

---

> ### Author Response · Authors · 2024-12-03
> **Additional Response to Reviewer 48Wz (Part 1)**
>
> We thank the reviewer for the suggestive comments. Below is our detailed response. 😄
>
>
> ## **1. Expanded Experiments on the Pubmed Dataset**
>
> Following your suggestion, we conducted additional experiments to compare the previously proposed transformation layers with our Riemannian FC (RieFC) layers on the Pubmed dataset. Tab. B presents the results across all four datasets.
>
> Pubmed exhibits moderate hyperbolicity ($\delta = 3.5$), where neither the Poincaré ball nor hyperboloid-based RiemFC demonstrates a significant improvement， and the hyperboloid-based RiemFC performs poorly. This aligns with our previous findings: **Poincaré ball-based methods are effective in highly hyperbolic settings, while hyperboloid-based or tangent space-based methods perform better in low-hyperbolicity scenarios.**
>
> **Table B**: Comparison of different transformation layers on the link prediction task. The hyperbolicity is denoted as $\delta$ (lower is more hyperbolic). The best two results are hightlighted in $\textcolor{red}{\textbf{red}}$ and $\textcolor{blue}{\textbf{blue}}$.
>
> |    Method   |    Geometry   |   Disease   |   Airport  |   Pubmed   |    Cora    |
> |:-----------:|:-------------:|:-----------:|:----------:|:----------:|:----------:|
> |             |               |      $\delta=0$      |      $\delta=1$     |     $\delta=3.5$    |     $\delta=11$     |
> | Möbius       | Poincaré Ball   | 75.1 ± 0.3                | 90.8 ± 0.2               | $\textcolor{red}{\textbf{94.9 ± 0.1}}$ | 89.0 ± 0.1               |
> | Poincaré FC  | Poincaré Ball   | $\textcolor{blue}{\textbf{77.8 ± 1.4}}$ | $\textcolor{red}{\textbf{94.0 ± 0.4}}$ | 94.3 ± 0.5                | 88.1 ± 0.3               |
> | RiemFC-P     | Poincaré Ball   | $\textcolor{red}{\textbf{79.2 ± 1.2}}$ | $\textcolor{blue}{\textbf{93.1 ± 0.7}}$               | $\textcolor{blue}{\textbf{94.8 ± 0.1}}$ | $\textcolor{blue}{\textbf{89.2 ± 0.6}}$               |
> | RiemFC-H     | Hyperboloid     | 71.2 ± 0.6                | 84.3 ± 1.7               | 88.2 ± 1.5                | $\textcolor{red}{\textbf{92.8 ± 0.4}}$ |

---

> ### Author Response · Authors · 2024-12-03
> **Additional Response to Reviewer 48Wz (Part 2)**
>
> ## **2. Comparison with RResNet [A]**
>
> ### Difference and Relation
>
> \begin{equation}
> \text{Residual Block: } f: \mathcal{M} \ni x  \mapsto \mathrm{Exp} _{x ^{(i-1)}}\left(\ell _i\left(x ^{(i-1)}\right)\right) \in \mathcal{M}, \quad \text{Eq. (A)}
> \end{equation}
> where $\ell _i: \mathcal{M} \rightarrow T \mathcal{M}$ generates the vector field.
>
> **RResNet can be combined with our layers.** Our proposed approach focuses on FC layers, while RResNet introduces ResBlocks. These two methods address different aspects of the problem and are not mutually exclusive; they can complement each other. Specifically, as illustrated by Eq. (A), RResNet can not perform dimensionality reduction. A transformation layer is normally required to reduce the input dimensionality before applying ResBlocks. In the original implementation [B], the hyperbolic RResNet (HResNet) uses a linear layer (`torch.nn.linear`) for dimensionality reduction, while the SPD RResNet employs BiMap. This transformation layer can be naturally replaced with our proposed FC layers.
>
> ### SPD Manifolds
>
> **Comparison on the SPD manifold:** In Tab. 3 of our main paper, we have directly compared RResNets with our SPDConvNets on the SPD manifold. Tab. C recalls the results from Tab. 3. Our SPDConvNets outperform RResNets across all three datasets. Specifically, SPDConvNets achieve improvements of **2.54, 11.04, and 5.2** on the Radar, HDM05, and FPHA datasets, respectively, when evaluated under their respective optimal metrics.
>
> **Table C**: Comparison of SPDConvNets against RResNets on the Radar, HDM05, and FPHA datasets.
> |     Methods    |     Radar    |       |       HDM05      |       |     FPHA     |   　  |
> |:--------------:|:------------:|:-----:|:----------------:|:-----:|:------------:|:-----:|
> |                |   Mean±STD   |  Max  |     Mean±STD     |  Max  |   Mean±STD   |  Max  |
> |  SPDResNet-AIM |  95.71±0.37  |  96.4 |     64.95 ± 0.82 | 66.19 | 86.63 ± 0.55 | 87.33 |
> |  SPDResNet-LEM |  95.89±0.86  | 97.07 |   70.12 ± 2.45   | 71.92 | 85.07 ± 0.99 | 86.17 |
> | SPDConvNet-LEM | 98.27 ± 0.48 | 98.93 |   **81.16 ± 0.93**   | **82.44** | **91.83 ± 0.41** |  **92.5** |
> | SPDConvNet-AIM | 97.63 ± 0.50 |  98.4 |   80.12 ± 0.78   | 81.55 | 91.57 ± 0.40 | 92.17 |
> | SPDConvNet-PEM | **98.43 ± 0.44** | **99.07** |   78.77 ± 0.45   | 79.19 | 90.33 ± 0.37 | 90.67 |
> | SPDConvNet-LCM | 97.65 ± 0.75 | 98.93 |   75.42 ± 0.95   | 76.74 | 91.33 ± 0.24 | 91.67 |
> | SPDConvNet-BWM | 96.40 ± 0.91 | 97.87 |   74.34 ± 0.86   | 75.85 | 90.03 ± 0.55 | 90.83 |
>
> ### Additional Experiments on the Hyperbolic RResNet (HResNet)
>
> As HResNet relies on horosphere projection based on the Poincaré ball model, we focus on the Poincaré ball model and integrate our RiemFC-P (based on the Poincaré ball model) into the HResNet backbone with 3 residual blocks. We evaluate RiemFC-P on HResNet under different configurations, including different numbers of horospheres and dimensions of the Poincaré ball. These experiments are conducted on the Disease and Airport datasets for the link prediction task, both of which exhibit high hyperbolicity.
>
>
> The results, as shown in Tabs. D and E, demonstrate that integrating RiemFC-P into HResNet consistently improves its performance. This highlights the advantage of using RiemFC-P to better align with the underlying hyperbolic geometry.
>
> **Table D**: Comparison of HResNet with or without RieFC-P under different settings on the Disease dataset.
> |   Horospheres  |       50       |                |                |       250      |                |                |
> |:--------------:|:--------------:|:--------------:|:--------------:|:--------------:|:--------------:|:--------------:|
> |       Dim      |        8       |       16       |       32       |        8       |       16       |       32       |
> |     RResNet    |   71.87±2.50   |   70.67±1.44   |   75.26±4.30   |   76.38±2.37   |   72.45±3.24   |   60.48±2.73   |
> | RiemFC+RResNet | **77.15±3.87** | **78.06±1.50** | **77.37±1.62** | **79.17±0.95** | **80.26±1.62** | **75.90±3.24** |
>
>
> **Table E**: Comparison of HResNet with or without RieFC-P under different settings on the Airport dataset.
> |   Horospheres  |        50        |                  |                  |        250       |                  |                  |
> |:--------------:|:----------------:|:----------------:|:----------------:|:----------------:|:----------------:|:----------------:|
> |       Dim      |         8        |        16        |        32        |         8        |        16        |        32        |
> |     HResNet    |   90.68 ± 1.72   |   92.17 ± 1.46   |   89.13 ± 2.43   |   91.04 ± 2.73   |   92.20 ± 1.68   |   90.46 ± 2.93   |
> | RiemFC+HResNet | **93.16 ± 0.35** | **93.29 ± 0.88** | **93.76 ± 0.66** | **92.90 ± 0.51** | **93.59 ± 0.32** | **93.29 ± 0.44** |
>
>
> ## **References**
>
> > [A] Riemannian Residual Neural Networks
> >
> > [B] https://github.com/KaimingHe/deep-residual-networks

---

### Official Review · Reviewer_gKrX · 2024-11-01

**Soundness:** 3
**Presentation:** 2
**Contribution:** 2
**Rating:** 3
**Confidence:** 4

**Summary:**

This work proposes a generalized Riemannian neural network by constructing a new linear layer and deriving several other constructions from it.

**Strengths:**

This work has some positive aspects

1. The theory is well motivated, and the derivations do not appear to be wrong to me.
2. The experimental results show a marked improvement over prior work in terms of empirical performance.

**Weaknesses:**

There are a few major weaknesses that prevent me from giving an accept rating for this paper.

1. The paper only compares on the SPD and Grassman manifold. Note that this paper is paper is proposing a "general framework" for constructing riemannian neural networks, so there should be an acknowledgement of other manifolds like hyperbolic space/sphere, which seem to be even more popular for experiments.
2. In my eyes, the paper does a bit of a disservice to prior work. For example, it is known from as early as Ganea that the gyro-multiplication/activation constructions are simple $\exp_0(f(\log_0(x))$ (where f is a standard linear or activation layer). If I remember correctly, this very premise was applied to more general geometries in the Skopek et al mixed curvature VAEs paper. If I also remember correctly (I believe I had derived this several years ago), the original SPDNet paper is effectively performing the same construction. It seems like a stretch to say the proposed FC layer is novel; rather, it would be more apt to say that the construction identifies a common thread in prior work while providing some references/examples of how this was done previously.
3. Building off of the previous points, beyond collecting some prior constructions under a common framework, the true contribution is the convolutional layer for practical purposes. The convolutional layer is not really a convolutional layer, but rather a linear layer between product manifolds $\mathcal{M} \to \mathcal{N}^d$. This does not seem morally right since convolution is defined in a separate way, and one can effectively view this as a hyperparameter/architecture change, which does not seem to be a good enough contribution.
4. One MAJOR GAP I see in the current construction is that there does not seem to be a mention of how to deal with activations/biases, which I looked for in the manuscript. If this is truly not talked about (which again, might be wrong since there are 32 pages of dense material where I can easily miss something in), then this seems to be a major problem. There needs to be a discussion of why the proposed construction does not simplify to $\exp_0(\mathrm{neuralnet}(\log_0(x))$, as is done in Ganea and Shizuma when they the introduce gyro-addition, which does not satisfy this form and breaks the $\exp-\log$ chain.

**Questions:**

Why is parallel transport denoted PP instead of something like PT?

---

> ### Author Response · Authors · 2024-11-29
> **Response to Reviewer gKrX (Part 1)**
>
> We thank Reviewer $\textcolor{blue}{gKrX}$ for the valuable comments. Below, we address the comments in detail. 😄
>
> ***
> ## **1. Validated Effectiveness and Flexibility on the Hyperbolic Space**
>
> We manifest our Riemannian FC layer on the Poincaré ball and hyperboloid models, respectively. The experiments validate the effectiveness and flexibility of our Riemannian FC layer. For further details, please refer to **CQ#1** in the common response.
>
> ***
> ## **2. Fundamental Differences with HNN [a, Sec. 3.2] and SPDNet [b, Eq. (1)]; Generalization of HNN++ [c, Sec. 3.2]**
>
> ### **Key Difference**
>
> \begin{align}
> \text{Möbius Transformation in HNN: }& \mathrm{Exp} _0\left(f\left(\mathrm{Log} _0(x)\right)\right), \text{ with } f \text{ as a linear map}, \\\\
> \text{BiMap Transformation in SPDNet: }& WSW^\top, \text{ with } W \text{ row-wisely orthogonal }.
> \end{align}
>
> - **Möbius \& BiMap:** The Möbius transformation in HNN is defined by tangent space, while the BiMap in SPDNet takes a special form of linear map. The former largely resorts to the local linearization by a single tangent space (typically at the identity element), while the latter has little connection to the latent Riemannian geometry, although it can preserve the SPDness. As a result, both methods fail to faithfully respect the complex latent geometry.
> - **Riemannian FC layer:** In contrast, our Riemannian FC layer (Thm. 4.2) is independent of the above two methods and is derived from the reformulation of the Euclidean FC layer. It is directly induced by the Riemannian geometry and does not resort to local linearization by a single tangent space. Its expression also greatly differs from the previous ones:
> \begin{equation}
> \mathcal{F}: \mathcal{N} \rightarrow \mathcal{M}: \operatorname{Exp} _E ^{\mathcal{M}}\left(\sum _{i=1} ^m\left(\left\langle\log _{P _i} ^{\mathcal{N}}(X), A _i\right\rangle _{P _i} ^{\mathcal{N}} B _i\right)\right), \quad \textbf{Eq. (A)}
> \end{equation}
> where each $P _i \in \mathcal{N}$ and $A _i \in T _{P _i} \mathcal{N}$ are parameters. Here $\lbrace B _i \rbrace _{i=1} ^m$ is an orthonormal basis over the tangent space at the origin $E$ of $\mathcal{M}$.
>
> ### **Connection: Möbius Transformation as a Trivial Riemannian Transformation**
>
> The tangent transformation, such as the Möbius one, is a special case of our Riemannian FC layer under trivial parameters $\lbrace P _i = E \rbrace  _{i=1} ^m$.
>
> For simplicity, let's take the Poincaré ball $\mathbb{P} _{K} ^n$ as an example. Setting all $P _k$ equal to zero vector $\mathbf{0}$ and $\lbrace B _i \rbrace _{i=1} ^m$ as the standard basis $\lbrace e _i = (\delta _{i1}, \delta _{i2}, \dots, \delta _{im}) ^\top \rbrace _{i=1} ^m$, our Riemannian FC layer are reduced to
> \begin{align}
> \mathcal{F}: \mathbb{P} _{K} ^n \rightarrow \mathbb{P} _{K} ^m:
> &\operatorname{Exp} _{\mathbf{0}}\left(\sum _{i=1} ^m\left(\left\langle\log _{\mathbf{0}}(x), a _i\right\rangle _{e_i} e _i\right)\right) \\\\
> &\stackrel{(1)}{=} \operatorname{Exp} _{\mathbf{0}}\left(\sum _{i=1} ^m \left(\left\langle\log _{\mathbf{0}}(x), \widetilde{a} _i\right\rangle e _i\right)\right) \\\\
> &\stackrel{(2)}{=} \operatorname{Exp} _{\mathbf{0}}\left( f(\mathrm{Log} _{\mathbf{0}}(x)) \right).
> \end{align}
> The last equation is exactly the Möbius transformation. Here, $\langle \cdot,\cdot \rangle$ is the standard Euclidean inner product and $f$ takes the form of $Ax$. The above comes from the following:
>
> (1) $\langle u,v \rangle _{\mathbf{0}}=4 \langle u,v \rangle$ and let $\widetilde{a} _i = 4 a _i$;
>
> (2) $Ax$ can be written element-wisely as $\langle x, \widetilde{a}_i \rangle$.
>
>
> ### **Related Work \& Revised Sec. 4.1**
>
> The most closely related work is the Poincaré FC layer in HNN++ [c, Sec. 3.2] and the three gyro SPD FC layers in [d, Sec. 3.2]. However, both approaches rely on specific properties, such as gyro structures, as discussed in Sec. 3.2. Our contributions lie in generalizing the Poincaré FC layer to general Riemannian geometries and providing a general solution. Additionally, as shown in Tab. 1, the previous three gyro SPD FC layers are incorporated into our framework as special cases.
>
> ***Remark:** To improve clarity, we have revised Sec. 4.1 to better present how the ideas in HNN++ are extended to the general case.*

---

> ### Author Response · Authors · 2024-11-29
> **Response to Reviewer gKrX (Part 2)**
>
> ***
> ## **3. Convolution within Each Receptive Field Is an FC Transformation**
>
> **From FC layer to convolution:** In each receptive field, the Euclidean convolution takes the form of the FC transformation ($Ax+b$). Let us focus on a single receptive field. Given a $c$-channel vector in a receptive field $\mathbf{x} = \text{concat}(x_1, \cdots, x_c) \in (\mathbb{R}^n)^c$ with $x_i \in \mathbb{R}^n$ as the feature vector in the $i$-th channel, the Euclidean convolution within this receptive field can be expressed as:
> $$
> \text{Conv}(\mathbf{x}) = \text{concat}\left(f^1(\mathbf{x}), \cdots, f^k(\mathbf{x})\right),
> $$
> where $f^i(\cdot) : (\mathbb{R}^n)^c \to \mathbb{R}^m$ represents the FC transformation parameterized by the $i$-th convolutional kernel. Therefore, the overall process of convolution can be characterized as: 1) splitting input features into multiple receptive fields; 2) applying the FC transformation within each receptive field; and 3) concatenating the resulting features. The above process can be easily adapted to a manifold, as long as the Riemannian FC layer is proposed. This is basically how HNN++ constructs the convolutional layer based on their proposed Poincaré FC layer [c, Sec. 3.4].
>
> **Revised Sec. 4.2:** We acknowledge that Sec. 4.2 in our previous manuscript did not clearly convey this idea, and we have revised this section in the updated paper. Additionally, we have included a figure for better illustration. Thank you for your careful review!
>
> ***
> ## **4. Biases Are Already Contained in Our Riemannian FC Layer.**
>
> In our Riemannian FC layer (Eq. (A) in this rebuttal), $\lbrace P _i \rbrace  _{i=1} ^m$ corresponds to the bias. An intuitive understanding can be gained by considering the special Euclidean case where $\mathcal{M}=\mathbb{R} ^m$ and $\mathcal{N}=\mathbb{R} ^n$. This makes our FC layer reduce to the familiar Euclidean FC layer, as shown in Prop. 4.4 and proven in App. K.2.
>
> In this case, Eq. (A) becomes:
> \begin{equation}
>     \mathcal{F}: \mathbb{R} ^n \ni x \mapsto \sum _{i=1} ^m\left(\left\langle x-p _i, a _i\right\rangle e _i\right) \in \mathbb{R} ^m,
> \end{equation}
> where $\lbrace e _i \rbrace _{i=1} ^m$ is the familiar standard orthonormal basis. Further considering our parameter manipulation discussed in Sec. 4.3:
> \begin{align}
>     p _i &= \mathrm{Exp} _{\mathbf{0}} (\gamma _i [z _i]) = \gamma _i [z _i], \\\\
>     a _i &= \Gamma _{\mathbf{0} \rightarrow p _i} (z _i) = z _i,
> \end{align}
> where each $\gamma _i \in \mathbb{R}$ and $z _i \in \mathbb{R} ^n$ are parameters, and $[z _i]$ are the unit vector of $Z _i$. Combining the above, we have:
> \begin{equation}
>     \mathcal{F}: \mathbb{R} ^n \ni x \mapsto \sum _{i=1} ^m\left( \left(\left\langle x, z _i\right\rangle  + b _i\right) e _i\right) \in \mathbb{R} ^m,
> \end{equation}
> where $b _i = -\gamma _i \Vert z _i \Vert$ comes from $p _i$. The right-hand side is the element-wise reformulation of $Zx+b$. Here, we can see that $p _i$ corresponds to the bias.
>
> For an in-depth understanding, please refer to our revised Sec. 4.1, where we technically extend the Euclidean FC layer into manifolds via the hyperplane projection.

---

> ### Author Response · Authors · 2024-11-29
> **Response to Reviewer gKrX (Part 3)**
>
> ## **5. Activation on the Manifold: from Feature Activation into Geometry Activation**
>
> **Matrix power as the non-linear activation function:** In our implementation, we use the matrix power function as the activate function in SPDConvNets, as discussed in App. I. 2. Matrix power has shown success in different applications [e,f]. As shown in [g, Sec. 3.1], the matrix power can deform the SPD Riemannian metric. An illustration can be found in [h, Fig. 1]. Intuitively speaking, matrix power can interpolate between a given metric to a metric of Log-Euclidean Metric (LEM) families. We expect this deforming effect of matrix power to activate the latent geometries, as our network is completely induced from the Riemannian geometry. Therefore, at the beginning of the network, we apply the matrix power to first activate the geometry.
>
> **Ablation on matrix power:** We further conduct ablations on different powers on the SPD manifold. We focus on SPDConvNet-LEM and SPDConvNet-AIM on the Radar dataset. The 5-fold average results are presented in Tab. B. We observe that the network benefits from a proper matrix power.
>
> **Table B**: Ablations on power activation.
> |      Power     |     -0.25    |     0.25     |      0.5     |       1      |
> |:--------------:|:------------:|:------------:|:------------:|:------------:|
> | SPDConvNet-LCM | 96.61 ± 0.65 | **97.65 ± 0.75** | 97.56 ± 0.58 | 96.71 ± 0.55 |
> | SPDConvNet-AIM | **97.98 ± 0.42** | 97.63 ± 0.50 | 97.67 ± 0.53 | 96.73 ± 0.56 |
>
>
> **Non-linearity of our Riemannian transformation:** We argue that our Riemannian transformation layer is already highly non-linear. Recalling Thms. 5.1, 6.1, and 6.2, our SPD and Grassmannian FC layers involve various kinds of non-linear matrix functions. Therefore, we did not use an additional activation function after our transformation layer. While activations, such as the matrix power, can be applied following each transformation layer, they will introduce additional computational costs. Given the inherent non-linearity of the transformation, we only employed such activations at the beginning of the network, as described in the main paper.
>
> ***
> ## **5. Our Riemannian FC Layer Does not Generally Reduce to an Exp-Log Chain.**
>
> Since the Möbius transformation is defined via the tangent space, it is natural that stacking Möbius transformations reduces to an exp-log chain. This is why gyro addition was introduced for biasing. However, this is not the case for our Riemannian FC layer. Referring to Eq. (A), stacking multiple Riemannian FC layers generally does not reduce to an exp-log chain.
>
> The primary reason lies in the presence of $\mathrm{Log} _{P _k}(X)$ instead of $\mathrm{Log} _{E}(X)$. Since $\mathrm{Log} _{P _k}$ does not generally cancel out with $\mathrm{Exp} _{E}$, the exp-log chain reduction generally does not occur in our framework.
>
> ***
> ## **6. PP --> PT.**
>
> Thanks for the careful review. We have revised this typo. In this paper, PT denotes the parallel transport, while PP denotes the projector perspective of the Grassmannian.
>
>
> ## References
>
> > [a] Hyperbolic neural networks
> >
> > [b] A Riemannian network for SPD matrix learning
> >
> > [c] Hyperbolic neural networks++
> >
> > [d] Matrix Manifold Neural Networks++
> >
> > [e] Deep CNNs meet global covariance pooling: Better representation and generalization
> >
> > [f] Fast differentiable matrix square root and inverse square root
> >
> > [g] The geometry of mixed-Euclidean metrics on symmetric positive definite matrices
> >
> > [h] RMLR: Extending Multinomial Logistic Regression into General Geometries

---

### Author Response · Authors · 2024-11-29
**Common Response (Part 1)**

We thank all the reviewers for their constructive suggestions and valuable feedback 👍.  Below, we address the common questions (CQs).

**Summary of Changes and Remarks:**
- During the rebuttal, we implemented two additional hyperbolic FC layers, which have been added to App. J. As a result, our revised paper now includes **nine FC layers spanning three manifolds under nine different geometries**: five for the SPD manifold, two for the Grassmannian, and two for the hyperbolic space.
- Sec. 4.1 (Riemannian FC layer) and Sec. 4.2 (convolutional layer) have been rewritten for improved clarity.
- All references to the main paper cited in the rebuttal correspond to the numbering in the revised version of the manuscript.

***
## **CQ#1: Experiments on the Hyperbolic FC Layers (Reviewers $\textcolor{blue}{gKrX}$, $\textcolor{green}{48Wz}$, and $\textcolor{purple}{qFyX}$)**

The Riemannian computations over the hyperbolic space are much simpler than matrix manifolds, as they do not involve any matrix decomposition. Therefore, our Riemannian FC layer in Thm. 4.2 can manifest in a plug-in manner under different hyperbolic geometries. We focus on the Poincaré ball and hyperboloid models. We compare our hyperbolic FC layers against the Möbius transformation layer [a, Sec. 3.2] and Poincaré FC layer [b, Sec. 3.2]. The Möbius transformation is defined via the tangent space at the zero vector, while the Poincaré FC layer is specifically tailored for the Poincaré ball model.

**Expressions:** We denote our Riemannian FC layers under the Poincaré and hyperboloid models as RiemFC-P and RiemFC-H, respectively. The detailed expressions are provided in Thms. J.1 and J.2 in the revised manuscript.

**Implementation:** Following the official implementation in [c], we conducted experiments on the Cora, Disease, and Airport datasets for the link prediction task. Specifically, the baseline encoder consists of two transformation layers: the first maps the input feature dimension to 16, and the second maps 16 to 16. These transformation layers can be instantiated as our hyperbolic FC layers, Möbius transformation layers, or Poincaré FC layers.

**Table A**: Comparison of different transformation layers on the link prediction task. The hyperbolicity is denoted as $\delta$ (lower is more hyperbolic).

|  Method |    Geometry   |    Disease    |   Airport  |    Cora    |
|:--------:|:-------------:|:-------------:|:----------:|:----------:|
|          |               |    $\delta=0$ |$\delta=1$  |$\delta=11$ |
|    Möbius   | Poincaré Ball |  75.1 ± 0.3 | 90.8 ± 0.2 | 89.0 ± 0.1 |
| Poincaré FC | Poincaré Ball |  77.8 ± 1.4 | **94.0 ± 0.4** | 88.1 ± 0.3 |
|   RiemFC-P  | Poincaré Ball |  **79.2 ± 1.2** | 93.1 ± 0.7 | 89.2 ± 0.6 |
|   RiemFC-H  |  Hyperboloid  |  71.2 ± 0.6 | 84.3 ± 1.7 | **92.8 ± 0.4** |

Tab. A presents the 5-fold average testing AUC results across three datasets. We have the following following key observations:

- **Effectiveness**: Our RiemFC achieves superior or comparable performance to the prior Möbius and Poincaré transformations.
- **Hyperbolicity \& Poincaré ball**: On datasets with high hyperbolicity, RiemFC and Poincaré FC transformations consistently outperform the Möbius one. Conversely, on the Cora dataset with the lowest hyperbolicity, all three Poincaré transformations perform similarly. This suggests that for highly hyperbolic data, intrinsic Riemannian transformations are more effective, as tangent Möbius transformations may distort the geometry.
- **Hyperbolicity \& flexible metrics**: On the dataset with the lowest hyperbolicity, hyperboloid-based RiemFC outperforms all other Poincaré-based layers, highlighting the importance of the underlying metric in Riemannian networks. Unlike the prior Poincaré FC layer, which is designed specifically for the Poincaré ball model, our Riemannian FC layer in Thm. 4.2 can adapt to various metrics in a plug-and-play manner. This adaptability enhances the representation power of HNNs, making them more versatile for diverse applications.

***Remark:** The above discussion has been added to App. J in our revised manuscript. Please refer to it for more details.*

---

> ### Author Response · Authors · 2024-11-29
> **Common Response (Part 2)**
>
> ***
> ## **CQ#2. Advantages: Intrinsicness, Generality, and Flexibility (Reviewers $\textcolor{green}{48Wz}$ and $\textcolor{purple}{qFyX}$)**
>
> - **Typical previous layers:**
> \begin{align}
> \text{BiMap Transformation in SPDNet [d, Eq. (1)]: }& WSW^\top, \\\\
> \text{FRMap + ReOrth in GrNet [e, Eq. (2-3)]: }& \mathcal{Q}(WX),
> \end{align}
> where $W$ is a row-wisely orthogonal parameter, and $\mathcal{Q}(\cdot)$ returns the orthogonal matrix from the QR decomposition.
>
> Compared with previous transformation layers, our Riemannian transformation offers three major advantages: it more faithfully respects the latent geometry, provides generality in handling different geometries, and offers flexibility in changing dimensionality.
>
> - **Intrinsicness:** Our Riemannian FC layer faithfully respects the underlying complex geometries.
>     - The widely used BiMap has little connection to the underlying geometry. While it numerically preserves SPDness, it is not associated with any of the five SPD Riemannian metrics (as shown in Tabs. 6-7). Similar issues exist with the FRMap + ReOrth layers for the Grassmannian, which lack connection to Grassmannian geometry (as shown in Tab. 8), apart from the preservation of orthogonality.
>     - In contrast, our SPD and Grassmannian transformation layers (Thms. 5.1, 6.1, 6.2) are fully derived from Riemannian geometries, faithfully respecting the underlying latent structures. Notably, for the SPD manifold, the concrete expressions of FC layers differ across various geometries. This distinction emphasizes a critical insight: Riemannian geometry is essential for computations on the manifold. Ignoring these vibrant Riemannian geometries might compromise the overall representation power and fidelity of the transformations.
>
> - **Generality in applicable metrics:** Our Riemannian FC layer in Thm. 4.2 can be implemented across different geometries, as it only requires explicit expressions of the associated Riemannian operators. Therefore, our method allows for direct comparison of different geometries within the same network architecture. Tabs. 3-4 indicate that the optimal metric may vary. We also observed similar findings on hyperbolic spaces, which are discussed in CQ#1. These indicate that the Riemannian metric should serve as a crucial hyperparameter in Riemannian networks. Supporting evidence can be also found in [f, Tabs. 1-2], [g, Tabs. 3-9], [h, Tabs. 4-5], and [i, Tab. 2].
>
> - **Flexibility in dimensionality:** Taking the Grassmannian $\mathrm{Gr}(p, n)$ as an example, our transformations, as summarized in Tab. 2, naturally support changes in the subspace dimension $p$, ambient dimension $n$, and channel dimension $c$. In contrast, prior layers are restricted to modifying at most the ambient dimension $n$. The experiments on the Grassmannian, shown in Tab. 4, demonstrate that this flexibility can be advantageous for network training.  While previous layers can be numerically modified to allow for more flexible dimensionality, such tricks often fail to respect the latent geometry and may lead to unsatisfactory results. The associated ablation studies are the 2nd response to Reviewer $\textcolor{red}{piKA}$.
>
> ***
> ## **References**
>
> > [a] Hyperbolic neural networks
> >
> > [b] Hyperbolic neural networks++
> >
> > [c] Hyperbolic graph convolutional neural networks
> >
> > [d] A Riemannian network for SPD matrix learning
> >
> > [e] Building deep networks on Grassmann manifolds
> >
> > [f] Building neural networks on matrix manifolds: A Gyrovector space approach
> >
> > [g] RMLR: Extending multinomial logistic regression into general geometries
> >
> > [h] A Lie group approach to Riemannian batch normalization
> >
> > [i] Riemannian residual neural networks

---

### Author Response · Authors · 2024-12-02
**Gentle Reminder (24h Left)**

Dear Reviewers,

We sincerely appreciate your dedicated time and effort in reviewing our paper. There are 24 hours left in the reviewer-author discussion period. As some reviewers have not acknowledged reading our response, we are uncertain if all concerns have been adequately addressed. If you have additional questions, we are more than willing to clarify further.

Your feedback is crucial to us, and we look forward to further discussions. 😄😄

Best regards,

Authors of paper 469

---

### Meta-Review · Area_Chair_efEu · 2024-12-20

**Metareview:**

This work studies extensions of fully connected and convolutional layers to Riemannian geometries for use in neural networks. The authors propose a unified framework for the Riemannian fully connected (FC) layer and develop specific formulations for various Riemannian manifolds, including SPD, Grassmannian, and Hyperbolic spaces. Experiments on several tasks, such as radar and MoCAP classification, suggest improvements over prior art.

The paper received three negative scores and one positive score. The primary concern raised by reviewers is the form of the solution, which is widely known—reformulating operations using a single tangent space (either at the origin of the manifold or, as done here, at a prototype point on the input manifold, followed by interpreting the result to lie on the identity tangent space of the output manifold). The idea of resortinbg to a chain of tangent spaces has been extensively utilized, and claims such as "faithfully respecting the underlying geometry" seem ambitious due to the potential for heavy distortion caused by linearization.

The AC agrees with the reviewers' concerns. While the work has merits, the final formulation is based on a well-known idea. As such, the AC, sadly, recommends **rejection**. We hope this review helps the authors identify areas for improvement. Specifically, offering intuition about the impact of changing dimensionality on manifolds (even through toy problems), developing a more elegant approach to performing convolution, and explaining the implications of using different metrics (e.g., on the SPD manifold) within the context of the algorithm would provide very valuable insights.

**Additional Comments On Reviewer Discussion:**

Reviewers raised several concerns, primarily regarding the novelty of the solution and the lack of comprehensive experiments (e.g., on hyperbolic spaces). During the author-reviewer discussion period, the authors expanded their experiments and provided explanations highlighting the efficiency of the algorithm, not only in terms of performance but also computational complexity. However, the reviewers did not change their scores, as the concerns regarding the novelty of the approach remained unresolved.

The AC agrees with the reviewers on this point and, therefore, recommends rejection.

---

### Decision · Program_Chairs · 2025-01-22

Reject